# The Global Convergence Time of Stochastic Gradient Descent in Non-Convex Landscapes: Sharp Estimates via Large Deviations

Waïss Azizian [1]  Franck Iutzeler [2]  Jérôme Malick [1]  Panayotis Mertikopoulos [3]

## Abstract

In this paper, we examine the time it takes for stochastic gradient descent (SGD) to reach the global minimum of a general, non-convex loss function. We approach this question through the lens of randomly perturbed dynamical systems and large deviations theory, and we provide a tight characterization of the global convergence time of SGD via matching upper and lower bounds. These bounds are dominated by the most "costly" set of obstacles that the algorithm may need to overcome in order to reach a global minimizer from a given initialization, coupling in this way the global geometry of the underlying loss landscape with the statistics of the noise entering the process. Finally, motivated by applications to the training of deep neural networks, we also provide a series of refinements and extensions of our analysis for loss functions with shallow local minima.

## 1 Introduction

Much of the success of modern machine learning applications hinges on solving non-convex problems of the form

$$\min_{x \in \mathbb{R}^d} f(x) \tag{Opt}$$

where $f \colon \mathbb{R}^d \to \mathbb{R}$ is a smooth function on $\mathbb{R}^d$. When $d$ is so large as to make gradient calculations computationally prohibitive, the go-to method for solving (Opt) is the *stochastic gradient descent* (SGD) algorithm

$$x_{n+1} = x_n - \eta \hat{g}_n \tag{SGD}$$

where $\eta > 0$ is the method's *step-size* – or *learning rate* – and $\hat{g}_n$, $n = 0, 1, \ldots$, is a computationally affordable stochastic approximation of the gradient of $f$ at $x_n \in \mathbb{R}^d$.

The study of (SGD) goes back to the seminal work of Robbins & Monro [74] and Kiefer & Wolfowitz [37], who introduced the method in the context of solving systems of nonlinear equations in the 1950's. Originally, the analysis of (SGD) involved a *vanishing* step-size $\eta_n$ satisfying the "$L^2 - L^1$" summability conditions $\sum_n \eta_n{}^2 < \infty = \sum_n \eta_n$, and gave rise to the ODE method of stochastic approximation. In this context, the first convergence results for (SGD) were obtained by Ljung [50, 51], Benaïm [8], and Bertsekas & Tsitsiklis [10], who established the almost sure convergence of the method in non-convex problems (with different regularity conditions for $f$). In conjunction with the above, a parallel thread in the literature launched by Pemantle [70] and Brandière & Duflo [13] showed that (SGD) avoids saddle points with probability 1, so, barring degeneracies, it only converges to local minimizers of $f$ – see also [2, 9, 28, 29, 35, 57, 58, 75] and references therein.

On the other hand, when (SGD) is run with a *constant* step-size – the standard choice in data science and machine learning – the situation is drastically different. The trajectories of (SGD) *do not converge*, but they instead wander around the problem's state space, spending most time near the critical points of $f$. This is quantified by "criticality bounds" of the form $\mathbb{E}[\sum_{k=0}^n \|\nabla f(x_k)\|^2] = \mathcal{O}(\sqrt{n})$, which certify an output with small gradient norm, in expectation or with high probability [41]. As in the vanishing step-size regime, these results are supplemented by a range of saddle-point avoidance results [24, 79] which, under certain conditions, imply that the output of (SGD) is approximately second-order optimal (and hence, in most cases, a near-minimizer).

Nevertheless, all these results for (SGD) are, at best, guarantees of *local* minimality, not global. When it comes to approximating the *global* minimum of $f$, we must tackle the following fundamental question:

*How much time does it take (SGD) to reach the vicinity of a global minimum of $f$?*

Of course, attaining the global minimum of a non-convex function is a lofty goal, so, before examining the time re-

[1]Univ. Grenoble Alpes, CNRS, Inria, Grenoble INP, LJK, 38000 Grenoble, France. [2]Institut de Mathématiques de Toulouse, Université de Toulouse, CNRS, UPS, 31062 Toulouse, France. [3]Univ. Grenoble Alpes, CNRS, Inria, Grenoble INP, LIG, 38000 Grenoble, France. Correspondence to: Waïss Azizian <waiss.azizian@univ-grenoble-alpes.fr>.

*Proceedings of the 42nd International Conference on Machine Learning*, Vancouver, Canada. PMLR 267, 2025. Copyright 2025 by the author(s).

quired to achieve it, one must first assess the likelihood of getting there in the first place. In this regard, Azizian et al. [5] recently showed that, in the long run, $x_n$ is exponentially concentrated near the local minimizers of $f$, with the degree of concentration depending on the landscape of $f$ and the noise entering the process.[1] In practice, this means that (SGD) ultimately gets arbitrarily close to arg min $f$, but before getting there, it may have spent an exponential amount of time away from arg min $f$. Consequently, any answer to the question of global convergence of (SGD) must incorporate global information about the geometry of $f$, as well as the statistics of the stochastic gradients $\hat{g}_n$.

**Our contributions.** Our aim in this paper is to provide quantifiable predictions for the global convergence time of (SGD). Building on the approach of Azizian et al. [5], we examine this question through the lens of the Freidlin–Wentzell (FW) theory of large deviations for randomly perturbed dynamical systems in continuous time [23], and we employ the subsampling theory of Kifer [38, 40] as a starting point to derive a similar theory for the discrete-time setting of (SGD). In so doing, we obtain a tight characterization for the global convergence time of (SGD) which can be expressed informally as

$$\mathbb{E}_x[\tau] \approx e^{E(x)/\eta} \qquad (1)$$

where $(i)$ $\tau$ denotes the number of iterations required to reach arg min $f$ within a given accuracy; $(ii)$ $x$ is the initialization of (SGD); and $(iii)$ $E(x)$ is an "energy function" that encodes the geometry of $f$ and the statistics of the noise in (SGD) via a construct called the "transition graph" of $f$.

The precise form of the hitting time estimate (1) is described via matching upper and lower bounds in Section 4 (cf. Theorem 1). Subsequently, in Section 5, we take an in-depth look at the impact of the loss landscape of $f$ on these bounds, and we link the energy $E(x)$ to the depth of the function's spurious, non-global minima. The resulting expression provides a crisp characterization of the global convergence time of (SGD) in terms of the geometry of $f$ – and, more precisely, the maximum relative depth of any spurious minimizers that the process encounters on its way to arg min $f$. The details of this construction rely on an intricate array of tools and techniques from the theory of large deviations and randomly perturbed dynamical systems, so they are difficult to describe here; for this reason, we begin by presenting a simplified version of the apparatus required to state our results in Section 2.

**Related work.** Owing to its importance, (SGD) and its variants have given rise to a vast corpus of literature which

cannot be adequately surveyed here. As we mentioned above, in the non-convex case, most of this literature concerns the criticality and saddle-point avoidance guarantees of the method, under different structural and regularity assumptions. For our purposes, the most relevant threads in the literature revolve around $(i)$ treating $x_n$ as a discrete-time Markov chain and examining its tails [26, 27, 69]; $(ii)$ viewing it as a discrete-time approximation of a stochastic differential equation (SDE) and employing tools like dynamic mean-field theory (DMFT) to study the resulting "diffusion approximation" limit [59, 60, 77]; and/or $(iii)$ focusing on the time it takes (SGD) to escape a spurious local minimum [7, 25, 30, 33, 61, 86]. Our analysis shares the same high-level goal as these general threads – that is, understanding the global convergence properties of (SGD) in non-convex landscapes – but we are not aware of any comparable results. To streamline our presentation, we provide a more detailed account of this literature in Appendix A.

The only thing we should highlight at this point is a range of phenomena that arise in the context of neural network training, where overparameterization and Gaussian initialization schemes can lead to global convergence [1, 19, 92]. Results of this kind typically require some specific structure on the underlying neural network: a width scaling quadratically with the data [65, 68] – or linearly for infinite-depth networks [54] – and/or initialization schemes that are attuned to the network's structure [49, 65]. By contrast, our work takes a more holistic viewpoint as we aim to obtain results for general non-convex landscapes, without making any structural assumptions about the problem's objective or the algorithm's initialization. To provide the necessary context, we survey the relevant literature on overparameterized neural networks in Appendix A.

## 2 A gentle start

Stating our results in their most general form requires some fairly involved technical apparatus, so we begin with a warmup section intended to introduce some basic concepts and develop intuition for the sequel. Specifically, our aim in this section is to give a high-level overview of our main results for a simple two-dimensional example which is easy to plot and visualize. We stress that the material in this section is presented at an informal level; the rigorous treatment is deferred to Section 4.

With this in mind, the example of (Opt) that we will work with is a modified version of the well-known "three-hump camel" test function, as detailed in Fig. 1. This is a multimodal function with five critical points, indexed $p_1$ through $p_5$: two are saddles ($p_2$ and $p_4$), three are minimizers ($p_1$, $p_3$, and $p_5$), and the global minimum is attained at $p_1 \approx (-2.573, 1.029)$. To keep things simple, we will further assume that (SGD) is run with stochastic gradients of

---

[1] Formally, Azizian et al. [5] showed that, in the limit $n \to \infty$, $x_n$ follows an approximate Boltzmann–Gibbs distribution with temperature equal to $\eta$, and energy levels determined by $f$ and the statistics of $\hat{g}_n$.

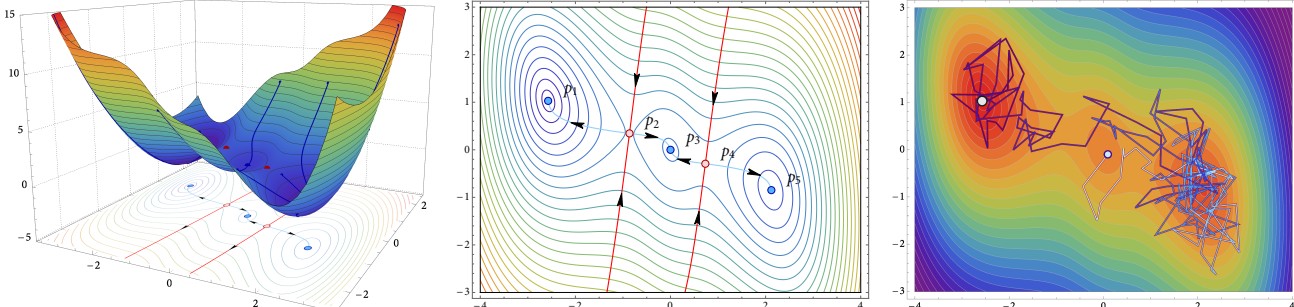

**Figure 1:** Visualization of the three-hump camel test function $f(x_1, x_2) = 2x_1^6/13 + x_1^5/8 - 91x_1^4/64 - 24x_1^3/48 + 42x_1^2/16 + 5x_2^2/4 + x_1x_2$. The figure to the left depicts the loss landscape of $f$ with several (deterministic) gradient flow orbits superimposed for visual convenience. The figure in the middle highlights the 5 critical points of $f$ (light blue for local minimizers, light red for saddle points): the red curves depict the stable manifolds of the saddle points $p_2$ and $p_4$, and they partition the space into three basins of attraction, one per minimizer; the blue curves are gradient flow orbits that realize the edges of the transition graph of $f$. Finally, the figure to the right illustrates a trajectory of (SGD) starting near $p_3$, and stopped when it gets within $10^{-2}$ of $p_1$, the global minimum of $f$; color represents time, with darker hues indicating later iterations of (SGD). The plotted trajectory gets to $p_1$ only after first being trapped by $p_5$.

the form $\hat{g}_n = \nabla f(x_n) + Z_n$, where $Z_n$ is an i.i.d. sequence of Gaussian random vectors with covariance $\sigma^2 I$. Then, fixing an initialization $x_0 \leftarrow x \in \mathbb{R}^2$, we will seek to estimate the global convergence time

$$\tau = \inf\{n = 0, 1, \dots : \|x_n - p_1\| \le \delta\} \quad (2)$$

for some fixed error margin $\delta > 0$ (e.g., $\delta = 10^{-2}$ in Fig. 1).

The core of our analysis may then be summarized as follows:

1. With exponentially high probability, (SGD) spends most of its time near the critical points of $f$ (and, in particular, its minimizers).

2. To leading order, the time that (SGD) takes to get to a neighborhood of $\arg\min f$ is determined by the chain of critical points visited by $x_n$, and by the average time required to transition from one to the next.

The technical scaffolding required to make this intuition precise is provided by a weighted directed graph which quantifies the difficulty of (SGD) making a direct transition between two critical points of $f$. In the context of our example, this graph is constructed as follows:

- The nodes of the graph are the critical points $p_i$, $i = 1, \dots, 5$, of $f$.

- Two nodes are joined by an edge if there exists a solution orbit of the gradient flow $\dot{x}(t) = -\nabla f(x(t))$ whose closure connects said points.[2] Importantly, if $p_i \rightsquigarrow p_j$, we also have $p_j \rightsquigarrow p_i$ by default.

- The weight of an edge $p_i \rightsquigarrow p_j$ is given by the expression

$$B_{ij} = 2[f(p_j) - f(p_i)]_+/\sigma^2. \quad (3)$$

___
[2]Formally, two distinct critical points $p_i$, $p_j$ of $f$ are joined by an edge if there exists a solution orbit $x(t)$ of the gradient flow of $f$ such that $\{p_i, p_j\} = \{\lim_{t \to \pm\infty} x(t)\}$.

An instructive way of interpreting this expression is as follows: if $f(p_j) \le f(p_i)$, the transition of (SGD) from $p_i$ to $p_j$ is "costless"; otherwise, if $f(p_j) > f(p_i)$, the noise in (SGD) can still lead to an *ascent* from $p_i$ to $p_j$, but the cost of such a transition is proportional to the potential difference $f(p_j) - f(p_i)$, and inversely proportional to the variance of the noise in (SGD).

In our example, this construction yields the path graph below (where, for visual clarity, the height of each node corresponds to its objective value):

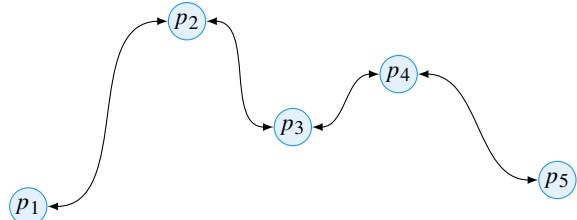

The topology of this graph captures the fact that the entire space is partitioned into the basins of attraction of $p_1$, $p_3$ and $p_5$, as shown in Fig. 1. Since $p_1$ and $p_5$ are separated by neighborhoods, there can be no gradient flow orbits joining "non-successive" critical points – e.g., $p_1$ to $p_3$ – leading to the path graph structure depicted above.

To proceed, we fix an initial condition $x$ for (SGD), say, in the basin of attraction of $p_3$. In this case, the most likely event is that (SGD) will first be attracted to $p_3$ on its way to the global minimum $p_1$, so, for simplicity, we just estimate the time it takes (SGD) to reach $p_1$ from $p_3$. However, this time cannot be determined only by the local geometry of $f$ along the "direct" transition path $p_3 \rightsquigarrow p_2 \rightsquigarrow p_1$: with positive probability, (SGD) may first jump over $p_4$ and be trapped in $p_5$. In that case, the process will first have to escape from $p_5$, and then follow the "indirect" transition

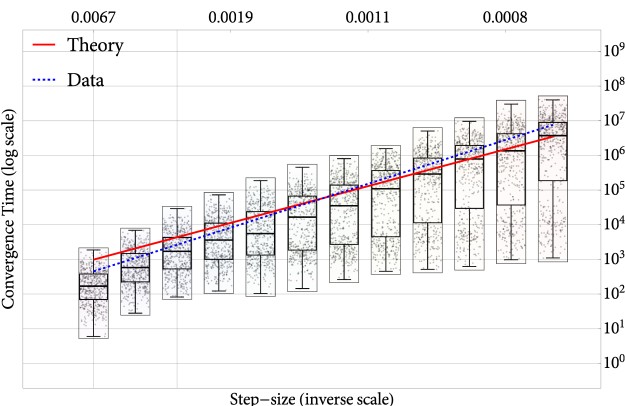

**Figure 2:** Statistical analysis of the number of iterations $\tau$ required for (SGD) to reach the global minimum of the three-hump camel function of Fig. 1. For each value of $\eta$, we performed 500 runs of (SGD) initialized near $p_3$ with Gaussian noise ($\sigma = 50$), and we recorded the number of iterations required to reach the vicinity of $p_1$ (within $\delta = 10^{-2}$). The resulting boxplots are displayed in log-inverse scale, with a point density overlay to illustrate the full empirical distribution of $\tau$ for every $\eta$. As per (4), we see that $\mathbb{E}[\tau]$ scales exponentially with $1/\eta$. To measure the goodness-of-fit, the dashed blue "Data" line represents a linear regression fit for our data, while the solid red "Theory" line is a linear fit with slope given by (4). The two fits have $R^2$ values of 0.99 and 0.97 respectively, indicating a strong agreement between theory and experiment.

path $p_5 \rightsquigarrow p_4 \rightsquigarrow p_3 \rightsquigarrow p_2 \rightsquigarrow p_1$. As a result, the mean global convergence time of (SGD) is not affected solely by the obstacles that lie on the most "direct" path from the initialization to the global minimum of $f$, but also by the traps that the process can fall into along every other "indirect" path as well.

With all this in mind, when instantiated to the example at hand, the general analysis of Section 4 yields the following expression for the mean convergence time of (SGD):

$$\mathbb{E}_x[\tau] \approx \exp\left(\frac{2(f(p_2) - f(p_5))}{\eta\sigma^2}\right). \quad (4)$$

This expression shows that the global convergence time of (SGD) scales exponentially with the inverse of the step-size and the variance of the noise, and it is dominated by the objective value gap that $x_n$ must clear if it is trapped by $p_5$. We verify experimentally the accuracy of this derivation in Fig. 2, where we measure $\tau$ over several runs of (SGD) for different values of $\eta$.

To provide some more context for all this, the precise expression for the coefficient of $1/\eta$ in (4) – which we denote as $E(p_1|p_3)$ in Section 4 – is obtained by aggregating the graph weights $B_{ij}$ according to the formula

$$\begin{aligned} E(p_1|p_3) &= B_{32} + B_{54} - \min\{B_{34}, B_{54}\} \\ &= \frac{2\max\{f(p_2) - f(p_3), f(p_2) - f(p_5)\}}{\sigma^2}. \quad (5) \end{aligned}$$

In words, the exponent $E(p_1|p_3)$ of (4) scales with the largest value gap that $x_n$ may need to clear in order to reach $\arg\min f$. In the specific example of Fig. 1, we have $f(p_3) > f(p_5)$, so $f(p_2) - f(p_5) > f(p_2) - f(p_3)$, and hence

$$E(p_1|p_3) = 2[f(p_2) - f(p_5)]/\sigma^2. \quad (6)$$

This is precisely the obstacle that (SGD) must clear if it gets trapped by $p_5$, hence our intuition (cf. Fig. 1).

All this may sound paradoxical at first: (4) shows that the global convergence time of (SGD) is not driven by the "shortest", *least costly* chain of transitions leading to $\arg\min f$, but by the "longest", *most costly* one. In this sense, $E(p_1 \mid p_3)$ directly encodes all non-local information of $f$ that ends up affecting the global convergence time of (SGD), so it can be seen as an explicit measure of the "hardness" of the non-convex landscape under study: in particular, a slight increase this gap would end up having an *exponential impact* on the global convergence time of (SGD), indicating the difficulty of learning even in this simple example

We find this characterization quite appealing because it allows us to quantify the features of (Opt) that make it harder or easier as a global problem. We will revisit this question several times in the sequel.

## 3 Problem setup and blanket assumptions

In this section, we begin our formal treatment of the global convergence time of (SGD). As a first step, we present below our standing assumptions for (Opt) and (SFO): these assumptions are not the weakest possible ones, but they lead to a more streamlined presentation; for a more general treatment, cf. Section 5 and Appendix B.

**Assumptions on the objective.** To begin, we assume throughout that $\mathbb{R}^d$ is equipped with the Euclidean inner product and norm, denoted by $\langle \cdot, \cdot \rangle$ and $\|\cdot\|$ respectively. We then make the following standing assumptions for $f$:

**Assumption 1.** The objective function $f\colon \mathbb{R}^d \to \mathbb{R}$ of (Opt) is $C^2$-smooth and satisfies the conditions below:

(a) *Coercivity:* $\lim_{\|x\|\to\infty} f(x) = \infty$.

(b) *Gradient norm coercivity:* $\lim_{\|x\|\to\infty}\|\nabla f(x)\| = \infty$.

(c) *Lipschitz smoothness:* the gradient of $f$ is $\beta$-Lipschitz continuous, i.e.,

$$\|\nabla f(x') - \nabla f(x)\| \le \beta\|x' - x\| \quad \text{for all } x, x' \in \mathbb{R}^d.$$

(d) *Critical set regularity:* The critical set

$$\operatorname{crit} f := \{x \in \mathbb{R}^d : \nabla f(x) = 0\}$$

of $f$ consists of a finite number of closed, disjoint, smoothly connected components $\mathcal{C}_i$, $i = 1, \ldots, N_{\text{crit}}$.

*Remark.* By "smoothly connected", we mean here that any two points in such a component can be joined by a smooth path contained therein. In the sequel, we will refer to these components simply as the *critical components* of $f$, and we will say that $C_i$ is a (locally) *minimizing component* of $f$ if $C_i = \arg\min_{x \in \mathcal{U}} f(x)$ for some neighborhood $\mathcal{U}$ of $C_i$. §

The requirements of Assumption 1 are fairly mild and quite standard in the literature. In more detail, Assumption 1(*a*) simply ensures that arg min $f$ is nonempty – otherwise, the question of global convergence may be meaningless. Similarly, Assumption 1(*b*) is a stabilization condition aiming to exclude functions with a specious, near-critical behavior at infinity – such as $f(x) = \log(1 + x^2)$. Finally, in terms of regularity, Assumption 1(*c*) is the go-to hypothesis for the analysis of gradient methods, while Assumption 1(*d*) rules out problems with pathological critical sets (such as Warsaw sine curves, Cantor staircase functions, etc.).[3]

**Assumptions on the gradient input.** Our second set of assumptions concerns the stochastic gradients $\hat{g}_n$ that enter (SGD). Here we will assume that the optimizer has access to a *stochastic first-order oracle* (SFO), that is, a black-box mechanism which returns a stochastic approximation of the gradient $\nabla f(x)$ of $f$ at the queried point $x \in \mathbb{R}^d$. Formally, an SFO is a random vector of the form

$$\mathsf{G}(x; \omega) = \nabla f(x) + \mathsf{Z}(x; \omega) \qquad \text{(SFO)}$$

where:

(*a*) The *random seed* $\omega$ is drawn from a compact subset $\Omega$ of $\mathbb{R}^m$ based on some probability measure $\mathbb{P}$.

(*b*) The *error term* $\mathsf{Z}(x; \omega)$ captures all sources of randomness and uncertainty in (SFO).

With all this in place, we will assume that (SGD) is run with stochastic gradients of the form

$$\hat{g}_n = \mathsf{G}(x_n; \omega_n) \qquad \text{(SG)}$$

where $\mathsf{G}$ is an SFO for $f$ and $\omega_n \in \Omega$, $n = 0, 1, \ldots$, is a sequence of i.i.d. random seeds as above. This well-established model accounts for all standard implementations of (SGD), from minibatch sampling to noisy gradient descent and Langevin-type methods. The only extra assumptions that we will make are as follows:

**Assumption 2.** The error term $\mathsf{Z}: \mathbb{R}^d \times \Omega \to \mathbb{R}^d$ of (SFO) has the following properties:

(*a*) *Properness:* $\mathbb{E}[\mathsf{Z}(x; \omega)] = 0$ and $\text{cov}(\mathsf{Z}(x; \omega)) > 0$ for all $x \in \mathbb{R}^d$.

(*b*) *Smooth growth:* $\mathsf{Z}(x; \omega)$ is $C^2$-smooth and satisfies

---

[3]This last requirement can be replaced by positing for example that $f$ is semi-algebraic, cf. [5, 15, 76].

the growth bound

$$\sup_{x, \omega} \frac{\|\mathsf{Z}(x; \omega)\|}{1 + \|x\|} < \infty. \qquad (7)$$

(*c*) *Sub-Gaussian tails:* The tails of $\mathsf{Z}$ are bounded as

$$\log \mathbb{E}\left[ e^{\langle p, \mathsf{Z}(x; \omega) \rangle} \right] \leq \frac{1}{2}\overline{\sigma}^2 \|p\|^2 \qquad (8)$$

for some $\overline{\sigma} > 0$ and for all $p \in \mathbb{R}^d$. §

Assumption 2(*a*) is standard in the field as it ensures that the oracle $\mathsf{G}(x; \omega)$ provides unbiased gradient estimates; as for the ancillary covariance requirement, it serves to differentiate (SGD) from deterministic versions of gradient descent that are run with *perfect* gradients. Assumption 2(*b*) is a regularity requirement imposing a mild limit on the growth of the noise as $\|x\| \to \infty$, while Assumption 2(*c*) is a widely used bound on the tails of the noise. Importantly, even though Assumption 2(*c*) is less general than finite variance assumptions which allow for fat-tailed error distributions, it provides much finer control on the process and leads to a much cleaner presentation. We only note here that it can be relaxed further by allowing the variance parameter $\overline{\sigma}^2$ to diverge to infinity as $\|x\| \to \infty$, in a way similar to Langevin sampling under dissipativity [21, 43, 52, 53, 73]; this case is of particular interest for certain deep learning models, and we treat it in detail in Appendix B.

**An illustrative use case.** We close this section with an example intended to illustrate the range of validity of our assumptions for (SGD). In particular, we will focus on the regularized empirical risk minimization (ERM) problem

$$f(x) = \frac{1}{N} \sum_{i=1}^{N} \ell(x; \xi_i) + \frac{\lambda}{2} \|x\|^2 \qquad (\text{ERM}_\lambda)$$

where $\ell(x; \xi)$ is the loss of model $x$ against input $\xi$ (e.g., a logistic or Savage loss), the data points $\xi_i$, $i = 1, \ldots, N$, comprise the problem's training set, and $\lambda > 0$ is a regularization parameter. For example, this setup could correspond to a linear model with non-convex losses [21, 52], a neural network with smooth activations and normalization layers [46], etc. Accordingly, if we estimate the gradient of $f$ by sampling a random minibatch $\omega \subseteq \{\xi_1, \ldots, \xi_N\}$ of training data (typically a small, fixed number thereof), the corresponding gradient oracle becomes

$$\mathsf{G}(x; \omega) = \frac{1}{|\omega|} \sum_{\xi \in \omega} \nabla \ell(x; \xi) + \lambda x. \qquad (9)$$

Under standard assumptions for $\ell$ – e.g., twice continuously differentiable, Lipschitz continuous and smooth, cf. [57] and references therein – Parts (*a*)–(*c*) of Assumption 1 are satisfied automatically. The resulting error term $\mathsf{Z}(x; \omega)$ is easily seen to be uniformly bounded, so Parts (*a*)–(*c*) of Assumption 2 are likewise verified [80, Exercise 2.4].

# 4 Analysis and results

We are now in a position to state our main results for the global convergence time of (SGD) under the blanket assumptions presented in the previous section. Our strategy to achieve this mirrors the general scheme outlined in Section 2: First, at a high level, once (SGD) has reached a neighborhood of a critical component of $f$ (and, especially, a locally minimizing component), it will be trapped for some time in its vicinity. To escape this near-critical region, (SGD) may have to climb the loss surface of $f$, and this can only happen if (SGD) takes a sufficient number of steps "against" the gradient flow $\dot{x} = -\nabla f(x)$ of $f$. Thus, to understand the global convergence time of (SGD), we need to quantify these "rare events", namely to characterize (*i*) how much time (SGD) spends near a given critical component; (*ii*) how likely it is to transition from one such component to another; and (*iii*) what are the possible chains of transitions leading to the global minimum of $f$.

To do this, we employ an approach inspired by the large deviations theory of Freidlin & Wentzell [23] and Kifer [38, 40] for randomly perturbed dynamical systems on compact manifolds in continuous time. In the rest of this section, we only detail the elements of our approach that are needed to state our results in a self-contained manner, and we defer the reader to the paper's appendix for the proofs (see also Appendix B.1 for an outline of the proof).

**Generalities.** To fix notation, we will assume in the sequel that (SGD) is initialized at some fixed $x_0 \leftarrow x \in \mathbb{R}^d$, and we will write $\mathbb{P}_x$ and $\mathbb{E}_x$ (or sometimes $\mathbb{P}_{x_0}$ and $\mathbb{E}_{x_0}$) for the law of the process starting at $x_0 = x$ and the induced expectation respectively. Then, given a target tolerance $\delta > 0$, our main objective will be to estimate the time required for (SGD) to get within $\delta$ of $\arg\min f$, that is, the hitting time

$$\tau = \inf\{n \in \mathbb{N} : \text{dist}(x_n, \mathcal{Q}) \le \delta\} \quad (10)$$

where, for notational brevity, we write

$$\mathcal{Q} = \arg\min f \quad (11)$$

for the minimum set of $f$ (which could itself consist of several connected components). Throughout what follows, we will refer to $\tau$ as the *global convergence time* of (SGD), and our aim will be to characterize $\mathbb{E}_x[\tau]$ as a function of the algorithm's initial state $x_0 \leftarrow x \in \mathbb{R}^d$.

For future use, we also define here the *attracting strength* of $\mathcal{Q}$ as the maximal value of the product $\mu R^2$, where $\mu$ and $R$ are such that, for any $x \in \mathbb{R}^d$ with $\text{dist}(x, \mathcal{Q}) \le R$,

$$\langle \nabla f(x), x - x^* \rangle \ge \mu \|x - x^*\|^2 \quad (12)$$

with $x^* \in \mathcal{Q}$ denoting a projection of $x$ onto $\mathcal{Q}$. In this way, (12) can be seen as a "setwise" second-order optimality condition for the global minimum of $f$. Intuitively, the large the attracting strength of $\mathcal{Q}$, the sharper it is.

**Elements of large deviations theory.** Moving forward, the first ingredient required to state our results is the *Hamiltonian* of $\mathsf{G}$, defined here as

$$\mathcal{H}_{\mathsf{G}}(x, p) := \log \mathbb{E}_\omega[\exp(-\langle p, \mathsf{G}(x; \omega)\rangle)] \quad (13)$$

for all $x \in \mathbb{R}^d$ and all $p \in \mathbb{R}^d$. Up to the sign in the exponent, $\mathcal{H}_{\mathsf{G}}(x, p)$ is simply the cumulant-generating function of the gradient oracle $\mathsf{G}$ at $x$, so it encodes all the statistics of (SGD) at $x$. Dually, the *Lagrangian* of $\mathsf{G}$ is given by the convex conjugate of $\mathcal{H}_{\mathsf{G}}(x, p)$ with respect to $p$, that is,

$$\mathcal{L}_{\mathsf{G}}(x, v) := \mathcal{H}_{\mathsf{G}}^*(x, v) = \max_{p \in \mathbb{R}^d} \{\langle v, p \rangle - \mathcal{H}_{\mathsf{G}}(x, p)\}. \quad (14)$$

The importance of the Lagrangian (14) lies in that it provides a "pointwise" large deviation principle (LDP) of the form

$$\mathbb{P}\left(\frac{1}{n}\sum_{k=0}^{n} \mathsf{G}(x; \omega_n) \in \mathcal{B}\right) \sim \exp\left(-n \inf_{v \in \mathcal{B}} \mathcal{L}_{\mathsf{G}}(x, v)\right) \quad (15)$$

for every Borel $\mathcal{B} \subseteq \mathbb{R}^d$. In view of this LDP, the long-run aggregate statistics of the sum $S_n = \sum_{k=0}^{n-1} \mathsf{G}(x; \omega_k)$ are fully determined by $\mathcal{L}_{\mathsf{G}}$ [16]; however, even though the iterates $x_n = x_0 - \eta \sum_{k=0}^{n-1} \mathsf{G}(x_k; \omega_k)$ of (SGD) are likewise defined as a sum of SFO queries, obtaining a "trajectory-wise" LDP for $x_n$ is far more involved.

To address this challenge, the seminal idea of Freidlin & Wentzell [23] was to treat the *entire* trajectory $x_n$, $n = 0, 1, \ldots$, of (SGD) as a point in some infinite-dimensional space of curves, and to derive a large deviation principle for (SGD) directly in that space. To that end, drawing inspiration from the Lagrangian formulation of classical mechanics, the (normalized) *action* of $\mathcal{L}_{\mathsf{G}}$ along a continuous curve $\gamma: [0, T] \to \mathbb{R}^d$ is defined as

$$\mathcal{S}_T[\gamma] = \int_0^T \mathcal{L}_{\mathsf{G}}(\gamma(t), \dot{\gamma}(t)) \, dt \quad (16)$$

with the convention $\mathcal{S}_T[\gamma] = \infty$ if $\gamma$ is not absolutely continuous. In a certain sense – which we make precise in Appendix B.4 – the quantity $\mathcal{S}_T[\gamma]$ is a "measure of likelihood" for the curve $\gamma$, with lower values indicating higher likelihoods. This is the so-called "least action principle" of large deviations [23, 40], which we leverage below to characterize the most – and *least* – likely transitions of (SGD).

**The transition graph of** (SGD)**.** To achieve this characterization, we start by associating a certain *transition cost* to each pair of critical components $\mathcal{C}_i, \mathcal{C}_j$ of $f$. We will then use these costs to quantify how likely it is to observe a given chain of transitions terminating at the global minimum of $f$.

To begin, following Freidlin & Wentzell [23], the *quasi-potential* between two points $x, x' \in \mathbb{R}^d$ is defined as

$$B(x, x') := \inf\{\mathcal{S}_T[\gamma] : \gamma \in \mathcal{C}_T(x, x'; \mathcal{Q}), T \in \mathbb{N}\} \quad (17)$$

where $\mathcal{C}_T(x, x'; \mathcal{Q})$ denotes the set of all continuous curves $\gamma \colon [0, T] \to \mathbb{R}^d$ with $\gamma(0) = x$, $\gamma(T) = x'$, and $\gamma(n) \notin \mathcal{Q}$ for all $n = 1, \ldots, T - 1$.[4] By construction, $B(x, x')$ is simply the cost of the "least action" path joining $x$ to $x'$ in time $T$ and not hitting $\mathcal{Q}$ before that time. As such, the induced setwise cost is given by

$$B(\mathcal{K}, \mathcal{K}') := \inf\{B(x, x') : x \in \mathcal{K}, x' \in \mathcal{K}'\}. \qquad (18)$$

i.e., as the action of the "least costly" transition from some point in $\mathcal{K}$ to some point in $\mathcal{K}'$ which does not go through $\mathcal{Q}$ in the meantime. In this way, focusing on the critical components of $f$, we obtain the matrix of *transition costs*

$$B_{ij} := B(\mathcal{C}_i, \mathcal{C}_j) \quad \text{for all } i, j = 1, \ldots, N_{\text{crit}} \qquad (19)$$

which compactly characterizes the "ease" with which $x_n$ may jump from $\mathcal{C}_i$ to $\mathcal{C}_j$.

We are now in a position to construct the *transition graph* of (SGD), generalizing the introductory example of Section 2. Formally, this is a weighted directed graph $\mathcal{G} \equiv \mathcal{G}(\mathcal{V}, \mathcal{E}, B)$ with the following primitives:

1. A set of vertices $\mathcal{V}$ indexed by $i = 1, \ldots, N_{\text{crit}}$, that is, one vertex per critical component of $f$. [To lighten notation, we will not distinguish between the index $i$ and the component $\mathcal{C}_i$ that it labels.]

2. A set of directed edges $\mathcal{E} = \{(i, j) : i, j = 1, \ldots, N_{\text{crit}}, i \neq j, B_{ij} < \infty\}$. In words, $i \rightsquigarrow j$ if and only if the (direct) transition cost from $\mathcal{C}_i$ to $\mathcal{C}_j$ is finite.

3. Finally, to each edge $(i, j) \in \mathcal{E}$ we associate the weight $B_{ij}$.

To further streamline our presentation and ensure that the minimum set of $f$ can be reached from any initialization of (SGD), we will make the following assumption for $\mathcal{G}$:

**Assumption 3.** $B_{ij} < \infty$ for all $i, j = 1, \ldots, N_{\text{crit}}$.

This assumption is purely technical and holds automatically if there is "sufficient noise" in the process; for a detailed discussion, we refer the reader to Appendix C.4.

**Transition forests, energy levels, and prunings.** We now have in our arsenal most of the elements required to quantify the difficulty of reaching $\mathcal{Q} = \arg\min f$ from an initial state $x \in \mathbb{R}^d$. As in the warmup setting of Section 2, we begin by describing the relevant walks of (SGD) on the associated transition graph $\mathcal{G}$.

Given that we are interested in chains of transitions terminating at the target set $\mathcal{Q}$, we will refer to all nodes in $\mathcal{Q}$

as *target nodes*, and all other nodes in $\mathcal{V} \setminus \mathcal{Q}$ as *non-target nodes*.[5] We then define a *transition forest* for $\mathcal{Q}$ to be a directed acyclic subgraph $\mathcal{T}$ of $\mathcal{G}$ such that (*i*) target nodes have no outgoing edges; and (*ii*) every non-target node of $\mathcal{T}$ has precisely *one* outgoing edge.[6] In particular, this means that there exists a unique path from *every* non-target node $i \in \mathcal{V} \setminus \mathcal{Q}$ to *some* target node $j \in \mathcal{Q}$. The *energy* of $\mathcal{Q}$ is then defined to be the minimal cost over such forests, viz.

$$E(\mathcal{Q}) := \min_{\mathcal{T} \in \mathcal{G}(\mathcal{Q})} \sum_{(i,j) \in \mathcal{T}} B_{ij} \qquad (20)$$

where $\mathcal{G}(\mathcal{Q})$ denotes the set of all transition forests toward $\mathcal{Q}$ on $\mathcal{G}$. [7] In this way, the energy of $\mathcal{Q}$ represents the minimum aggregate cost of getting to $\mathcal{Q}$, and it can be seen as an overall measure of how "easy" it is to reach $\mathcal{Q}$.

To account for the initialization of (SGD), we will need to perform a "pruning" construction, whereby we will systematically delete different edges of $\mathcal{G}$ and record the impact of this deletion on the energy of $\mathcal{Q}$. Formally, given a starting node $p \in \mathcal{V} \setminus \mathcal{Q}$, we let $r = |\mathcal{V}| - |\mathcal{Q}| - 1$ denote the number of "residual nodes" $j \in \mathcal{V}$ that are neither starting ($j \neq p$) nor targets ($j \notin \mathcal{Q}$). A *pruning* of $p$ from $\mathcal{Q}$ is then defined to be a directed acyclic subgraph $\mathcal{S}$ of $\mathcal{G}$ such that

1. $\mathcal{S}$ has $r$ edges, at most one per non-target node $j \in \mathcal{V} \setminus \mathcal{Q}$.
2. Target nodes have no outgoing edges.
3. There is no path from $p$ to $\mathcal{Q}$.[8]

The *energy required to prune $p$ from $\mathcal{Q}$* is then defined as the minimal such cost, viz.

$$E(p \not\rightsquigarrow \mathcal{Q}) := \min_{\mathcal{S} \in \mathcal{G}(p \not\rightsquigarrow \mathcal{Q})} \sum_{(i,j) \in \mathcal{S}} B_{ij} \qquad (21)$$

where $\mathcal{G}(p \not\rightsquigarrow \mathcal{Q})$ denotes the set of all prunings of $p$ from $\mathcal{Q}$ in $\mathcal{G}$. Then, to complete the picture, we define the *energy of $\mathcal{Q}$ relative to $p \in \mathcal{V} \setminus \mathcal{Q}$* as

$$E(\mathcal{Q} \mid p) := E(\mathcal{Q}) - E(p \not\rightsquigarrow \mathcal{Q}) \qquad (22)$$
$$= \min_{\mathcal{T} \in \mathcal{G}(\mathcal{Q})} \sum_{(i,j) \in \mathcal{T}} B_{ij} - \min_{\mathcal{S} \in \mathcal{G}(p \not\rightsquigarrow \mathcal{Q})} \sum_{(i,j) \in \mathcal{S}} B_{ij}.$$

Finally, for a given initialization $x \in \mathbb{R}^d$, we set

$$E(\mathcal{Q} \mid x) := \max_{p \in \mathcal{V} \setminus \mathcal{Q}} [E(\mathcal{Q} \mid p) - B(x, p)]_+ \qquad (23)$$

---

[4] We should note here that the curves that go in the definition (17) of $B$ are only required to avoid $\mathcal{Q}$ at *integer* times. This is because, while a continuous curve may reach $\mathcal{Q}$ at non-integer times, we only need to exclude integer times because we are only interested in the discrete-time process $x_n$, not its continuous-time interpolations. We discuss this in detail in Appendices B.4 and C.

[5] In our case, the target nodes are simply the globally minimizing components of $f$. To streamline notation, $\mathcal{Q}$ will be viewed interchangeably as a set of points in $\mathbb{R}^d$ or as a set of nodes in $\mathcal{V}$.

[6] Equivalently, a transition forest for $\mathcal{Q}$ can be seen as a spanning union of disjoint in-trees, each converging to a target node.

[7] When $\mathcal{Q}$ is a singleton (i.e., $\arg\min f$ is connected), a transition forest for $\mathcal{Q}$ is a spanning in-tree, so our definition generalizes and extends that of Azizian et al. [5].

[8] Equivalently, a pruning of $p$ from $\mathcal{Q}$ can be seen as a spanning union of disjoint rooted in-trees, each rooted at some target node, except the one containing the starting node $p$, which is rooted at a *non-target* node.

i.e., $E(\mathcal{Q}\,|\,x)$ is the highest energy of $\mathcal{Q}$ relative to starting nodes $p \in \mathcal{V}\setminus\mathcal{Q}$ that can be reached from $x$, modulo the cost of the initial transition $x \rightsquigarrow p$. This quantity measures the difficulty of reaching $\mathcal{Q}$ from $x$, and it plays a central role in our estimates of the global convergence time of (SGD).

**The global convergence time of** (SGD). We are finally in a position to state our main estimate for the global convergence time of (SGD).

**Theorem 1.** *Suppose that Assumptions 1–3 hold and* (SGD) *is initialized at $x \in \mathbb{R}^d$. Then, given a tolerance level $\varepsilon > 0$, and small enough $\delta, \eta > 0$, we have*

$$\exp\left(\frac{E(\mathcal{Q}\,|\,x) - \varepsilon}{\eta}\right) \le \mathbb{E}_x[\tau] \le \exp\left(\frac{E(\mathcal{Q}\,|\,x) + \varepsilon}{\eta}\right)$$

*provided that the attracting strength of $\mathcal{Q}$ is large enough for the left inequality (lower bound).*

This theorem provides matching upper and lower bounds for the global convergence time of (SGD) starting at $x \in \mathbb{R}^d$, and we prove it in Appendices B–D as a special case of the more general Theorems D.2 and D.3.[9] Importantly, the sandwich bounds for $\tau$ scale exponentially with the inverse of the step-size $\eta$ and show that the global convergence time of (SGD) is characterized by the pruned energy $E(\mathcal{Q}\,|\,x)$. In this sense, $E(\mathcal{Q}\,|\,x)$ characterizes the hardness of the non-convex optimization landscape for (SGD) and captures the fact that the hardness of global convergence stems from the presence of *spurious* local minima (that is, locally minimizing components that are not *globally* minimizing). Indeed, as we show in Appendix D.6, the energy $E(\mathcal{Q}\,|\,x)$ of $\mathcal{Q}$ relative to $x$ vanishes for all $x$ if and only if there are no spurious minima, in which case $\tau$ becomes subexponential.

Moreover, when the initialization $x \in \mathbb{R}^d$ belongs to the basin of a specific minimizing component $p$,[10] Theorem 1 can be restated in a sharper manner that involves directly $E(\mathcal{Q}\,|\,p)$ the energy of $\mathcal{Q}$ relative to $p$, instead of $x$ (cf. (22)).

**Theorem 2.** *Suppose that Assumptions 1–3 hold. Then, given a tolerance $\varepsilon > 0$ and an initialization $x \in \mathbb{R}^d$ that belongs to the basin of attraction of the (minimizing) component $p \in \mathcal{V}$, there exist $\Delta > 0$ and an event $H$ with $\mathbb{P}_x(H) \ge 1 - e^{-\Delta/\eta}$ such that, for small enough $\delta, \eta > 0$, we have*

$$\exp\left(\frac{E(\mathcal{Q}\,|\,p) - \varepsilon}{\eta}\right) \le \mathbb{E}_x[\tau \,|\, H] \le \exp\left(\frac{E(\mathcal{Q}\,|\,p) + \varepsilon}{\eta}\right)$$

*provided that the attracting strength of $\mathcal{Q}$ is large enough for the left inequality (lower bound).*

---

[9]We should note here that the requirement on the attracting strength for the lower bound is purely technical; for more details, see the proof sketch Appendix B.1.

[10]To dispel any ambiguity, we mean here a basin of attraction for the gradient flow of $f$.

This theorem is proved in Appendix D.4 as a special case of the more general Theorems D.5 and D.6, and it shows that, conditioned on a high probability event, the global convergence time of (SGD) starting at $x \in \mathbb{R}^d$ is determined by the energy $E(\mathcal{Q}\,|\,p)$ of $\mathcal{Q}$ relative to the minimizing component $p \in \mathcal{V}$ that attracts $x$ under the gradient flow of $f$.[11] Intuitively, Theorem 2 shows that all initial points $x$ in the basin of $p$ will move to a small neighborhood of $p$ with overwhelmingly high probability and, because this transition takes relatively little time, the global convergence time from to the global minimum is approximately equal to the convergence time from $p$ to $\mathcal{Q}$. In the next section, we quantify the precise way that $E(\mathcal{Q}\,|\,p)$ depends on the geometry of the problem's loss landscape.

## 5 Influence of the loss landscape

In this section, we explain how the energy $E(\mathcal{Q}\,|\,p)$ of (22), which controls the global convergence time, depends on the noise, the depths of the spurious minima and their relative positions with respect to each other. We sketch the key elements and we refer to Appendix E for the full details.

The first step is to derive upper and lower bounds on the transition costs $B_{ij}$. From Assumption 2($c$), we have a lower-bound on $B_{ij}$ for any $i, j$ : we have $B_{ij} \ge \underline{B}_{ij}$ where

$$\underline{B}_{ij} := \inf_{T,\gamma}\left\{\sup_{s<t} \frac{2\left(f(\gamma(t)) - f(\gamma(s))\right)}{\overline{\sigma}^2} : \gamma(0) \in \mathcal{C}_i, \; \gamma(T) \in \mathcal{C}_j\right\}. \tag{24}$$

This means that the transition cost is at least equal to the maximal upward jump in the objective function that cannot be avoided when going from $\mathcal{C}_i$ to $\mathcal{C}_j$. To obtain the upper-bound, we need an extra assumption on the noise.

**Extra noise assumption.** While we only assumed a bound on the magnitude of the gradient noise (Assumption 2($c$)), we now require in addition a bound on the "minimal level of noise", through the following assumption.

**Assumption 4.** The Lagrangian of G is bounded as

$$\mathcal{L}_{\mathsf{G}}(x, v) \le \frac{\|v + \nabla f(x)\|^2}{2\underline{\sigma}^2} \quad \forall v \text{ s.t. } \|v\| \le 2\|\nabla f(x)\|$$

for all $x$ in some large enough compact set.

This assumption is satisfied by general types of noise (including finite-sum models) given a lower-bound on the variance of $\mathsf{Z}(x;\omega)$ and a condition on the support; see Appendix E.1. In particular, if $\mathsf{Z}(x;\omega)$ follows a (truncated) Gaussian distribution with a variance $\sigma^2$, then Assumption 4 is satisfied with $\underline{\sigma}^2 = (1 - \varepsilon)\sigma^2$, where $\varepsilon$ depends

---

[11]Importantly, this situation is generic: under standard assumptions, the set of points that do not belong to such a basin has measure zero.

on the truncation level. In that case, the constant $\overline{\sigma}^2$ of Assumption 2(*c*) can be taken as $\overline{\sigma}^2 = (1 + \varepsilon)\sigma^2$ (see Appendix E.1). Note then that the ratio $\underline{\sigma}^2/\overline{\sigma}^2$ is close to 1, which will matter in the next theorem.

**Transition between basins.** With Assumption 4, we obtain quantitative upper-bounds on the transition costs between two neighboring components. More precisely, if $\mathcal{C}_i$ and $\mathcal{C}_j$ are such that their basins of attraction intersect, then the cost of transitioning from $\mathcal{C}_i$ to $\mathcal{C}_j$ is bounded by

$$B_{ij} \leq \frac{2(f_{i,j} - f_i)}{\underline{\sigma}^2} \tag{25}$$

where $f_{i,j}$ is the minimum of $f$ over that intersection and $f_i$ is the value of $f$ on $\mathcal{C}_i$.

With these bounds, we recover immediately the example of Section 2. When the closure of the basin of $\mathcal{C}_i$ intersects $\mathcal{C}_j$ itself, $f_{i,j}$ is simply the value of $f$ on $\mathcal{C}_j$ and (25) simplifies to the same expression as in (3). The equality in (3) is thus obtained by combining (24) and (25) in the Gaussian case.

**Graph structure.** From the bounds of $B_{ij}$, we can get bounds on $E(\mathcal{Q} \mid p)$ as follows. We restrict the transition graph of Section 4 to neighboring components: we consider $\mathcal{G}'$ with vertices $\mathcal{V}$ and edges $j \rightsquigarrow k$ if the closures of the basins of $\mathcal{C}_j$ and $\mathcal{C}_k$ intersect. Given a path $\mathcal{P}_j = j \rightsquigarrow j_1 \rightsquigarrow \cdots \rightsquigarrow j_m$ in $\mathcal{G}'$ that ends in $\mathcal{Q}$, we define its cost as

$$\text{cost}(\mathcal{P}_j) := \max_{n=0,\dots,m-1} \max\left\{ f_{j_n, j_{n+1}} - f_{j_n}, f_{j_n, j_{n+1}} - f_j \right\}.$$

This cost involves the values of the objective function along the path from $j$ to $\mathcal{Q}$ and captures the maximum depth of a minima encountered on the path: indeed, $f_{j_n, j_{n+1}} - f_{j_n}$ represents the jump in the loss function when going from $\mathcal{C}_{j_n}$ to $\mathcal{C}_{j_{n+1}}$ while $f_{j_n, j_{n+1}} - f_j$ represents the total jump when going from $\mathcal{C}_j$ to $\mathcal{C}_{j_{n+1}}$.

**Result and discussion.** With the above quantities, we can establish that the energy $E(\mathcal{Q} \mid p)$, that governs the global convergence time of Theorem 2, is bounded as follows.

**Theorem 3.** *Under Assumptions 1–4, for $p \in \mathcal{V} \setminus \mathcal{Q}$,*

$$E(\mathcal{Q} \mid p) \leq \max_{j : \underline{B}_{pj} \leq r} \min_{\mathcal{P}_j} \frac{2\,\text{cost}(\mathcal{P}_j)}{\underline{\sigma}^2} + \mathcal{O}\left(1 - \frac{\underline{\sigma}^2}{\overline{\sigma}^2}\right) \tag{26}$$

*for some $r > 0$ that depends on the graph structure $\mathcal{G}'$.*

This means that $E(\mathcal{Q} \mid p)$ is bounded by the maximum depth of the minimizers that SGD must go through in order to reach $\mathcal{Q}$. This involves all the paths in $\mathcal{G}'$ that start at a component close to $p$, as measured by $\underline{B}_{ij}$.

Consider, for instance, the example of Section 2: the bound of Theorem 3 is attained for $p_5 \rightsquigarrow p_3 \rightsquigarrow p_1$ and we recover

the formula of (5). In this case, the convergence time depends exponentially on the depths of the minima. However, different noise models would lead to qualitatively different results: Gaussian noise with variance scaling linearly with the objective value [61] would lead to a polynomial dependence on the depths of the minima. Our framework in Appendix E encompasses this case.

Importantly, in the context of neural network training, our results offer a way to connect geometric properties of the loss landscape to sharp estimates of the convergence time of (SGD). Indeed, under certain conditions, neural networks have no spurious local minima [62, 64], in which case Theorem 1 guarantees *subexponential* convergence to the global minimum of $f$. More to the point, even when spurious minima of bounded depth *are* present [66], Theorems 2 and 3 provide a way to translate these depth bounds directly into convergence time estimates for (SGD).

## 6 Concluding remarks

Our aim was to characterize the global convergence time of (SGD) in non-convex landscapes. Our characterization involves a pair of matching lower and upper bounds which captures the delicate interplay between (*a*) the geometry of the loss landscape of the problem's objective function; (*b*) the statistical profile of the stochastic first-order oracle that provides gradient input to (SGD); and (*c*) the hardest set of obstacles that separate the algorithm's initialization from the function's global minimum. In this sense, the characteristic exponent of our global convergence time estimate can be seen as a measure of the hardness of the non-convex minimization problem at hand – and, importantly, it vanishes (resp. nearly vanishes) if the function admits no spurious local minima (resp. sufficiently shallow local minima), indicating a transition to subexponential convergence times.

A concrete take-away of our analysis and results is that the global convergence time of (SGD) scales exponentially in (*a*) the inverse of the step-size; (*b*) the variance of the noise; and (*c*) the depth of any spurious minima of $f$. Among others, this has explicit ramifications for (overparameterized) neural nets: in order to escape the exponential regime, the depth of spurious minimizers must scale logarithmically in the dimension of the problem which, in turn, indicates the depth (or width, depending on the model) that a neural net must have in order to reach a state with near-zero empirical loss in subexponential time.

From a broader theoretical perspective, our results also provide a flexible analysis template to tackle a wide range of stochastic gradient-based algorithms, such as Adam, oscillatory step-size schedules, etc., as well as the study of interpolation and its influence on the global convergence landscape of (SGD). We defer these investigations to the future.

## Acknowledgments

This research was supported in part by the French National Research Agency (ANR) in the framework of the PEPR IA FOUNDRY project (ANR-23-PEIA-0003), the "Investissements d'avenir" program (ANR-15-IDEX-02), the LabEx PERSYVAL (ANR-11-LABX-0025-01), and project SPICE (G7H-IRG24E90). PM is also a member of the Archimedes Research Unit/Athena Research Center, and was partially supported by project MIS 5154714 of the National Recovery and Resilience Plan Greece 2.0 funded by the European Union under the NextGenerationEU Program. This work was granted access to the HPC resources of IDRIS under the allocation AD011015865 made by GENCI. The authors would like to thank the anonymous reviewers for their constructive feedback and suggestions that helped improve the quality of this work.

## Impact Statement

This paper presents work whose goal is to advance the field of Machine Learning. There are many potential societal consequences of our work, none of which we feel must be specifically highlighted here.

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

# Appendices

# A    Related work

## A.1    (SGD) as a discrete-time Markov chain

In a relatively recent thread in the literature, Dieuleveut et al. [17] and Lu et al. [52] undertook a study of (SGD) as a discrete-time Markov chain. This allowed Dieuleveut et al. [17], Lu et al. [52] to derive conditions under which (SGD) is (geometrically) ergodic and, in this way, to quantify the distance to the minimizer under global growth conditions. Building further on this perspective, Gurbuzbalaban et al. [26], Hodgkinson & Mahoney [27] and Pavasovic et al. [69] showed that, under general conditions, the asymptotic distribution of the iterates of (SGD) is heavy-tailed; As such, these results concern the probability of observing the iterates of (SGD) at very large distances from the origin. This is in contrast with our work, which focuses on the time it takes (SGD) to reach a global minimizer. These two types of results are orthogonal and complementary to each other.

## A.2    The diffusion approximation

A classical thread in the literature – and the foundation of Langevin sampling methods – revolves around gradient flows perturbed by Brownian noise in continuous time; in this case, it is possible to get en explicit expression for the invariant measure of the resulting stochastic differential equation (SDE) [34], or estimate it in more difficult, constrained cases [8, 12, 55, 56, 89, 90]. Owing to our understanding of gradient-based SDEs, an important thread in the literature concerns the so-called *SDE approximation* of (SGD), as pioneered by Li et al. [44, 45]. Applications of this SDE approximation include the study of the dynamics of (SGD) close to manifolds of minimizers [11, 47] or Yang et al. [87] which quantifies the global convergence of the diffusion approximation of (SGD) when the function has no spurious local minima. When the objective function is scale-invariant [46], we can obtain further results: Wang & Wang [82] describes the convergence of the SDE approximation to its asymptotic regime, while Li et al. [47, 48] also quantify the convergence (SGD) initialized close to minimizers.

Another line of work leverages DMFT to study the behavior of the diffusion approximation of (SGD) [59, 60, 77]. The DMFT, or "path-integral" approach, comes from statistical physics and bears a close resemblance to the Freidlin-Wentzell theory of large deviations for SDEs. All these results are either local or concern the asymptotic behaviour of the continuous-time approximation of (SGD). They do not provide information of the global convergence time of the actual discrete-time (SGD), since the approximation guarantees fail to hold on large enough time intervals.

## A.3    Exit times for (SGD)

In our work, we study how long it takes for (SGD) to reach a global minimizer. There is vast literature that instead seeks to understand how long it takes for (SGD) to exit a local minimum. For this, these works study either the diffusion approximation [7, 25, 30, 33, 61, 86] or on heavy-tail versions of this diffusion [67, 81]. Interestingly, some of these works use elements of the continuous-time Freidlin–Wentzell theory [22], which is also the point of departure of our paper.

Note that studying exit times it not enough to characterize the convergence time of (SGD) to a global minimizer. In fact, a simple way to see this is given by our introductory example in Section 2. Indeed, in this case, the escape time of $p_3$ scales exponentially in $f(p_2) - f(p_3)$ (the depth of $p_3$) while the escape time of $p_5$ would scale exponentially in $f(p_4) - f(p_5)$. However, the global convergence time of (SGD) as determined by (4) is greater than either of these two exponentials: indeed, our result takes into account that, when in the vicinity of $p_3$, (SGD) could transition to $p_5$ instead of $p_1$ an arbitrary number of times. Hence, characterizing escape times is not enough to describe the global convergence time of (SGD) in general.

## A.4    Overparameterized neural networks

The pioneering work of [1, 19, 92] show convergence for neural networks: overparameterization and Gaussian initialization enable convergence to a global minimum. Among the many subsequent works, let us mention [4, 84, 88, 91, 93]. In particular, with $N$ denoting the number of training datapoints, this type of results require either a $O(N^2)$-width with Gaussian initialization [65, 68] or a $O(N)$-width under additional structure: infinite depth [54] or specific initialization [49, 65]. Our work addresses a different question as it provides a convergence analysis on general non-convex functions, regardless of the structure of the objective or the initialization.

A fruitful line of works studies the geometric properties of the loss landscapes of neural networks, and in particular whether they contain spurious local minima [36]. Nguyen & Hein [62], Nguyen et al. [64] show that there are no spurious local

minima, or "bad valley", if the architecture possesses an hidden layer with width a least $N$. There are many refinements: e.g. leveraging the structure of the data [78], considering regularization [83], convolutional neural networks [63] or other variations of the architecture [3, 71, 72, 83]. Moreover, Li et al. [42] shows that this requirement is tight for one-hidden layer neural networks, in some settings. When the sizes of the hidden layers are smaller than $N$, the loss landscape does generally have spurious local minima [14]. Interestingly, in this case, the depth of these spurious can be explicitly bounded [66] and made arbitrarily small with hidden layers of size only $\sqrt{N}$.

Another fundamental property of overparameterized neural networks is that they can perfectly fit the training data and therefore, the noise of (SGD) vanishes at the global minimum. In this setting, under sufficient noise assumption everywhere else, Wojtowytsch [85], shows that (SGD) reaches a neighborhood of a global minimizer almost surely, gets trapped in this neighborhood and then converges to the global minimizer with a provided asymptotic rate. Our work can be seen as a precise estimation of the time to reach the neighborhood of the global minimizer.

Also motivated by the interpolation phenomenon, Islamov et al. [32] introduces the so-called $\alpha - \beta$ condition and shows a global convergence of (SGD) under this condition. Though this condition is much weaker than convexity, it only ensures that the function value of the iterates becomes less than the max of the values over all critical points. In contrast, our work characterizes the time to convergence to global minimizer for any non-convex landscape.

### A.5 LDP for stochastic algorithms

Our mathematical development use large deviation results for stochastic processes; see e.g., the monographs of Dupuis & Ellis [20], Freidlin & Wentzell [22]. More precisely, we rely on the large deviation result [5, Cor. C.2], which is an application of the theory of Freidlin & Wentzell [22]. Let us also mention two recent works on large deviations in optimization settings: [6] that studies (SGD) with vanishing step-size on strongly convex functions and Hult et al. [31] that considers general stochastic approximation algorithms.

# B   Large deviation analysis of SGD

## B.1   Outline of proof

Before reviewing notations and assumptions, we provide an overview of the proof of our main results Theorems 1 and 2.

Our framework is built on a large deviation principle for SGD (Appendix B.4): it yields precise estimations of the probability that SGD approximately follows an arbitrary continuous-time path. In particular, this LDP concerns the accelerated (subsampled) SGD process $x_n^\eta$, $n = 0, 1, \dots$, defined as

$$x_n^\eta = x_{n\lfloor 1/\eta \rfloor} \tag{B.1}$$

We rely on the LDP from [5] that we restate in Appendix B.4. Note that this LDP is obtained as a consequence of [22, Chap. 7] by comparing the process $(x_n^\eta)_{n \geq 0}$ to its continuous-time linear interpolation $(X(t))_{t \geq 0}$ defined by

$$X(t) = x_n + \frac{t - n\eta}{\eta}(x_{n+1} - x_n) \tag{B.2}$$

for all $n = 0, 1, \dots$, and all $t \in [n\eta, (n+1)\eta]$. This is why the action function (16) is defined in terms of continuous time.

Our framework also builds on two other key results from [5]. First, we need Lemma B.3 restated in Appendix B.5 that provides a control on the return time of SGD to a neighborhood of the set of critical points. In particular, it shows that the distribution of this return time is roughly sub-exponential. Second, we require the careful analysis done in [5] that exploits the structure of the set of critical points to obtain regularity properties on (16) locally, see Appendix B.6.

In Appendix C, we study the transitions of SGD, or more precisely the sequence of (B.1), between connected components of critical points. In short, we characterize the probability that the iterates of (B.1) transition from one component of critical points to another as well as the time it takes to do so. It is because of this new requirement compared to [5] that we have to refine their estimates. For this, we introduce an induced Markov chain $(z_n)_{n \geq 0}$ (Definition 8) that lives on the set of critical components and captures the essential global convergence properties.

With the transition probabilities and times in hand, the system can now be analyzed as a finite state space Markov chain whose states are the components of the set of critical points. Leveraging the tools of Freidlin & Wentzell [22], restated in Appendix D.1, we obtain a characterization of the global convergence time of (B.1) to the set of critical points in Appendix D.2.

The only thing left to do is to translate the results obtained for the subsampled process of (B.1) back to the original process of SGD (Appendix D.3). The upper-bound is immediate but the lower-bound needs some care. This is where the attracting strength defined in Section 3 comes into play: if the attracting strength is large enough, we can show that when SGD reaches a neighborhood of a global minimum, it will remain there enough time for the next iterate of the subsampled process of (B.1) to still be in that neighborhood. This in turn allows us to show the global convergence times of SGD and the subsampled sequence roughly coincide, concluding the proof of Theorem 1.

Appendix D.4 is then dedicated to obtaining Theorem 2 while Appendix D.6 establishes the link between the presence of spurious local minima and a non-zero energy.

The results of Section 5 are then obtained in Appendix E.

For convenience, the main dependencies between sections of the appendix are summarized in Fig. 3.

## B.2   Setup and assumptions

Before we begin our proof, we introduce here notations and we revisit and discuss our standing assumptions. In particular, to extend the range of our results, we provide in the rest of this appendix a weaker version of the blanket assumptions of Section 3 which will be in force throughout the appendix.

We equip $\mathbb{R}^d$ with the canonical inner product $\langle \cdot, \cdot \rangle$ and the associated Euclidean norm $\|\cdot\|$. We denote by $\mathbb{B}(x, r)$ (resp. $\overline{\mathbb{B}}(x, r)$) the open (resp. closed) ball of radius $r$ centered at $x$.

We also also define, for any $A \subset \mathbb{R}^d$,

$$\mathcal{U}_\delta(A) \coloneqq \{x \in \mathbb{R}^d : d(x, A) < \delta\}. \tag{B.3}$$

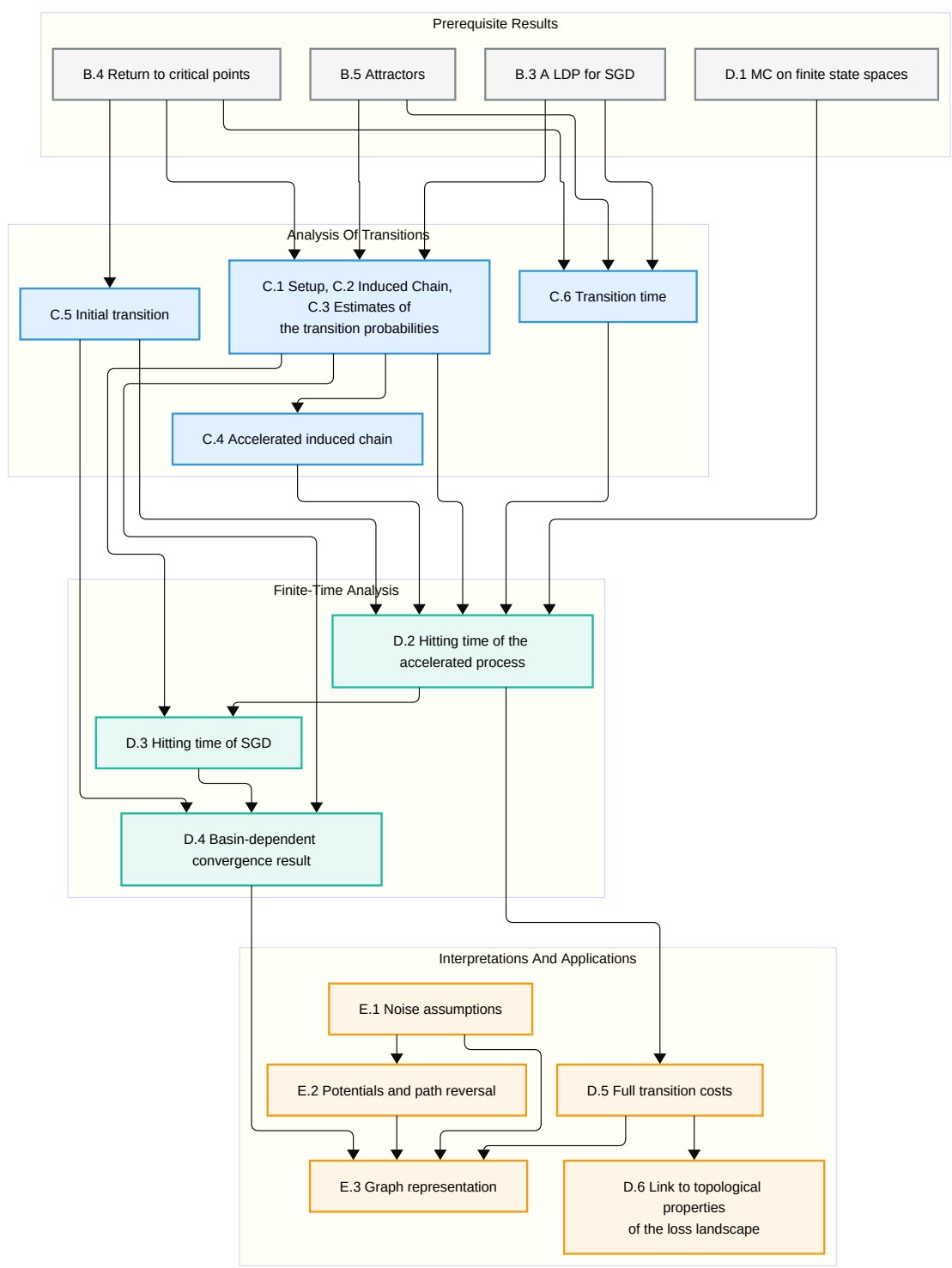

**Figure 3:** Structure of the proof: arrows represent the main logical implications between the main parts of the proof. The group "Prerequisite results" gathers the tools from the literature that we rely on, mostly either from [4, 20, 21]. Our key technical contributions that lead to Thm. 1-2 are "Analysis Of Transitions" and "Finite Time Analysis". The results of §5 come from "Interpretation and Applications".

For any $T > 0$, we denote by $\mathcal{C}([0,T]) = \mathcal{C}([0,T], \mathbb{R}^d)$ the set of continuous functions from $[0,T]$ to $\mathbb{R}^d$.

We begin with our assumptions for the objective function $f$, which are a weaker version of Assumption 1.

**Assumption 5.** The objective function $f \colon \mathbb{R}^d \to \mathbb{R}$ satisfies the following conditions:

(a) *Coercivity:* $f(x) \to \infty$ as $\|x\| \to \infty$.

(b) *Smoothness:* $f$ is $C^2$-differentiable and its gradient is $\beta$-Lipschitz continuous, namely

$$\|\nabla f(x') - \nabla f(x)\| \le \beta \|x' - x\| \qquad \text{for all } x, x' \in \mathbb{R}^d. \tag{B.4}$$

**Assumption 6** (Critical set regularity)**.** The critical set

$$\mathrm{crit}(f) \coloneqq \left\{ x \in \mathbb{R}^d : \nabla f(x) = 0 \right\} \tag{B.5}$$

of $f$ consists of a finite number of (compact) connected components. Moreover, each of these components $\mathcal{K}$ is connected by piecewise absolutely continuous paths, i.e., for any $x, x' \in \mathcal{K}$, there exists $\gamma \in \mathcal{C}([0,1], \mathcal{K})$ such that $\gamma_0 = x$, $\gamma_1 = x'$ and such that it is piecewise absolutely continuous, i.e., $\gamma$ is differentiable almost everywhere and there exists $0 = t_0 < t_1 < \cdots < t_N = 1$ such that $\dot{\gamma}$ is integrable on every closed interval of $(t_n, t_{n+1})$ for $n = 0, \ldots, N - 1$

Note that, since connected components of a closed set are closed, Assumption 5 automatically ensure that the connected components of $\mathrm{crit}(f)$ are compact.

Unlike in Assumption 1, we do not require the connected components of $\mathrm{crit} f$ to be smoothly connected but only piecewise absolutely continuous.

*Remark* B.1. The path-connectedness requirement of Assumption 6 is satisfied whenever the connected components of $\mathrm{crit}(f)$ are isolated critical points, smooth manifolds, or finite unions of closed manifolds. More generally, Assumption 6 is satisfied whenever $f$ is definable – in which case $\mathrm{crit} f$ is also definable, so each component can be connected by piecewise smooth paths [15, 76]. The relaxation provided by Assumption 6 represents the "minimal" set of hypotheses that are required for our analysis to go through. §

Moving forward, to align our notation with standard conventions in large deviations theory, it will be more convenient to work with $-\mathsf{Z}(x; \omega)$ instead of $\mathsf{Z}(x; \omega)$ in our proofs. To make this clear, we restate below Assumption 2 in terms of the noise process

$$u(x, \omega) = -\mathsf{Z}(x; \omega). \tag{B.6}$$

We also take the chance to relax the definition of the variance proxy of $u$, which requires the new assumption Assumption 8.

**Assumption 7.** The error term $u \colon \mathbb{R}^d \times \Omega \to \mathbb{R}^d$ satisfies the following properties:

(a) *Properness:* $\mathbb{E}[u(x, \omega)] = 0$ and $\mathrm{cov}(u(x, \omega)) > 0$ for all $x \in \mathbb{R}^d$.

(b) *Smooth growth:* $u(x, \omega)$ is $C^2$-differentiable and satisfies the growth condition

$$\sup_{x, \omega} \frac{\|u(x, \omega)\|}{1 + \|x\|} < +\infty. \tag{B.7}$$

(c) *Sub-Gaussian tails:* There is $\sigma_\infty^2 \colon \mathbb{R} \to (0, +\infty)$ continuous, with $\inf_{\mathbb{R}^d} \sigma_\infty^2 > 0$, such that $u(x, \omega)$ satisfies

$$\log \mathbb{E}\left[ e^{\langle p, u(x, \omega) \rangle} \right] \le \frac{\sigma_\infty^2(f(x))}{2} \|p\|^2 \qquad \text{for all } p \in \mathbb{R}^d. \tag{B.8}$$

**Assumption 8.** The signal-to-noise ratio of $\mathsf{G}$ is bounded as

$$\liminf_{\|x\| \to \infty} \frac{\|\nabla f(x)\|^2}{\sigma_\infty^2 \circ f(x)} > 16 \log 6 \cdot d. \tag{B.9}$$

Furthermore, $\sigma_\infty^2(t) = o(t^2)$ as $t \to +\infty$ and, $\frac{\sigma_\infty^2 \circ f(x)}{\|x\|^s}$ is bounded above and below at infinity for $s \in [0, 2]$, i.e.,

$$0 < \liminf_{\|x\| \to +\infty} \frac{\sigma_\infty^2 \circ f(x)}{\|x\|^s} \quad \text{and} \quad \limsup_{\|x\| \to +\infty} \frac{\sigma_\infty^2 \circ f(x)}{\|x\|^s} < +\infty. \tag{B.10}$$

*Remark* B.2. The key distinction between Assumption 2 and Assumptions 7–8 stems from their differing requirements for the variance proxy $\sigma_\infty^2$ of the noise in (SGD). Since $f$ is coercive, allowing $\sigma_\infty^2$ to depend on $f(x)$ enables us to consider noise processes whose variance may grow unbounded as $\|x\| \to \infty$; we specifically choose to express this dependence through $f(x)$ rather than directly through $x$ as it substantially simplifies both our proofs and calculations. A version of Assumption 4 adapted to this generalized setting will be stated as Assumption 11 in Appendix E.1. §

We now introduce the notation for the target set $\mathcal{Q}$. In the main text, we choose $\mathcal{Q}$ to be the set of global minima of $f$ but this needs not be the case in general. In the rest of the appendix, $\mathcal{Q}$ will a union of connected components of crit($f$).

**Definition 1** (Choice of the components of interest). Denote by $\mathcal{Q}_1, \ldots, \mathcal{Q}_{N_{\mathrm{targ}}}$ the connected components of crit($f$) that form the *target* set.

Denote by $\mathcal{K}_1, \ldots, \mathcal{K}_K$ the remaining connected components of crit($f$) and denote by

$$(\mathcal{C}_1, \ldots, \mathcal{C}_{N_{\mathrm{crit}}}) \coloneqq (\mathcal{K}_1, \ldots, \mathcal{K}_K, \mathcal{Q}_1, \ldots, \mathcal{Q}_{N_{\mathrm{targ}}}) \tag{B.11}$$

their union with $N_{\mathrm{crit}} = K + N_{\mathrm{targ}}$.

Since the connected components of crit($f$) are pairwise disjoint by definition, their compactness implies that there exists $\delta > 0$ such that $\mathcal{U}_\delta(\mathcal{C}_i)$, for $i = 1, \ldots, N_{\mathrm{crit}}$, are pairwise disjoint.

In this framework, the iterates of (SGD), started at $x \in \mathbb{R}^d$, are defined by the following recursion:

$$\begin{cases} x_0 \in \mathbb{R}^d \\ x_{n+1} = x_n - \eta \nabla f(x_n) + \eta u_n, \quad \text{where } u_n = u(x_n, \omega_n) \end{cases} \tag{B.12}$$

where $(\omega_n)_{n \geq 0}$ is a sequence of random variables in $\mathbb{R}^m$. We will denote by $\mathbb{P}_x$ the law of the sequence $(\omega_n)_{n \geq 0}$ when the initial point is $x$ and by $\mathbb{E}_x$ the expectation with respect to $\mathbb{P}_x$.

Assumptions 5 and 7 imply the following growth condition, that we assume holds with the same constant for the sake of simplicity. There is $M > 0$ such that, for all $x \in \mathbb{R}^d$, $\omega \in \Omega$,

$$\|\nabla f(x)\| \leq M(1 + \|x\|) \quad \text{and} \quad \|u(x, \omega)\| \leq M(1 + \|x\|). \tag{B.13}$$

## B.3  Hamiltonian and Lagrangian

Following the notation of [5], we introduce the cumulant generating functions of the noise $u(x, \omega)$ and of the drift $-\nabla f(x) + u(x, \omega)$, that we denote by $\bar{\mathcal{H}}, \mathcal{H}$ to avoid confusion. We also define their convex conjugates, $\bar{\mathcal{L}}, \mathcal{L}$.

**Definition 2** (Hamiltonian and Lagrangian). Define, for $x \in \mathbb{R}^d$, $v \in \mathbb{R}^d$,

$$\bar{\mathcal{H}}(x, v) = \log \mathbb{E}[\exp(\langle v, u(x, \omega) \rangle)] \tag{B.14a}$$

$$\mathcal{H}(x, v) = -\langle \nabla f(x), v \rangle + \bar{\mathcal{H}}(x, v) \tag{B.14b}$$

$$\bar{\mathcal{L}}(x, v) = \bar{\mathcal{H}}(x, \cdot)^*(v) \tag{B.14c}$$

$$\mathcal{L}(x, v) = \mathcal{H}(x, \cdot)^*(v) = \bar{\mathcal{L}}(x, v + \nabla f(x)). \tag{B.14d}$$

$\mathcal{L}$ is thus equal to the Lagrangian $\mathcal{L}_{\mathsf{G}}(\cdot, \cdot)$.

We restate [5, Lem. B.1] that provides basic properties of the Hamiltonian and Lagrangian functions.

**Lemma B.1** (Properties of $\mathcal{H}$ and $\mathcal{L}$, [5, Lem. B.1]).

*1. $\mathcal{H}$ is $C^2$ and $\mathcal{H}(x, \cdot)$ is convex for any $x \in \mathbb{R}^d$.*

*2. $\mathcal{L}(x, \cdot)$ is convex for any $x \in \mathbb{R}^d$, $\mathcal{L}$ is lower semi-continuous (l.s.c.) on $\mathbb{R}^d \times \mathbb{R}^d$.*

*3. For any $x \in \mathbb{R}^d$, $v \in \mathbb{R}^d$, $\mathcal{H}(x, v) \leq 2M(1 + \|x\|)\|v\|^2$ and dom $\mathcal{L}(x, \cdot) \subset \overline{\mathbb{B}}(0, 2M(1 + \|x\|))$.*

*4. For any $x \in \mathbb{R}^d$, $v \in \mathbb{R}^d$, $\mathcal{L}(x, v) \geq 0$ and $\mathcal{L}(x, v) = 0 \iff v = \nabla f(x)$.*

The following lemma provides a lower bound on the Lagrangian and is an immediate consequence of the sub-Gaussian tails assumption ((c)).

**Lemma B.2** ([5, Lem. D.5]). *For any $x \in \mathbb{R}^d$, $v \in \mathbb{R}^d$,*

$$\mathcal{L}(x, v) \geq \frac{\|v + \nabla f(x)\|^2}{2\sigma_\infty^2 \circ f(x)} . \tag{B.15}$$

## B.4 A large deviation principle for SGD

In this section, we present and restate the large deviation principles established in [5]for SGD. Note that their proof is itself an application of the general theory of Freidlin & Wentzell [22]. From the sequences $(x_n)_{n \geq 0}$ and $(\omega_n)_{n \geq 0}$, we define another discrete sequence: a subsampled or, accelerated, sequence

$$x_n^\eta := x_{n\lfloor 1/\eta \rfloor} . \tag{B.16}$$

From the Lagrangian defined in (B.14d), we define, on $\mathcal{C}([0, T], \mathbb{R}^d)$, the normalized action functional $\mathcal{S}_{0,T}$ by

$$\mathcal{S}_{0,T}(\gamma) = \begin{cases} \int_0^T \mathcal{L}(\gamma_t, \dot{\gamma}_t) \, dt & \text{if } \gamma \text{ absolutely continuous} \\ +\infty & \text{otherwise} \end{cases} \tag{B.17}$$

following Freidlin & Wentzell [22, Chap. 3.2], as a manner to quantify how "probable" a trajectory is.

For some $N > 0$, we will first equip $(\mathbb{R}^d)^N$ with the distance

$$\text{dist}_N(\xi, \zeta) = \max_{0 \leq n \leq N-1} \|\xi_n - \zeta_n\| . \tag{B.18}$$

Now, for $N \geq 0$, $\xi = (\xi_0, \ldots, \xi_{N-1}) \in \mathbb{R}^{dN}$, let us define the normalized discrete action functional

$$\mathcal{A}_N(\xi) := \sum_{n=0}^{N-2} \rho(\xi_n, \xi_{n+1}) \tag{B.19}$$

where the cost of moving from one iteration to the next is defined for any $x, x' \in \mathbb{R}^d$ from the previous continuous normalized action functional (cf. (B.17)) with horizon 1 as

$$\rho(x, x') := \inf\{\mathcal{S}_{0,1}(\gamma) : \gamma \in \mathcal{C}([0, 1], \mathbb{R}^d), \gamma_0 = x, \gamma_1 = x'\} . \tag{B.20}$$

We now present the large deviation principle on the discrete accelerated sequence $(x_n^\eta)_{0 \leq n \leq N-1} = (x_{n\lfloor \eta^{-1} \rfloor})_{0 \leq n}$. In the following result, the functional $\eta^{-1}\mathcal{A}_N$ is thus the action functional in $(\mathbb{R}^d)^N$ of the process $(x_n^\eta)_{0 \leq n \leq N-1}$ uniformly with respect to the starting point $x_0$ in any compact set $\mathcal{K} \subset \mathbb{R}^d$, as $\eta \to 0$.

**Proposition B.1.** *Fix $N \geq 0$.*

- *For any $s > 0$, the set*

$$\Gamma_N^{\mathcal{K}}(s) := \{\xi \in \mathbb{R}^{dN} : \xi_0 \in \mathcal{K}, \mathcal{A}_N(\xi) \leq s\} \tag{B.21}$$

 *is compact and $\mathcal{A}_N$ is l.s.c. on $\mathbb{R}^{dN}$.*

- *For any $s, \delta, \varepsilon > 0$, $\mathcal{K} \subset \mathbb{R}^d$ compact, there exists $\eta_0 > 0$ such that, for any $\eta \in (0, \eta_0]$, for any $x_0 \in \mathcal{K}$, $n \leq N$, $\xi \in \Gamma_n^{\{x_0\}}(s)$, we have that*

$$\mathbb{P}_{x_0}\left(\text{dist}_n(x^\eta, \xi) < \delta\right) \geq \exp\left(-\frac{\mathcal{A}_n(\xi) + \varepsilon}{\eta}\right) \tag{B.22a}$$

$$\mathbb{P}_{x_0}\left(\text{dist}_n(x^\eta, \Gamma_n^{\{x_0\}}(s)) > \delta\right) \leq \exp\left(-\frac{s - \varepsilon}{\eta}\right) . \tag{B.22b}$$

## B.5 Return to critical points

In this section, we restate a key result from [5]: the lemma below provides a control on the return time to a neighborhood of the set of critical points. In particular, it shows that the distribution of this return time is roughly sub-exponential.

**Definition 3** (Stopping times for the accelerated process). For any set $A \subset \mathbb{R}^d$, we define the hitting and exit times of $A$:

$$\tau_A^\eta := \inf\{n \geq 0 : x_n^\eta \in A\}, \tag{B.23a}$$

$$\sigma_A^\eta := \inf\{n \geq 0 : x_n^\eta \notin A\}. \tag{B.23b}$$

**Lemma B.3** ([5, Lem. D.21]). *Consider* $\mathrm{crit}(f) \subset \mathcal{U} \subset \mathcal{X} \subset \mathbb{R}^d$ *with* $\mathcal{U}$ *an open set and* $\mathcal{X}$ *a compact set. Then, there is some* $\eta_0, \alpha_0, a, b > 0$ *such that,*

$$\forall \eta \leq \eta_0,\ \alpha \leq \alpha_0,\ x \in \mathcal{X}, \quad \mathbb{E}_x\left[e^{\frac{\alpha \tau_\mathcal{U}^\eta}{\eta}}\right] \leq e^{\frac{a\alpha}{\eta}+b}. \tag{B.24}$$

## B.6 Attractors

We again build on the work of [5] which took inspiration from the framework of Kifer [39].We first need to define the gradient flow of $f$.

**Definition 4.** Define, for $x \in \mathbb{R}^d$, the flow $\Theta$ of $-\nabla f$ started at $x$, i.e.,

$$\Theta_0(x) = x \tag{B.25}$$

$$\dot{\Theta}_t(x) = -\nabla f(\Theta_t(x)). \tag{B.26}$$

and let $F(x)$ be the value of this flow at time 1, i.e.,

$$F(x) = \Theta_1(x). \tag{B.27}$$

We first list some basic properties.

**Lemma B.4** (Properties of the flow [5, Lem. D.1]). $\Theta$ *is well-defined and continuous in both time and space, and, for any* $T \geq 0,\ \gamma \in \mathcal{C}([0,T],\mathbb{R}^d)$ *such that* $\gamma_0 = x$,

$$\mathcal{S}_{0,T}(\gamma) = 0 \iff \gamma_t = \Theta_t(x) \quad \text{for all } t \in [0,T]. \tag{B.28}$$

The following lemma translates this for $F$.

**Lemma B.5** (Properties of $F$,[5, Lem. D.2]). $F$ *is well-defined and continous and, for any* $x, x' \in \mathbb{R}^d$,

$$\rho(x, x') = 0 \iff x' = F(x). \tag{B.29}$$

The next two lemmas are key regularity results on the connected components of the critical set.

**Lemma B.6** ([5, Lem. D.8]). *For any* $\mathcal{C} \subset \mathrm{crit}(f)$ *connected component of the critical set, there is* $r_0 > 0$ *such that, for any* $0 < r \leq r_0$,

$$\mathcal{W}_r(\mathcal{C}) := \{x \in \mathbb{R}^d : \rho(x, \mathcal{C}) < r, \rho(\mathcal{C}, x) < r\} \tag{B.30}$$

*is open and contains* $\mathcal{C}$.

**Lemma B.7** ([5, Lem. D.9]). *Let* $\mathcal{C} \subset \mathrm{crit}(f)$ *be a connected component of the critical set. Then, for any* $\varepsilon > 0$, *there is some* $N \geq 1$ *such that, for any* $x, z \in \mathcal{C}$, *there is* $\xi \in \mathcal{D}_r(N)$ *such that* $\xi_0 = x$, $\xi_{N-1} = z$, $\mathcal{A}_N(\xi) < \varepsilon$ *and* $\max_{0 \leq n < N} d(\xi_n, \mathcal{C}) < \varepsilon$.

## B.7 Convergence and stability

**Lemma B.8** ([5, Lem. D.28]). *For any* $x \in \mathcal{X}_r$, *there exists* $i \in \{1, \ldots, N_{crit}\}$ *such that*

$$\lim_{t \to +\infty} d(\Theta_t(x), \mathcal{C}_i) = 0. \tag{B.31}$$

**Definition 5.** A connected component of the critical points $C \subset \operatorname{crit}(f)$ is said to be *asymptotically stable* if there exists $\mathcal{U}$ a neighborhood of $C$ such that, for any $x \in \mathcal{U}$, $\Theta_t(x)$ converges to $C$, i.e.,

$$\lim_{t \to +\infty} d(\Theta_t(x), C) = 0. \tag{B.32}$$

The notions of minimizing component and asymptotic stability are equivalent in our context.

**Definition 6.** $C$ connected component of $\operatorname{crit}(f)$ is *minimizing* if there exists $\mathcal{U}$ a neighborhood of $C$ such that

$$\operatorname*{arg\,min}_{x \in \mathcal{U}} f(x) = C \tag{B.33}$$

Note that since $\operatorname{crit}(f)$ is closed, $C$ is closed as well as a connected component of a closed set.

**Lemma B.9** ([5, Lem. D.29]). *For any $C \subset \operatorname{crit}(f)$ connected component of the set of critical points, $C$ is minimizing if and only if it is asymptotically stable.*

# C Transitions

In this section, we study the transitions of the accelerated sequence of SGD iterates between the different sets of critical points. As in [5], we build on the work of Kifer [39] and Freidlin & Wentzell [22]. More precisely, Appendices C.1–C.5 consists in refining the framework of [5] to be able to obtain precise time estimates on the transitions between the different sets of critical points. Such results are provided in Appendix C.6.

## C.1 Setup

We adapt to our context Kifer [39, Lem. 5.4] and simplify it using ideas from Freidlin & Wentzell [22, Chap. 6].

**Definition 7** (Freidlin & Wentzell [22, Chap. 6,§2])**.** For $i \in \{1, \ldots, K\}$, $j \in \{1, \ldots, N_{\text{crit}}\}$, $\delta > 0$,

$$\mathcal{B}_{i,j} := \inf \left\{ \mathcal{A}_N(\xi) : N \geq 1, \xi \in \mathcal{D}_r(N), \xi_0 \in \mathcal{C}_i, \xi_{N-1} \in \mathcal{C}_j, \xi_n \notin \bigcup_{l \neq i,j} \mathcal{C}_l \text{ for all } n = 1, \ldots, N-2 \right\}$$

$$\mathcal{B}_{i,j}^{\delta} := \inf \left\{ \mathcal{A}_N(\xi) : N \geq 1, \xi \in \mathcal{D}_r(N), \xi_0 \in \mathcal{U}_{\delta}(\mathcal{C}_i), \xi_{N-1} \in \mathcal{U}_{\delta}(\mathcal{C}_j), \xi_n \notin \bigcup_{l \neq i,j} \mathcal{C}_l \text{ for all } n = 1, \ldots, N-2 \right\} \quad \text{(C.1)}$$

While $\mathcal{B}_{i,j}$ is the usual definition from Freidlin & Wentzell [22, Chap. 6,§2], $\mathcal{B}_{i,j}^{\delta}$ is a variant that will prove helpful.

Let us now first list a few immediate properties.

**Lemma C.1.** *For $i \in \{1, \ldots, K\}$, $j \in \{1, \ldots, N_{\text{crit}}\}$, $\mathcal{B}_{i,j}^{\delta}$ is non-decreasing in $\delta$ and, for any $\delta > 0$, $\mathcal{B}_{i,j}^{\delta} \leq \mathcal{B}_{i,j}$.*

Note that the fact that $\mathcal{B}_{i,j}^{\delta}$ is non-decreasing in $\delta$ implies that the limit $\lim_{\delta \to 0} \mathcal{B}_{i,j}^{\delta}$ exists.

We now exploit the regularity around the $\mathcal{K}_i$, to obtain an alternate expression for $\mathcal{B}_{i,j}$ as the limit of $\mathcal{B}_{i,j}^{\delta}$ as $\delta \to 0$.

**Lemma C.2.** *For $i \in \{1, \ldots, K\}$, $j \in \{1, \ldots, N_{\text{crit}}\}$, the following equality holds:*

$$\mathcal{B}_{i,j} = \lim_{\delta \to 0} \mathcal{B}_{i,j}^{\delta} . \quad \text{(C.2)}$$

*Proof.* By Lemma C.1, we have

$$\mathcal{B}_{i,j}^{\delta} \leq \mathcal{B}_{i,j}^{\delta} , \quad \text{(C.3)}$$

and therefore

$$\lim_{\delta \to 0} \mathcal{B}_{i,j}^{\delta} \leq \mathcal{B}_{i,j} . \quad \text{(C.4)}$$

It remains to show the reverse inequality.

Take $\varepsilon > 0$. Apply Lemma B.6 to $\mathcal{C}_i$: at the potential cost of reducing $\varepsilon$, we get that $\mathcal{W}_{\varepsilon}(\mathcal{C}_i)$ is an open neighborhood of $\mathcal{C}_i$.

Take $\delta > 0$ small enough so that $\mathcal{U}_{\delta}(\mathcal{C}_i) \subset \mathcal{W}_{\varepsilon}(\mathcal{C}_i)$, $\mathcal{U}_{\delta}(\mathcal{C}_i)$ does not intersect with any other $\mathcal{C}_l$, $l \neq j$.

Now, take any $N \geq 1$ and $\xi \in \mathcal{D}_r(N)$ such that $\xi_0 \in \mathcal{U}_{\delta}(\mathcal{C}_i)$, $\xi_{N-1} \in \mathcal{U}_{\delta}(\mathcal{C}_j)$, $\xi_n \notin \bigcup_{l \neq i,j} \mathcal{C}_l$ for all $n = 1, \ldots, N-2$.

By the choice of $\delta$, $\xi_0$ cannot be in $\bigcup_{l \neq i,j} \mathcal{C}_l$ and $\xi_0$ is inside $\mathcal{W}_{\varepsilon}(\mathcal{C}_i)$ so that there exists $x \in \mathcal{C}_i$ such that $\rho(x, \xi_0) < \varepsilon$. Similarly, there exists $x' \in \mathcal{C}_j$ such that $\rho(\xi_{N-1}, x') < \varepsilon$. Now consider the path $\zeta \in \mathcal{D}_r(N+2)$ defined as $\zeta = (x, \xi_0, \xi_1, \ldots, \xi_{N-1}, x')$ that satisfies $\zeta_0 \in \mathcal{C}_i$, $\zeta_N \in \mathcal{C}_j$ and $\zeta_n \notin \bigcup_{l \neq i,j} \mathcal{C}_l$ for all $n = 1, \ldots, N-2$. Furthermore, $\mathcal{A}_{N+2}(\zeta) \leq \mathcal{A}_N(\xi) + 2\varepsilon$ and thus $\mathcal{B}_{i,j} \leq \mathcal{A}_N(\xi) + 2\varepsilon$. Passing to the infimum over such paths $\xi$ yields

$$\mathcal{B}_{i,j}^{\delta} \leq \mathcal{B}_{i,j}^{\delta} + 2\varepsilon , \quad \text{(C.5)}$$

which concludes the proof. ∎

## C.2 Induced chain

We now define an important object: the law of the (accelerated) iterated at the first time they reach some set $\mathcal{V}$ (typically a neighborhood of the critical set). Due to our interest in the finite-time dynamics, we slightly deviate from the classical definitions of Douc et al. [18, Chap. 3.4] and Kifer [39, Prop. 5.3] to follow more closely the one of Freidlin & Wentzell [22, Chap. 6,§2].

**Definition 8.** Consider $\mathcal{V}_i$, $i = 1, \ldots, N_{\text{crit}}$ disjoint neighborhoods of $\mathcal{C}_i$ and denote by $\mathcal{V} := \cup_{i=1}^{N_{\text{crit}}} \mathcal{V}_i$ their union. We define recursively the sequences of stopping times $(\sigma_n^\eta)_{n \geq 0}$ and $(\tau_n^\eta)_{n \geq 0}$ by

- For $n = 0$,

$$\sigma_0^\eta := 0 \tag{C.6}$$
$$\tau_0^\eta := 0. \tag{C.7}$$

- For $n \geq 0$, if $x_{\tau_n^\eta}^\eta$ is in $\mathcal{V}_{i_n}$, then

$$\sigma_{n+1}^\eta := \begin{cases} \inf\{k \geq \tau_n^\eta : x_k^\eta \notin \mathcal{V}_{i_n}\} & \text{if } x_{\tau_n^\eta}^\eta \in \mathcal{V}_{i_n} \text{ for some } i_n \text{ s.t. } \mathcal{C}_{i_n} \in \{\mathcal{K}_1, \ldots, \mathcal{K}_K\}, \\ \tau_n^\eta & \text{otherwise}. \end{cases} \tag{C.8}$$

$$\tau_{n+1}^\eta := \inf\{k \geq \sigma_{n+1}^\eta : x_k^\eta \in \mathcal{V}\}. \tag{C.9}$$

We denote by $z_n = x_{\tau_n^\eta}^\eta$, $n = 0, 1, \ldots$, the induced Markov chain and by $\mathbb{Q}_\mathcal{V}(x, \cdot)$ the corresponding Markov transition probability i.e., the law of $z_1$ started at $z_0 = x$.

A few remarks are in order on $(z_n)_{n \geq 0}$:

- $z_n \in \mathcal{V}$ for all $n \geq 1$.
- If the chain reaches a neighborhood of $\mathcal{Q}_1, \ldots, \mathcal{Q}_{N_{\text{targ}}}$ it stays there forever.

## C.3 Estimates of the transition probabilities

**Lemma C.3.** *For any $\varepsilon > 0$, $A > 0$, for any small enough neighborhoods $\mathcal{V}_i$ of $\mathcal{C}_i$, $i = 1, \ldots, N_{\text{crit}}$, there is some $\eta_0 > 0$ such that for all $i \in \{1, \ldots, K\}$, $j \in \{1, \ldots, N_{\text{crit}}\}$, $x \in \mathcal{V}_i$, $0 < \eta < \eta_0$,*

$$\mathbb{Q}_\mathcal{V}(x, \mathcal{V}_j) \leq \begin{cases} \exp\left(-\frac{\mathcal{B}_{i,j}}{\eta} + \frac{\varepsilon}{\eta}\right) & \text{if } \mathcal{B}_{i,j} < +\infty, \\ \exp\left(-\frac{A}{\eta}\right) & \text{otherwise}. \end{cases} \tag{C.10}$$

*Proof.* Following the alternative definition of the $\mathcal{B}_{i,j}$'s Lemma C.2, choose $\delta > 0$ small enough so that both $\mathcal{U}_{\frac{\delta}{2}}(\mathcal{C}_i)$, $i = 1, \ldots, N_{\text{crit}}$ are pairwise disjoints and the following holds for all $i, j \in \{1, \ldots, N_{\text{crit}}\}$:

$$\mathcal{B}_{i,j}^\delta \geq \begin{cases} \mathcal{B}_{i,j} - \varepsilon & \text{if } \mathcal{B}_{i,j} < +\infty, \\ A & \text{otherwise}. \end{cases} \tag{C.11}$$

Require then that $\mathcal{V}_i$ be contained in $\mathcal{U}_{\frac{\delta}{2}}(\mathcal{C}_i)$ for all $i = 1, \ldots, N_{\text{crit}}$. Note that the $\mathcal{V}_i$'s are pairwise disjoint by construction.

Given neighborhoods $\mathcal{V}_i$, $i = 1, \ldots, N_{\text{crit}}$, by compactness of $\text{crit}(f)$, there exists $\delta' > 0$ such that $\delta' \leq \frac{\delta}{2}$, $\mathcal{U}_{2\delta'}(\mathcal{C}_i)$ is contained in $\mathcal{V}_i$ for all $i = 1, \ldots, N_{\text{crit}}$.

Fix $i, j \in \{1, \ldots, N_{\text{crit}}\}$ and consider $\xi \in \mathcal{D}_r(N)$ such that $\xi_0 \in \mathcal{U}_{\delta'}(\mathcal{V}_i) \subset \mathcal{U}_\delta(\mathcal{C}_i)$, $\xi_{N-1} \in \mathcal{U}_{\delta'}(\mathcal{V}_j) \subset \mathcal{U}_\delta(\mathcal{C}_j)$ and $\xi_n \in \mathcal{U}_{\delta'}(\mathbb{R}^d \setminus \cup_{l \neq i,j} \mathcal{V}_l)$ for all $n = 1, \ldots, N-2$. By the choice of $\delta'$, $\xi_n$ cannot be in $\cup_{l \neq i,j} \mathcal{U}_{\delta'}(\mathcal{C}_l)$ for any $n = 1, \ldots, N-2$. By construction of $\xi$ we thus have that

$$\mathcal{A}_N(\xi) \geq \mathcal{B}_{i,j}^\delta$$
$$\geq \begin{cases} \mathcal{B}_{i,j} - \varepsilon & \text{if } \mathcal{B}_{i,j} < +\infty, \\ A & \text{otherwise}. \end{cases} \tag{C.12}$$

where the last inequality follows from (C.11).

Fix $x \in \mathcal{V}_i$. Let us now bound the probability

$$\mathbb{Q}_\mathcal{V}(x, \mathcal{V}_j) = \mathbb{P}_x\left(x_{\tau_\mathcal{V}^\eta}^\eta \in \mathcal{V}_i\right), \tag{C.13}$$

and start with the case where $\mathcal{B}_{i,j} < +\infty$.

We have, for any $N \geq 0$,

$$\mathbb{P}_x\left(x^\eta_{\tau^\eta_\mathcal{V}} \in \mathcal{V}_i\right) \leq \mathbb{P}_x\left(x^\eta_{\tau^\eta_\mathcal{V}} \in \mathcal{V}_i, \tau^\eta_\mathcal{V} < N\right) + \mathbb{P}_x\left(\tau^\eta_\mathcal{V} \geq N\right). \tag{C.14}$$

We first bound the second probability using Lemma B.3 applied to $\mathcal{U} \leftarrow \mathcal{V}_i$. Take $N$ such that $\alpha_0(a-N) + \eta_0 b \leq -\mathcal{B}_{i,j}$. Then, by Markov's inequality and Lemma B.3, it holds that for all $\eta \leq \eta_0$

$$\begin{aligned}
\mathbb{P}_x\left(\tau^\eta_\mathcal{V} \geq N\right) &\leq \mathbb{P}_x\left(\exp\left(\frac{\alpha_0 \tau^\eta_\mathcal{V}}{\eta}\right) \geq \exp\left(\frac{\alpha_0 N}{\eta}\right)\right) \\
&\leq \exp\left(\frac{\alpha_0(a-N)}{\eta} + b\right) \\
&\leq \exp\left(\frac{-\mathcal{B}_{i,j}}{\eta}\right).
\end{aligned} \tag{C.15}$$

We now bound the term $\mathbb{P}_x\left(x^\eta_{\tau^\eta_\mathcal{V}} \in \mathcal{V}_j, \tau^\eta_\mathcal{V} < N\right)$ for this choice of $N$.

For this, we show that $x^\eta_{\tau^\eta_\mathcal{V}} \in \mathcal{V}_j$ with $\tau^\eta_\mathcal{V} < N$ implies that

$$\mathrm{dist}_N\left(x^\eta, \Gamma_N^{\{x\}}(\mathcal{B}_{i,j} - 2\varepsilon)\right) > \frac{\delta'}{2}. \tag{C.16}$$

Indeed, on the event $x^\eta_{\tau^\eta_\mathcal{V}} \in \mathcal{V}_j$ with $\tau^\eta_\mathcal{V} < N$, there is some $N' \leq N$ such that $\tau^\eta_\mathcal{V} = N'-1$. If $\mathrm{dist}_N\left(x^\eta, \Gamma_N^{\{x\}}(\mathcal{B}_{i,j} - 2\varepsilon)\right) > \frac{\delta'}{2}$ did not hold, this would mean that there exists $\xi \in (\mathbb{R}^d)^{N'}$ such that $\mathrm{dist}_{N'}(x^\eta, \xi) < \delta'$, $\xi_0 = x$ and, $\mathcal{A}_{N'}(\xi) \leq \mathcal{B}_{i,j} - 2\varepsilon$. In particular, $\xi$ would also satisfy $\xi_{N'-1} \in \mathcal{U}_{\delta'}(\mathcal{V}_j)$, $\xi_n \in \mathcal{U}_{\delta'}(\mathbb{R}^d \setminus \mathcal{V})$ for all $n = 1, \ldots, N'-2$ and, as a consequence, $\xi \in \mathcal{D}_r(N')$. This would be thus in direct contradiction of (C.12).

Therefore, we have that

$$\begin{aligned}
\mathbb{P}_x\left(x^\eta_{\tau^\eta_\mathcal{V}} \in \mathcal{V}_j, \tau^\eta_\mathcal{V} < N\right) &\leq \mathbb{P}_x\left(\mathrm{dist}_N\left(x^\eta, \Gamma_N^{\{x\}}(\mathcal{B}_{i,j} - 2\varepsilon)\right) > \frac{\delta'}{2}\right) \\
&\leq \exp\left(-\frac{\mathcal{B}_{i,j} - 3\varepsilon}{\eta}\right),
\end{aligned} \tag{C.17}$$

by Proposition B.1.

Combining this bound with (C.15) yields

$$\mathbb{P}_x(x^\eta_{\tau^\eta_\mathcal{V}} \in \mathcal{V}_j) \leq \exp\left(-\frac{\mathcal{B}_{i,j} - 3\varepsilon}{\eta}\right) + \exp\left(\frac{-\mathcal{B}_{i,j}}{\eta}\right), \tag{C.18}$$

which concludes the proof when $\mathcal{B}_{i,j} < +\infty$.

Let us now examine the case where $\mathcal{B}_{i,j} = +\infty$. Again, we have, for any $N \geq 0$,

$$\mathbb{P}_x\left(x^\eta_{\tau^\eta_\mathcal{V}} \in \mathcal{V}_i\right) \leq \mathbb{P}_x\left(x^\eta_{\tau^\eta_\mathcal{V}} \in \mathcal{V}_i, \tau^\eta_\mathcal{V} < N\right) + \mathbb{P}_x\left(\tau^\eta_\mathcal{V} \geq N\right). \tag{C.19}$$

We first bound the second probability using Lemma B.3 applied to $\mathcal{U} \leftarrow \mathcal{V}_i$. Take $N$ such that $\alpha_0(a-N) + \eta_0 b \leq -A$. Then, by Markov's inequality and Lemma B.3, it holds that for all $\eta \leq \eta_0$

$$\begin{aligned}
\mathbb{P}_x\left(\tau^\eta_\mathcal{V} \geq N\right) &\leq \mathbb{P}_x\left(\exp\left(\frac{\alpha_0 \tau^\eta_\mathcal{V}}{\eta}\right) \geq \exp\left(\frac{\alpha_0 N}{\eta}\right)\right) \\
&\leq \exp\left(\frac{\alpha_0(a-N)}{\eta} + b\right)
\end{aligned}$$

$$\leq \exp\left(\frac{-A}{\eta}\right). \tag{C.20}$$

We now bound the term $\mathbb{P}_x\left(x^\eta_{\tau^\eta_{\mathcal{V}}} \in \mathcal{V}_j, \ \tau^\eta_{\mathcal{V}} < N\right)$ for this choice of $N$.

For this, we show that $x^\eta_{\tau^\eta_{\mathcal{V}}} \in \mathcal{V}_j$ with $\tau^\eta_{\mathcal{V}} < N$ implies that

$$\text{dist}_N\left(x^\eta, \Gamma^{\{x\}}_N(A - \varepsilon)\right) > \frac{\delta'}{2}. \tag{C.21}$$

Indeed, on the event $x^\eta_{\tau^\eta_{\mathcal{V}}} \in \mathcal{V}_j$ with $\tau^\eta_{\mathcal{V}} < N$, there is some $N' \leq N$ such that $\tau^\eta_{\mathcal{V}} = N' - 1$. If $\text{dist}_N\left(x^\eta, \Gamma^{\{x\}}_N(A - \varepsilon)\right) > \frac{\delta'}{2}$ did not hold, this would mean that there exists $\xi \in (\mathbb{R}^d)^{N'}$ such that $\text{dist}_{N'}(x^\eta, \xi) < \delta'$, $\xi_0 = x$ and, $\mathcal{A}_{N'}(\xi) \leq A - \varepsilon$. In particular, $\xi$ would also satisfy $\xi_{N'-1} \in \mathcal{U}_{\delta'}(\mathcal{V}_j)$, $\xi_n \in \mathcal{U}_{\delta'}(\mathbb{R}^d \setminus \mathcal{V})$ for all $n = 1, \dots, N' - 2$ and, as a consequence, $\xi \in \mathcal{D}_r(N')$. This would be thus in direct contradiction of (C.12).

Therefore, we have that

$$\mathbb{P}_x\left(x^\eta_{\tau^\eta_{\mathcal{V}}} \in \mathcal{V}_j, \ \tau^\eta_{\mathcal{V}} < N\right) \leq \mathbb{P}_x\left(\text{dist}_N\left(x^\eta, \Gamma^{\{x\}}_N(A - \varepsilon)\right) > \frac{\delta'}{2}\right)$$

$$\leq \exp\left(-\frac{A - 2\varepsilon}{\eta}\right), \tag{C.22}$$

by Proposition B.1.

Combining this bound with (C.20) yields

$$\mathbb{P}_x(x^\eta_{\tau^\eta_{\mathcal{V}}} \in \mathcal{V}_j) \leq \exp\left(-\frac{A - 2\varepsilon}{\eta}\right) + \exp\left(\frac{-A}{\eta}\right), \tag{C.23}$$

which concludes the proof.

∎

**Lemma C.4.** *For any $\varepsilon > 0$, for any neighborhoods $\mathcal{V}_i$ of $\mathcal{C}_i$, $i = 1, \dots, N_{crit}$ small enough, there exists $\eta_0 > 0$ such that for all $i \in \{1, \dots, K\}$, $j \in \{1, \dots, N_{crit}\}$, $x \in \mathcal{V}_i$, $0 < \eta < \eta_0$,*

$$\mathbb{Q}_{\mathcal{V}}(x, \mathcal{V}_j) \geq \exp\left(-\frac{\mathcal{B}_{i,j}}{\eta} - \frac{\varepsilon}{\eta}\right). \tag{C.24}$$

Note that this result is trivially valid if $\mathcal{B}_{i,j} = +\infty$.

*Proof.* For any $(i,j) \in \{1, \dots, K\} \times \{1, \dots, N_{crit}\}$, there exists $N_{i,j} \geq 1$, $\xi^{i,j} \in \mathcal{D}_r(N_{i,j})$ such that $\xi^{i,j}_0 \in \mathcal{C}_i$, $\xi^{i,j}_{N_{i,j}-1} \in \mathcal{C}_j$, $\xi^{i,j}_n \notin \bigcup_{l \neq i,j} \mathcal{C}_l$ for all $n = 1, \dots, N_{i,j} - 2$ and $\mathcal{A}_{N_{i,j}}(\xi^{i,j}) \leq \mathcal{B}_{i,j} + \varepsilon$.

Define $\delta_{i,j} := \min\left\{d(\xi^{i,j}_n, \bigcup_{l \neq i,j} \mathcal{C}_l) : n = 1, \dots, N_{i,j} - 2\right\}$ and $\delta := \min_{i,j \in I} \delta_{i,j}$. By construction, it holds that $\delta > 0$.

Without loss of generality, at the potential cost of reducing $\varepsilon$, we can assume that Lemma B.6 can be applied to every $\mathcal{C}_i$, $i = 1, \dots, K$ with $r \leftarrow \varepsilon$ and denote by $\mathcal{W}_i$ the corresponding neighborhoods of $\mathcal{C}_i$. Require that $\mathcal{V}_i$ be contained in $\mathcal{W}_i \cap \mathcal{U}_{\delta/2}(\mathcal{K}_i)$ for all $i = 1, \dots, K$. Now, given such $\mathcal{V}_i$ neighborhoods of $\mathcal{C}_i$, $i = 1, \dots, N_{crit}$, by compactness, there exists $0 < \delta' \leq \delta/2$ such that $\mathcal{U}_{\delta'}(\mathcal{C}_i)$ is contained in $\mathcal{V}_i$ for all $i = 1, \dots, N_{crit}$.

Apply Lemma B.7 to $\mathcal{C}_i$, $i = 1, \dots, K$ with $\varepsilon \leftarrow \min(\varepsilon, \delta'/2)$ and denote by $N_i$ the bound on the length of paths obtained.

We are now ready to prove the result. Fix $(i,j) \in \{1, \dots, K\} \times \{1, \dots, N_{crit}\}$ and consider $x \in \mathcal{V}_i$. Since $\mathcal{V}_i \subset \mathcal{W}_i$, there exists $z \in \mathcal{K}_i$ such that $\rho(x, z) < \varepsilon$.

By Lemma B.7, there exists $n \leq N$, $\xi \in \mathcal{D}_r(n)$ such that $\xi_0 = z$, $\xi_{N-1} = \xi^{i,j}_0$, $\xi_k \in \mathcal{U}_{\delta'/2}(\mathcal{K}_i)$ for all $k = 1, \dots, n - 2$ and $\mathcal{A}_n(\xi) < \varepsilon$.

Considering the concatenation

$$\zeta := \left( x, \xi_0, \xi_1, \ldots, \xi_{n-2}, \xi_{n-1}, \xi_1^{i,j}, \ldots, \xi_{N_{i,j}-1}^{i,j} \right) \tag{C.25}$$

which is a path of length $n + N_{i,j} + 1$ made of $x \in \mathcal{V}_i$, then exactly $n$ points in $\mathcal{U}_{\delta'/2}(\mathcal{K}_i)$ then $N_{i,j} - 2$ in $\mathbb{R}^d \setminus \mathcal{U}_{\delta/2}(\mathcal{V})$ and $\xi_{N_{i,j}-1}^{i,j} \in \mathcal{K}_j$. Moreover, by construction, $\mathcal{A}_{N+N_{i,j}}(\zeta) \leq \widetilde{\mathcal{B}}_{i,j} + 3\varepsilon$. Therefore, if

$$\mathrm{dist}_{n+N_{i,j}+1}\left( x^\eta, \zeta \right) < \delta'/2, \tag{C.26}$$

with $x_0^\eta = x$, then $x_1^\eta, \ldots, x_n^\eta$ are in $\mathcal{U}_{\delta'}(\mathcal{K}_i) \subset \mathcal{V}_i$ and, since $\delta' \leq \delta/2$, $x_{n+2}^\eta, \ldots, x_{n+N_{i,j}-1}^\eta$ are not in $\mathcal{U}_{\delta/4}(\mathcal{V})$, and therefore not in $\mathcal{V}$. Moreover, $x_{n+N_{i,j}}^\eta$ would be in $\mathcal{U}_{\delta'/2}(\mathcal{K}_j) \subset \mathcal{V}_j$.

Thus, all the paths $x^\eta$ satisfying (C.26) with $x_0^\eta = x$ correspond to exactly one transition of the induced chain $(z_n)_n$ from $x$ to $\mathcal{V}_j$.

Therefore, using the definition of $\mathbb{Q}_\mathcal{V}$, we have that

$$\begin{aligned}
\mathbb{Q}_\mathcal{V}(x, \mathcal{V}_j) &\geq \mathbb{P}_x\left( \mathrm{dist}_{n+N,j+1}\left( x^\eta, \zeta \right) < \delta'/2 \right) \\
&\geq \exp\left( -\frac{\mathcal{B}_{i,j} + 4\varepsilon}{\eta} \right),
\end{aligned} \tag{C.27}$$

by Proposition B.1.  ∎

## C.4 Accelerated induced chain

We will take the following convention, for any $i \in \{1, \ldots, N_{\mathrm{crit}}\}$, $\delta > 0$:

$$\mathcal{B}_{i,i}^\delta = \mathcal{B}_{i,i} = \mathcal{B}_{i,i} = 0. \tag{C.28}$$

Recall that $\mathbb{Q}_\mathcal{V}$ is the transition probabilities of the induced chain $(z_n)_{n \geq 0}$, defined in Definition 8. In other words $\mathbb{Q}_\mathcal{V}(x, \mathcal{V}_j)$ is the probability that the sequence of (B.16) starting from $x$ at time 0 enters $\mathcal{V}_j$ before any other $\mathcal{V}_l$ for $l \neq j$. Moreover, $K$ still denotes the number of connected components of the critical set of $f$ that are not part of the global minimum, see Definition 1 in Appendix B.2.

**Definition 9.** Given the induced chain $(z_n)_{n \geq 0}$, we define the accelerated induced chain $(z^K{}_n)_{n \geq 0}$ as follows:

$$z^K{}_n := z_{nK}. \tag{C.29}$$

We denote by $\mathbb{Q}^K{}_\mathcal{V}$ the transition probabilities of the accelerated induced chain which corresponds to the $K - th$ power of $\mathbb{Q}_\mathcal{V}$: it satisfies[12], for any $x \in \mathcal{V}_i$, $i \in \{1, \ldots, N_{\mathrm{crit}}\}$ and any measurable set $A \subset \mathbb{R}^d$,

$$\mathbb{Q}^K{}_\mathcal{V}(x, A) = \int \int \cdots \int \mathbb{1}\{z_K \in A\} \, \mathbb{Q}_\mathcal{V}(x, dz_1) \, \mathbb{Q}_\mathcal{V}(z_1, dz_2) \ldots \mathbb{Q}_\mathcal{V}(z_{K-1}, dz_K). \tag{C.30}$$

We now define transition costs for the accelerated induced chain, which correspond the ones introduced in the main text (19) in Section 4.

**Definition 10** (Inspired by Freidlin & Wentzell [22, Chap. 6,§2]). For $i \in \{1, \ldots, K\}$, $j \in \{1, \ldots, N_{\mathrm{crit}}\}$, $\delta > 0$,

$$B_{i,j} := \inf\left\{ \mathcal{A}_N(\xi) : N \geq 1, \xi \in \mathcal{D}_r(N), \xi_0 \in \mathcal{C}_i, \xi_{N-1} \in \mathcal{C}_j, \xi_n \notin \bigcup_{l \neq i, j, \, l > K} \mathcal{C}_l \text{ for all } n = 1, \ldots, N-2 \right\} \tag{C.31}$$

**Lemma C.5.** *For any* $(i, j) \in \{1, \ldots, K\} \times \{1, \ldots, N_{crit}\}$,

$$B_{i,j} = \min\left\{ \sum_{l=0}^{n-2} \mathcal{B}_{i_l, i_{l+1}} : i_0 = i, \, i_{n-1} = j, \, i_l \in \{1, \ldots, K\} \text{ for } l = 1, \ldots, n-2, \, n \geq 1 \right\} \tag{C.32}$$

$$= \min\left\{ \sum_{l=0}^{K-1} \mathcal{B}_{i_l, i_{l+1}} : i_0 = i, \, i_K = j, \, i_l \in \{1, \ldots, K\} \text{ for } l = 1, \ldots, K-1 \right\}. \tag{C.33}$$

---

[12]This equation is sometimes referred to as the Chapman-Kolmogorov equation.

*Proof.* We focus on proving the first equality for $B_{i,j}$; the proof for $\mathcal{B}_{i,j}$ follows similarly. Let us denote by $\mathrm{RHS}(i,j)$ the right-hand side:

$$\mathrm{RHS}(i,j) := \min\left\{ \sum_{l=0}^{n-2} \mathcal{B}_{i_l, i_{l+1}} : i_0 = i, \, i_{n-1} = j, \, i_l \in \{1, \ldots, K\} \text{ for } l = 1, \ldots, n-2, \, n \geq 1 \right\} \tag{C.34}$$

We will prove that $B_{i,j} = \mathrm{RHS}(i,j)$.

($\geq$): Let us show that $B_{i,j} \geq \mathrm{RHS}(i,j)$. Fix $\varepsilon > 0$. There is a path $\xi \in \mathcal{D}_r(N)$ with $\xi_0 \in \mathcal{C}_i$, $\xi_{N-1} \in \mathcal{C}_j$ that avoids $\bigcup_{l \neq i, j, \, l > K} \mathcal{C}_l$ so that, by definition of $B_{i,j}$ (Definition 10),

$$B_{i,j} \geq \mathcal{A}_N(\xi) - \varepsilon. \tag{C.35}$$

Let $1 \leq n_1 < n_2 < \cdots < n_{N'} < N$ be the times when $\xi$ belongs to some $\mathcal{C}_l$ with $l \leq K$. Define $i_0 = i$, $i_{N'} = j$ and for $1 \leq i < N'$, let $i_k$ be such that $\xi_{n_k} \in \mathcal{C}_{i_k}$. Then, by definition of $\mathcal{B}_{i_k, i_{k+1}}$ (Definition 7),

$$\mathcal{A}_N(\xi) \geq \sum_{k=0}^{N'-1} \mathcal{B}_{i_k, i_{k+1}}. \tag{C.36}$$

Combining (C.35) and (C.36) and taking yields

$$B_{i,j} \geq \sum_{k=0}^{N'-1} \mathcal{B}_{i_k, i_{k+1}} - \varepsilon \tag{C.37}$$

$$\geq \mathrm{RHS}(i,j) - \varepsilon, \tag{C.38}$$

since this sequence $(i_k)_{k=0}^{N'}$ satisfies $i_0 = i$, $i_{N'} = j$ and $i_k \in \{1, \ldots, K\}$ for $1 \leq k < N'$.

($\leq$): Let us show that $B_{i,j} \leq \mathrm{RHS}(i,j)$. Take $\varepsilon > 0$ and let $(i_k)_{k=0}^{N'}$ be a sequence achieving the minimum in $\mathrm{RHS}(i,j)$ up to $\varepsilon$, i.e.,

$$\sum_{k=0}^{N'-1} \mathcal{B}_{i_k, i_{k+1}} \leq \mathrm{RHS}(i,j) + \varepsilon. \tag{C.39}$$

Take $\delta > 0$ small enough such that $\mathcal{U}_\delta(\mathcal{C}_l)$, $l = 1, \ldots, N_{\mathrm{crit}}$ are pairwise disjoint.

For each $0 \leq k < N'$, by definition of $\mathcal{B}_{i_k, i_{k+1}}$ (Definition 7), there exists $\delta_k \in (0, \delta)$ and a path $\xi^k \in \mathcal{D}_r(N_k)$ such that: (i) $\xi_0^k \in \mathcal{C}_{i_k}$, (ii) $\xi_{N_k-1}^k \in \mathcal{C}_{i_{k+1}}$, (iii) $\xi_n^k \notin \bigcup_{l \neq i_k, i_{k+1}} \mathcal{C}_l$ for all $n = 1, \ldots, N_k - 2$, and (iv) $\mathcal{A}_{N_k}(\xi^k) \leq \mathcal{B}_{i_k, i_{k+1}} + \varepsilon/N'$.

By Lemma B.7, for each $0 \leq k < N' - 1$, there exists $\tilde{\xi}^k \in \mathcal{D}_r(\tilde{N}_k)$ such that: (i) $\tilde{\xi}_0^k = \xi_{N_k-1}^k$, (ii) $\tilde{\xi}_{\tilde{N}_k-1}^k = \xi_0^{k+1}$, (iii) $\tilde{\xi}_n^k \in \mathcal{U}_{\delta/2}(\mathcal{C}_{i_{k+1}})$ and therefore $\tilde{\xi}_n^k \notin \bigcup_{l \neq i_{k+1}} \mathcal{C}_l$ for all $n = 1, \ldots, \tilde{N}_k - 2$, and (iv) $\mathcal{A}_{\tilde{N}_k}(\tilde{\xi}^k) \leq \varepsilon/N'$.

Concatenating the paths $\xi^0, \tilde{\xi}^0, \xi^1, \tilde{\xi}^1, \ldots, \xi^{N'-1}$ yields a path $\zeta$ that: (i) starts in $\mathcal{C}_i$, (ii) ends in $\mathcal{C}_j$, (iii) avoids $\bigcup_{l \neq i, j, \, l > K} \mathcal{C}_l$, and (iv) has total cost at most $\sum_{k=0}^{N'-1} \mathcal{B}_{i_k, i_{k+1}} + 3\varepsilon$. Therefore, by definition of $B_{i,j}$ (Definition 10),

$$B_{i,j} \leq \sum_{k=0}^{N'-1} \mathcal{B}_{i_k, i_{k+1}} + 3\varepsilon$$

$$\leq \mathrm{RHS}(i,j) + 4\varepsilon. \tag{C.40}$$

The second equality in each case follows from the fact that optimal paths between different components can be chosen without cycles (since all costs are non-negative), and therefore their length can be bounded by the number of components $K$.

$\blacksquare$

**Lemma C.6** (Upper bound on accelerated induced chain transition probability)**.** *For any $\varepsilon > 0$, $A > 0$, for any small enough neighborhoods $\mathcal{V}_i$ of $\mathcal{C}_i$, $i = 1, \ldots, N_{\mathrm{crit}}$, there is some $\eta_0 > 0$ such that for all $i \in \{1, \ldots, K\}$, $j \in \{1, \ldots, N_{\mathrm{crit}}\}$, $x \in \mathcal{V}_i$, $0 < \eta < \eta_0$,*

$$\mathbb{Q}^K_{\mathcal{V}}(x, \mathcal{V}_j) \leq \begin{cases} \exp\left(-\dfrac{B_{i,j}}{\eta} + \dfrac{\varepsilon}{\eta}\right) & \text{if } B_{i,j} < +\infty, \\ \exp\left(-\dfrac{A}{\eta}\right) & \text{otherwise}. \end{cases} \tag{C.41}$$

*Proof.* Fix $i \in \{1, \ldots, K\}$, $j \in \{1, \ldots, N_{\text{crit}}\}$. By Lemma C.5, we have that

$$B_{i,j} = \min\left\{ \sum_{l=0}^{K-1} \mathcal{B}_{i_l, i_{l+1}} : i_0 = i, \ i_K = j, \ i_l \in \{1, \ldots, K\} \text{ for } l = 1, \ldots, K-1 \right\}. \tag{C.42}$$

First, let us consider the case where $B_{i,j} < +\infty$. Take $\varepsilon > 0$ and let $(i_k)_{k=0}^K$ be a sequence achieving the minimum in (C.42) up to $\varepsilon$, i.e.,

$$\sum_{k=0}^{K-1} \mathcal{B}_{i_k, i_{k+1}} \leq B_{i,j} + \varepsilon. \tag{C.43}$$

By definition of the accelerated induced chain (Definition 9), we have that for any $x \in \mathcal{V}_i$,

$$\mathbb{Q}^{\mathrm{K}}{}_{\mathcal{V}}(x, \mathcal{V}_j) = \sum_{i_1, \ldots, i_{K-1} \in \{1, \ldots, K\}} \int_{\mathcal{V}_{i_1}} \cdots \int_{\mathcal{V}_{i_{K-1}}} \prod_{k=0}^{K-1} \mathbb{Q}_{\mathcal{V}}(x_k, dx_{k+1}) \, \mathbb{Q}_{\mathcal{V}}(x_K, \mathcal{V}_j), \tag{C.44}$$

where we define $x_0 = x$.

Therefore, combining (C.44) with Lemma C.3 and (C.43), we obtain that for any $x \in \mathcal{V}_i$,

$$\begin{aligned}
\mathbb{Q}^{\mathrm{K}}{}_{\mathcal{V}}(x, \mathcal{V}_j) &\leq \sum_{i_1, \ldots, i_{K-1} \in \{1, \ldots, K\}} \prod_{k=0}^{K-1} \exp\left( -\frac{\mathcal{B}_{i_k, i_{k+1}}}{\eta} + \frac{\varepsilon}{\eta K} \right) \\
&\leq K^{K-1} \exp\left( -\frac{B_{i,j}}{\eta} + \frac{2\varepsilon}{\eta} \right).
\end{aligned} \tag{C.45}$$

For the case where $B_{i,j} = +\infty$, by the same reasoning but using the alternative bound from Lemma C.3, we get

$$\mathbb{Q}^{\mathrm{K}}{}_{\mathcal{V}}(x, \mathcal{V}_j) \leq K^{K-1} \exp\left( -\frac{A}{\eta} \right). \tag{C.46}$$

Taking $\eta_0$ small enough so that $K^{K-1} \leq \exp(\varepsilon/\eta)$ for all $\eta \leq \eta_0$, the result follows from (C.45) and (C.46). ∎

**Lemma C.7** (Lower bound on accelerated induced chain transition probability). *For any $\varepsilon > 0$, for any small enough neighborhoods $\mathcal{V}_i$ of $\mathcal{C}_i$, $i = 1, \ldots, N_{\text{crit}}$, there is some $\eta_0 > 0$ such that for all $i \in \{1, \ldots, K\}$, $j \in \{1, \ldots, N_{\text{crit}}\}$, $x \in \mathcal{V}_i$, $0 < \eta < \eta_0$,*

$$\mathbb{Q}^{\mathrm{K}}{}_{\mathcal{V}}(x, \mathcal{V}_j) \geq \exp\left( -\frac{B_{i,j}}{\eta} - \frac{\varepsilon}{\eta} \right). \tag{C.47}$$

The proof of Lemma C.7 is very similar to the proof of Lemma C.6 and is therefore omitted.

We end this section by restating a result from [5] that provides sufficient conditions for $B_{i,j}$ to be finite for any $(i, j) \in \{1, \ldots, K\} \times \{1, \ldots, N_{\text{crit}}\}$.

**Lemma C.8** ([5, Lem. D.12]). *Consider $x, x' \in \mathbb{R}^d$ and assume that there exists $T \in \mathbb{N}$, $\gamma \in \mathcal{C}^1([0, T], \mathbb{R}^d)$ such that $\gamma_0 = x$, $\gamma_T = x'$, $\gamma_n \notin \bigcup_{l \neq i, j, \, l > K} \mathcal{C}_l$ for all $n = 1, \ldots, T-1$ and, for every $t \in [0, T]$, $\nabla f(\gamma_t)$ is in the interior of the closed convex hull of the support of $u(\gamma_t, \omega)$, i.e.,*

$$\nabla f(\gamma_t) \in \operatorname{int} \overline{\operatorname{conv}} \operatorname{supp} u(\gamma_t, \omega). \tag{C.48}$$

*Then, $\mathcal{B}(x, x') < +\infty$.*

## C.5 Initial transition

**Definition 11** (Inspired by Freidlin & Wentzell [22, Chap. 6, §2]). For $j \in \{1, \ldots, N_{\text{crit}}\}$, $\delta > 0$, $x \in \mathbb{R}^d$,

$$\mathcal{B}_{x,j} := \inf\left\{ \mathcal{A}_N(\xi) : N \geq 1, \xi \in \mathcal{D}_r(N), \xi_0 \in x, \xi_{N-1} \in \mathcal{C}_j, \xi_n \notin \bigcup_{l \neq i, j} \mathcal{C}_l \text{ for all } n = 1, \ldots, N-2 \right\} \tag{C.49}$$

$$\mathcal{B}^{\delta}_{x,j} := \inf\left\{ \mathcal{A}_N(\xi) : N \geq 1, \xi \in \mathcal{D}_r(N), \xi_0 = x, \xi_{N-1} \in \mathcal{U}_{\delta}(\mathcal{C}_j), \xi_n \notin \bigcup_{l \neq i, j} \mathcal{C}_l \text{ for all } n = 1, \ldots, N-2 \right\}. \tag{C.50}$$

The proof of the following lemmas are very similar to the ones of Lemmas C.2–C.4 and are therefore omitted.

**Lemma C.9.** *For any $j \in \{1, \ldots, N_{crit}\}$, $\delta > 0$, $x \in \mathbb{R}^d$,*

$$\mathcal{B}_{x,j} = \lim_{\delta \to 0} \mathcal{B}^{\delta}_{x,j}. \tag{C.51}$$

**Lemma C.10.** *For any $\varepsilon > 0$, $x \in \mathbb{R}^d$, for any neighborhoods $\mathcal{V}_i$ of $\mathcal{C}_i$, $i = 1, \ldots, N_{crit}$ small enough, there exists $\eta_0 > 0$ such that for all $j \in \{1, \ldots, N_{crit}\}$, $0 < \eta < \eta_0$,*

$$\mathbb{Q}_{\mathcal{V}}(x, \mathcal{V}_j) \geq \exp\left(-\frac{\mathcal{B}_{x,j}}{\eta} - \frac{\varepsilon}{\eta}\right). \tag{C.52}$$

**Lemma C.11.** *For any $\varepsilon > 0$, $A > 0$, $x \in \mathbb{R}^d$, for any small enough neighborhoods $\mathcal{V}_i$ of $\mathcal{C}_i$, $i = 1, \ldots, N_{crit}$, there is some $\eta_0 > 0$ such that for all $j \in \{1, \ldots, N_{crit}\}$, $0 < \eta < \eta_0$,*

$$\mathbb{Q}_{\mathcal{V}}(x, \mathcal{V}_j) \leq \begin{cases} \exp\left(-\frac{\mathcal{B}_{x,j}}{\eta} + \frac{\varepsilon}{\eta}\right) & \text{if } \mathcal{B}_{x,j} < +\infty, \\ \exp\left(-\frac{A}{\eta}\right) & \text{otherwise}. \end{cases} \tag{C.53}$$

Following the same methodology as for transitions between critical components, we now establish similar results for transitions starting from an arbitrary initial state.

**Definition 12** (Inspired by Freidlin & Wentzell [22, Chap. 6,§2]). For $x \in \mathbb{R}^d$, $j \in \{1, \ldots, N_{\text{crit}}\}$, $\delta > 0$,

$$B_{x,j} := \inf\left\{\mathcal{A}_N(\xi) : N \geq 1, \xi \in \mathcal{D}_r(N), \xi_0 = x, \xi_{N-1} \in \mathcal{C}_j, \xi_n \notin \bigcup_{l \neq j, l > K} \mathcal{C}_l \text{ for all } n = 1, \ldots, N - 2\right\} \tag{C.54}$$

This definition mirrors the one of Definition 10, adapting it to account for an arbitrary initial state rather than starting from a critical component. We can then establish the following decomposition result, analogous to Lemma C.5.

**Lemma C.12.** *For any $x \in \mathbb{R}^d$, $j \in \{1, \ldots, N_{crit}\}$,*

$$B_{x,j} = \min\left\{\mathcal{B}_{x,i_1} + \sum_{l=1}^{n-2} \mathcal{B}_{i_l, i_{l+1}} : i_{n-1} = j, i_l \in \{1, \ldots, K\} \text{ for } l = 1, \ldots, n-2, n \geq 1\right\} \tag{C.55}$$

$$= \min\left\{\mathcal{B}_{x,i_1} + \sum_{l=1}^{K-1} \mathcal{B}_{i_l, i_{l+1}} : i_K = j, i_l \in \{1, \ldots, K\} \text{ for } l = 1, \ldots, K-1\right\} \tag{C.56}$$

The proof follows the same arguments as in Lemma C.5, replacing the initial component with the given initial state.

These structural results allow us to establish bounds on the transition probabilities from an arbitrary initial state, paralleling those of Lemmas C.6 and C.7.

**Lemma C.13** (Upper bound on accelerated induced chain initial transition probability). *For any $x \in \mathbb{R}^d$, for any $\varepsilon > 0$, $A > 0$, for any small enough neighborhoods $\mathcal{V}_i$ of $\mathcal{C}_i$, $i = 1, \ldots, N_{crit}$, there is some $\eta_0 > 0$ such that for all $j \in \{1, \ldots, N_{crit}\}$, $0 < \eta < \eta_0$,*

$$\mathbb{Q}^{\mathrm{K}}_{\mathcal{V}}(x, \mathcal{V}_j) \leq \begin{cases} \exp\left(-\frac{B_{x,j}}{\eta} + \frac{\varepsilon}{\eta}\right) & \text{if } B_{x,j} < +\infty, \\ \exp\left(-\frac{A}{\eta}\right) & \text{otherwise}. \end{cases} \tag{C.57}$$

**Lemma C.14** (Lower bound on accelerated induced chain initial transition probability). *For any $x \in \mathbb{R}^d$, for any $\varepsilon > 0$, for any small enough neighborhoods $\mathcal{V}_i$ of $\mathcal{C}_i$, $i = 1, \ldots, N_{crit}$, there is some $\eta_0 > 0$ such that for all $j \in \{1, \ldots, N_{crit}\}$, $0 < \eta < \eta_0$,*

$$\mathbb{Q}^{\mathrm{K}}_{\mathcal{V}}(x, \mathcal{V}_j) \geq \exp\left(-\frac{B_{x,j}}{\eta} - \frac{\varepsilon}{\eta}\right). \tag{C.58}$$

The proofs of these last two lemmas follow very closely those of Lemmas C.6 and C.7, with the main modification being the treatment of the initial state instead of an initial component. The key arguments involving the Chapman-Kolmogorov equation and the handling of the intermediate transitions remain essentially unchanged.

## C.6 Transition time

Let us begin with a preliminary escape result.

**Lemma C.15.** *Given $C$ a connected components of $\mathcal{K}_{crit}$, for any $\varepsilon > 0$, for $\eta > 0$ small enough and $\mathcal{V}$ a small enough neighborhood of $C$, it holds that, for any $x \in \mathcal{V}$,*

$$\mathbb{E}_x[\sigma_{\mathcal{V}}^{\eta}] \leq \exp\left(\frac{\varepsilon}{\eta}\right). \tag{C.59}$$

*Proof.* Fix $\varepsilon > 1$. By Lemma B.6, $\mathcal{W}_{\varepsilon}(C)$ is an open neighborhood of $C$. Since $C$ is closed, $\mathcal{W}_{\varepsilon}(C) \setminus C$ is not empty and open. Therefore, there exists $x_3 \in \mathcal{W}_{\varepsilon}(C) \setminus C$ and $\delta > 0$ such that $\mathbb{B}(x_3, \delta) \subset \mathcal{W}_{\varepsilon}(C) \setminus C$. By construction, there exists $x_2 \in C$ such that $\rho(x_2, x_3) < \varepsilon$.

Let us now apply Lemma B.7 to obtain that there exists $N \geq 2$ such that, for any $x_1 \in C$, there exists $n \leq N$, $\xi \in \mathcal{D}_r(N)$ such that $\xi_0 = x_1, \xi_{N-1} = x_2, \mathcal{A}_n(\xi) < \varepsilon$.

Require then that $\mathcal{V}$ be contained in $\mathcal{W}_{\varepsilon}(C) \setminus \overline{\mathbb{B}}(x_3, \delta/2)$ which is an open neighborhood of $C$.

Take $x \in \mathcal{V}$. Since $\mathcal{V}$ is in particular contained in $\mathcal{W}_{\varepsilon}(C)$, there exists $x_1 \in C$ such that $\rho(x, x_1) < \varepsilon$. By our application of Lemma B.7 above, there exists $n \leq N$, $\xi \in \mathcal{D}_r(n)$ such that $\xi_0 = x_1, \xi_{n-1} = x_2, \mathcal{A}_n(\xi) < \varepsilon$. Now consider the path $\zeta \in \mathcal{D}_r(n+2)$ defined by

$$\zeta = (x, \xi_0, \xi_1, \ldots, \xi_{n-1}, x_3), \tag{C.60}$$

which satisfies

$$\mathcal{A}_{n+3}(\zeta) = \rho(x, x_1) + \mathcal{A}_n(\xi) + \rho(x_2, x_3) < 3\varepsilon. \tag{C.61}$$

If a trajectory of SGD $x^{\eta}$ with $x^{\eta}{}_0 = x$ satisfies $\mathrm{dist}_{n+2}(x^{\eta}, \zeta) < \delta/2$, then $x_{n+1}^{\eta}$ is inside $\mathbb{B}(x_3, \delta/2)$ and therefore not in $\mathcal{V}$.

Hence, we have, for $\eta > 0$ small enough,

$$\begin{aligned}
\mathbb{P}_x(\sigma_{\mathcal{V}}^{\eta} < N+3) &\geq \mathbb{P}_x(\sigma_{\mathcal{V}}^{\eta} < n+3) \\
&\geq \mathbb{P}_x(\mathrm{dist}_{n+2}(x^{\eta}, \zeta) < \delta/2) \\
&\geq \exp\left(-\frac{\mathcal{A}_{n+2}(\zeta) + \varepsilon}{\eta}\right) \\
&\geq \exp\left(-\frac{4\varepsilon}{\eta}\right),
\end{aligned} \tag{C.62}$$

where we invoked Proposition B.1 and used (C.61).

For any $x \in \mathcal{V}$, for $n \geq 2$, the (weak) Markov property yields that

$$\begin{aligned}
\mathbb{P}_x\left(\sigma_{\mathcal{V}}^{\eta} > n(N+2)\right) &= \mathbb{E}_x\left[\mathbb{1}\{\sigma_{\mathcal{V}}^{\eta} > N+2\}\,\mathbb{E}_{x_{N+2}^{\eta}}\left[\mathbb{1}\{\sigma_{\mathcal{V}}^{\eta} > N+2\} \ldots \mathbb{E}_{x_{(n-2)(N+1)}^{\eta}}\left[\mathbb{1}\{\sigma_{\mathcal{V}}^{\eta} > N+2\}\right]\ldots\right]\right] \\
&\leq \left(2 - \exp\left(-\frac{4\varepsilon}{\eta}\right)\right)^n,
\end{aligned} \tag{C.63}$$

with (C.62).

We can now finally estimate the expectation of $\sigma_{\mathcal{V}}^{\eta}$: for $x \in \mathcal{V}$,

$$\begin{aligned}
\mathbb{E}_x[\sigma_{\mathcal{V}}^{\eta}] &= \sum_{n=1}^{\infty} \mathbb{P}_x(\sigma_{\mathcal{V}}^{\eta} > n) \\
&= \sum_{n=1}^{\infty} \sum_{k=0}^{N} \mathbb{P}_x(\sigma_{\mathcal{V}}^{\eta} > n(N+1) + k) \\
&\leq (N+2) \sum_{n=0}^{\infty} \mathbb{P}_x(\sigma_{\mathcal{V}}^{\eta} > n(N+1))
\end{aligned}$$

$$\leq (N + 2) \sum_{n=0}^{\infty} \left( 1 - \exp\left( -\frac{4\varepsilon}{\eta} \right) \right)^n$$

$$= (N + 2) \exp\left( \frac{4\varepsilon}{\eta} \right), \tag{C.64}$$

where we used (C.63).

Finally, taking $\eta > 0$ small enough so that $N + 1 \leq \exp\left( \frac{\varepsilon}{\eta} \right)$ yields that

$$\mathbb{E}_x \left[ \sigma_{\mathcal{V}}^{\eta} \right] \leq \exp\left( \frac{5\varepsilon}{\eta} \right), \tag{C.65}$$

which concludes the proof. ∎

**Lemma C.16.** *For any $\mathcal{X}_0 \subset \mathbb{R}^d$ compact, for any $\varepsilon > 0$, for any small enough neighborhoods $\mathcal{V}_i$ of $\mathcal{C}_i$, $i = 1, \ldots, N_{crit}$, there is some $\eta_0 > 0$ such for all $0 < \eta < \eta_0$, for any $x \in \mathcal{X}_0 \setminus \cup_{i=K+1}^{N_{crit}} \mathcal{V}_i$,*

$$1 \leq \mathbb{E}_x [\tau_1^{\eta}] \leq e^{\frac{\varepsilon}{\eta}} . \tag{C.66}$$

*Proof.* Let us begin with the LHS inequality. There are two cases: either $x$ belongs to $\cup_{i=1}^K \mathcal{V}_i$ or not. If $x$ belongs to $\cup_{i=1}^K \mathcal{V}_i$, then by definition $\sigma_1^{\eta} \geq 1$ and therefore $\tau_1^{\eta} \geq 1$. If $x$ does not belong to $\cup_{i=1}^K \mathcal{V}_i$, then $\sigma_1^{\eta} = 0$ but necessarily $\tau_1^{\eta} \geq 1$. Hence, in all cases, $\tau_1^{\eta} \geq 1$ so that the LHS inequality holds.

We now turn to the RHS inequality. As before, we will separate the proof into two cases: either $x$ belongs to $\cup_{i=1}^K \mathcal{V}_i$ or not. If $x$ belongs to some $\mathcal{V}_i$ for some $i \in \{1, \ldots, K\}$ then, by Lemma C.15, we have that

$$\mathbb{E}_x [\sigma_1^{\eta}] = \mathbb{E}_x [\sigma_{\mathcal{V}_i}^{\eta}] \leq e^{\frac{\varepsilon}{\eta}} . \tag{C.67}$$

In particular, $\sigma_1^{\eta}$ is finite almost surely. Therefore, the strong Markov property implies that

$$\mathbb{E}_x [\tau_1^{\eta}] = \mathbb{E}_x [\sigma_1^{\eta}] + \mathbb{E}_x [\mathbb{E}_{x_{\sigma_1^{\eta}}^{\eta}} [\tau_{\mathcal{V}}^{\eta}]] . \tag{C.68}$$

Applying Lemma B.3 with $\mathcal{U} \leftarrow \cup_{i=1}^{N_{crit}} \mathcal{V}_i$ and $\mathbb{R}^d \leftarrow \mathcal{X}_0$, and using Jensen's inequality, we obtain that

$$\mathbb{E}_{x_{\sigma_1^{\eta}}^{\eta}} [\tau_{\mathcal{V}}^{\eta}] \leq a + \frac{b\eta_0}{\alpha_0} =: c . \tag{C.69}$$

Plugging this bound into (C.68) and using (C.67) yields

$$\mathbb{E}_x [\tau_1^{\eta}] \leq e^{\frac{\varepsilon}{\eta}} + c , \tag{C.70}$$

which concludes the proof of this case.

Finally, if $x$ does not belong to $\cup_{i=1}^K \mathcal{V}_i$, then one only needs to apply Lemma B.3 as above to obtain the result. ∎

**Corollary C.1.** *For any $\mathcal{X}_0 \subset \mathbb{R}^d$ compact, for any $N \geq 1$, for any $\varepsilon > 0$, for any small enough neighborhoods $\mathcal{V}_i$ of $\mathcal{C}_i$, $i = 1, \ldots, N_{crit}$, there is some $\eta_0 > 0$ such that for all $0 < \eta < \eta_0$, for any $x \in \mathcal{X}_0 \setminus \cup_{i=K+1}^{N_{crit}} \mathcal{V}_i$,*

$$1 \leq \mathbb{E}_x [\tau_N^{\eta}] \leq e^{\frac{\varepsilon}{\eta}} . \tag{C.71}$$

*Proof.* The lower bound follows directly from the definition of $\tau_N^{\eta}$ since $\tau_k^{\eta} \geq k$ for all $k \geq 1$ by construction.

For the upper bound, first note that we can write $\tau_N^{\eta}$ as a telescoping sum:

$$\tau_N^{\eta} = \sum_{k=0}^{N-1} (\tau_{k+1}^{\eta} - \tau_k^{\eta}) , \tag{C.72}$$

with the convention that $\tau_0^\eta = 0$. Therefore,

$$\mathbb{E}_x[\tau_N^\eta] = \sum_{k=0}^{N-1} \mathbb{E}_x[\tau_{k+1}^\eta - \tau_k^\eta] . \tag{C.73}$$

By the strong Markov property applied at time $\tau_k^\eta$ for each term, we have

$$\mathbb{E}_x[\tau_{k+1}^\eta - \tau_k^\eta] = \mathbb{E}_x[\mathbb{E}_{x_{\tau_k^\eta}^\eta}[\tau_1^\eta]]$$
$$\leq e^{\frac{\varepsilon}{\eta}} , \tag{C.74}$$

where the inequality follows from Lemma C.16 since $x_{\tau_k^\eta}^\eta$ belongs to $\cup_{i=1}^{N_{\text{crit}}} \mathcal{V}_i$ by definition of $\tau_k^\eta$ for all $k \geq 1$ (and for $k = 0$, we can apply Lemma C.16 directly to the initial point $x$).

Summing over $k$ from 0 to $N - 1$ yields

$$\mathbb{E}_x[\tau_N^\eta] \leq N e^{\frac{\varepsilon}{\eta}}$$
$$\leq e^{\frac{2\varepsilon}{\eta}} , \tag{C.75}$$

for $\eta$ small enough. ∎

# D  Finite-Time Analysis

## D.1  Markov Chains on finite state spaces

In this subsection, we introduce the necessary notation and then restate a key lemma from Freidlin & Wentzell [22, Chap. 6,§3].

Consider a finite set $\mathcal{V}$ and $\mathcal{Q} \subseteq \mathcal{V}$. Denote by $r := \operatorname{card} \mathcal{V} \setminus \mathcal{Q}$.

Given $p : (\mathcal{V} \setminus \mathcal{Q}) \times \mathcal{V} \to [0, 1]$, we define the probability of a $g$ with edges in $(\mathcal{V} \setminus \mathcal{Q}) \times \mathcal{V}$ as

$$\pi(g, p) := \prod_{i \to j \in g} p(i \to j) . \tag{D.1}$$

A graph $g$ consisting of arrows $j \to k$ with $j \in \mathcal{V} \setminus \mathcal{Q}$, $k \in \mathcal{V}$, $j \neq k$ is called a $\mathcal{Q}$-graph on $\mathcal{V}$ if

   (i)  Every vertex in $\mathcal{V} \setminus \mathcal{Q}$ has exactly one outgoing arrow.

   (ii)  There are no cycles, or, equivalently, from every vertex in $\mathcal{V} \setminus \mathcal{Q}$ there is a directed path to a vertex in $\mathcal{Q}$.

The set of $\mathcal{Q}$-graphs is denoted by $\mathcal{G}(\mathcal{Q})$.

We denote by $\mathcal{G}(i \nrightarrow \mathcal{Q})$ the set of graphs with exactly $\operatorname{card}(\mathcal{V} \setminus \mathcal{Q}) - 1$ edges from $\mathcal{V} \setminus \mathcal{Q}$ to $\mathcal{V}$, no cycles and no path from $i$ to $\mathcal{Q}$. Note that, equivalently, this set is made of all $\mathcal{Q}$-graphs from which a single edge from the path from $i$ to $\mathcal{Q}$ has been removed.

**Lemma D.1** (Freidlin & Wentzell [22, Chap. 6,§3,Lem. 3.4]). *Consider a Markov Chain on state space $X = \bigcup_{i \in \mathcal{V}} X_i$ with $X_i, i \in \mathcal{V}$ disjoint and non-empty, with transition probabilities that satisfy: for $i \in \mathcal{V} \setminus \mathcal{Q}$, $j \in \mathcal{V}$ with $i \neq j$,*

$$\forall x \in X_i, \quad a^{-1} p_{ij} \leq \mathbb{P}(x, X_j) \leq a p_{ij} , \tag{D.2}$$

*for some $a \geq 1$, $p_{ij} > 0$. For $i \in \mathcal{V} \setminus \mathcal{Q}$, the expected time to reach $\mathcal{Q}$ when starting at $x$ denoted by $m_{\mathcal{Q}}(x)$ satisfies: for all $x \in X_i$,*

$$a^{3 \times 4^r} \frac{\sum_{g \in \mathcal{G}(i \nrightarrow \mathcal{Q})} \pi(g, \overline{p})}{\sum_{g \in \mathcal{G}} \pi(g, \overline{p})} \leq m_{\mathcal{Q}}(x) \leq a^{3 \times 4^r} \frac{\sum_{g \in \mathcal{G}(i \nrightarrow \mathcal{Q})} \pi\left(g, \underline{p}\right)}{\sum_{g \in \mathcal{G}} \pi\left(g, \underline{p}\right)} \tag{D.3}$$

## D.2  Hitting time of the accelerated process

We will instantiate the lemmas of the previous section Appendix D.1 with $\mathcal{V} = \{1, \ldots, N_{\mathrm{crit}}\}$, $\mathcal{Q} = \{K + 1, \ldots, N_{\mathrm{crit}}\}$ and $X_i = \mathcal{V}_i$ for $i \in \{1, \ldots, N_{\mathrm{crit}}\}$. We have $r = N_{\mathrm{crit}} - K$. Note that the notation $\mathcal{Q}$ is in line with the notation of the main text where we denote by $\mathcal{Q}$ both the set of indices $\{K + 1, \ldots, N_{\mathrm{crit}}\}$ and the set of union of the corresponding components $\bigcup_{j \in \mathcal{Q}} \mathcal{C}_j = \bigcup_{j=1}^{N_{\mathrm{targ}}} \mathcal{Q}_j = \arg \min f$.

Define by $\tau_{\mathcal{Q}}^{\eta} := \tau_{\bigcup_{j \in \mathcal{Q}} \mathcal{V}_j}^{\eta}$ the hitting time of $\bigcup_{j \in \mathcal{Q}} \mathcal{V}_j$ for the accelerated SGD process. We also consider the induced chain $(z_n)_{n \geq 0}$ on $\bigcup_{i=1}^{N_{\mathrm{crit}}} \mathcal{V}_i$ defined in Definition 8 as well as its accelerated version defined in Definition 9 and denote by $\widetilde{\tau}_{\mathcal{Q}}^{K}$ the hitting time of $\bigcup_{j \in \mathcal{Q}} \mathcal{V}_j$ for this accelerated induced chain. These two hitting times are related by the following lemma, which is a key consequence of Lemma C.16.

**Lemma D.2.** *For any $\varepsilon > 0$, for any small enough neighborhoods $\mathcal{V}_i$ of $\mathcal{C}_i$, $i = 1, \ldots, N_{\mathrm{crit}}$, there is some $\eta_0 > 0$ such for all $0 < \eta < \eta_0$, for any $x \in \bigcup_{i=1}^{N_{\mathrm{crit}}} \mathcal{V}_i$,*

$$\mathbb{E}_x[\widetilde{\tau}_{\mathcal{Q}}^{K}] \leq \mathbb{E}_x[\tau_{\mathcal{Q}}^{\eta}] \leq \mathbb{E}_x[\widetilde{\tau}_{\mathcal{Q}}^{K}] \times e^{\frac{\varepsilon}{\eta}} . \tag{D.4}$$

*Proof.* First, if $x$ belongs to $\bigcup_{i=K+1}^{N_{\mathrm{crit}}} \mathcal{V}_i = \bigcup_{j \in \mathcal{Q}} \mathcal{V}_j$, then $\tau_{\mathcal{Q}}^{\eta} = 0$ and $\widetilde{\tau}_{\mathcal{Q}}^{K} = 0$ so the statement holds trivially.

We now consider the case where $x$ belongs to $\bigcup_{i=1}^{K} \mathcal{V}_i$. Since $\mathcal{C}_1, \ldots, \mathcal{C}_K$ are compact, we can require that $\bigcup_{i=1}^{K} \mathcal{V}_i$ be relatively compact. We can now apply Corollary C.1 to the accelerated process to obtain that with $\mathcal{X}_0 \leftarrow \operatorname{cl} \bigcup_{i=1}^{K} \mathcal{V}_i$, for any small enough neighborhoods $\mathcal{V}_i$ of $\mathcal{C}_i$, $i = 1, \ldots, N_{\mathrm{crit}}$, there is some $\eta_0 > 0$ such for all $0 < \eta < \eta_0$, for any $x \in \bigcup_{i=1}^{N_{\mathrm{crit}}} \mathcal{V}_i$, we have,

$$1 \leq \mathbb{E}_x[\tau_K^{\eta}] \leq e^{\frac{\varepsilon}{\eta}} . \tag{D.5}$$

We now have that, by Definition 8,

$$
\begin{aligned}
\mathbb{E}_x[\tau_{\mathcal{Q}}^{\eta}] &= \mathbb{E}_x\left[\sum_{n=0}^{\infty} \mathbb{1}\left\{x_{\tau_{nK}^{\eta}}^{\eta} \notin \bigcup_{j \in \mathcal{Q}} \mathcal{V}_j\right\}\left(\tau_{(n+1)K}^{\eta} - \tau_{nK}^{\eta}\right)\right] \\
&= \sum_{n=0}^{\infty} \mathbb{E}_x\left[\mathbb{1}\left\{x_{\tau_{nK}^{\eta}}^{\eta} \notin \bigcup_{j \in \mathcal{Q}} \mathcal{V}_j\right\}\left(\tau_{(n+1)K}^{\eta} - \tau_{nK}^{\eta}\right)\right] \\
&= \sum_{n=0}^{\infty} \mathbb{E}_x\left[\mathbb{1}\left\{x_{\tau_{nK}^{\eta}}^{\eta} \notin \bigcup_{j \in \mathcal{Q}} \mathcal{V}_j\right\} \mathbb{E}_{x_{\tau_{nK}^{\eta}}^{\eta}}\left[\tau_K^{\eta}\right]\right],
\end{aligned} \tag{D.6}
$$

where we used the strong Markov property in the last equality, since $\tau_1^{\eta}$ is always finite almost surely by (D.5).

Combining (D.6) with (D.5), we obtain the bound:

$$
\mathbb{E}_x[\tau_{\mathcal{Q}}^{\eta}] = \sum_{n=0}^{\infty} \mathbb{E}_x\left[\mathbb{1}\left\{x_{\tau_n^{\eta}}^{\eta} \notin \bigcup_{j \in \mathcal{Q}} \mathcal{V}_j\right\}\right] \le \mathbb{E}_x[\tau_{\mathcal{Q}}^{\eta}] \le \sum_{n=0}^{\infty} \mathbb{E}_x\left[\mathbb{1}\left\{x_{\tau_n^{\eta}}^{\eta} \notin \bigcup_{j \in \mathcal{Q}} \mathcal{V}_j\right\}\right] \times e^{\frac{\varepsilon}{\eta}} = \mathbb{E}_x[\widetilde{\tau}_{\mathcal{Q}}^K] \times e^{\frac{\varepsilon}{\eta}}, \tag{D.7}
$$

which yields the result. ∎

**Assumption 9.** For any $i \in \{1, \ldots, K\}$, $j \in \{1, \ldots, N_{\text{crit}}\}$,

$$
B_{i,j} < +\infty. \tag{D.8}
$$

From now on, we will assume that Assumption 9 holds.

**Definition 13.** For $i \in \mathcal{V} \setminus \mathcal{Q}$, we define the following quantities:

$$
E(i \nrightarrow \mathcal{Q}) := \min\left\{\sum_{k \to l \in g} B_{k,l} : g \in \mathcal{G}(i \nrightarrow \mathcal{Q})\right\} \tag{D.9}
$$

$$
E(\mathcal{Q}) := \min\left\{\sum_{k \to l \in g} B_{k,l} : g \in \mathcal{G}(\mathcal{Q})\right\} \tag{D.10}
$$

$$
E(\mathcal{Q} \mid i) := E(\mathcal{Q}) - E(i \nrightarrow \mathcal{Q}). \tag{D.11}
$$

Note that Assumption 9 ensures that all these quantities are finite.

**Lemma D.3.** *For any $i \in \mathcal{V} \setminus \mathcal{Q}$, for any $\varepsilon > 0$, for $\mathcal{V}_1, \ldots, \mathcal{V}_{N_{\text{crit}}}$ neighborhoods of $\mathcal{C}_1, \ldots, \mathcal{C}_{N_{\text{crit}}}$ small enough, there is $\eta_0 > 0$ such that for all $0 < \eta < \eta_0$, for any $x \in \mathcal{V}_i$,*

$$
e^{\frac{E(\mathcal{Q} \mid i) - \varepsilon}{\eta}} \le \mathbb{E}_x\left[\widetilde{\tau}_{\mathcal{Q}}^K\right] \le e^{\frac{E(\mathcal{Q} \mid i) + \varepsilon}{\eta}}. \tag{D.12}
$$

*Proof.* Fix $i \in \mathcal{V} \setminus \mathcal{Q}$. We will apply Lemma D.1 to the accelerated induced chain.

Let us first verify the assumptions of these lemmas. By Lemmas C.6 and C.7, for any $\varepsilon > 0$, for small enough neighborhoods $\mathcal{V}_i$ of $\mathcal{C}_i$, $i = 1, \ldots, N_{\text{crit}}$, there exists $\eta_0 > 0$ such that for all $i \in \{1, \ldots, K\}$, $j \in \{1, \ldots, N_{\text{crit}}\}$, $x \in \mathcal{V}_i$, $0 < \eta < \eta_0$:

$$
\mathbb{Q}^K{}_{\mathcal{V}}(x, \mathcal{V}_j) \le \exp\left(-\frac{B_{i,j}}{\eta} + \frac{\varepsilon}{\eta}\right), \tag{D.13}
$$

$$
\mathbb{Q}^K{}_{\mathcal{V}}(x, \mathcal{V}_j) \ge \exp\left(-\frac{B_{i,j}}{\eta} - \frac{\varepsilon}{\eta}\right). \tag{D.14}
$$

Note that Assumption 9 ensures that both $B_{i,j}$ and $B_{i,j}$ are finite.

We define, for any $i \in \mathcal{V} \setminus \mathcal{Q}$, $j \in \mathcal{V}$ with $i \ne j$:

$$
p_{ij} := \exp\left(-\frac{B_{i,j}}{\eta}\right) \quad \text{and,} \quad p_{ij} := \exp\left(-\frac{B_{i,j}}{\eta}\right). \tag{D.15}
$$

Let us verify the conditions of Lemma D.1 with $a := e^{\varepsilon/\eta}$:

By (D.13) and (D.14), for all $i \in \mathcal{V} \setminus \mathcal{Q}$, $j \in \mathcal{V}$ with $i \neq j$, $x \in \mathcal{V}_i$:

$$a p_{ij} \leq \mathbb{Q}^K{}_\mathcal{V}(x, \mathcal{V}_j) \leq a p_{ij}. \tag{D.16}$$

Now we can apply Lemma D.1 to obtain that for $x \in \mathcal{V}_i$:

$$a^{-3 \times 4^r} e^{\frac{E(\mathcal{Q}|i)}{\eta}} \leq \mathbb{E}_x[\widetilde{\tau}_\mathcal{Q}^K] \leq a^{3 \times 4^r} e^{\frac{E(\mathcal{Q}|i)}{\eta}}. \tag{D.17}$$

Recalling that $a = e^{\varepsilon/\eta}$, the bounds in (D.17) become:

$$e^{\frac{E(\mathcal{Q}|i) - 3 \times 4^r \varepsilon}{\eta}} \leq \mathbb{E}_x[\widetilde{\tau}_\mathcal{Q}^K] \leq e^{\frac{E(\mathcal{Q}|i) + 3 \times 4^r \varepsilon}{\eta}}, \tag{D.18}$$

which concludes the proof since $\varepsilon > 0$ was arbitrary.

$\blacksquare$

**Definition 14.** Let us now define, for $x \in \mathbb{R}^d$,

$$E(\mathcal{Q}|x) = \left[ \max_{i \in \mathcal{V} \setminus \mathcal{Q}} (E(\mathcal{Q}|i) - B_{xi}) \right]_+. \tag{D.19}$$

with the convention that these quantities are zero if $x \in \cup_{j \in \mathcal{Q}} \mathcal{V}_j$ or equal to the quantities defined in Definition 13 if $x \in \cup_{i=1}^K \mathcal{V}_i$.

Note that these quantities defined in Definition 14 are non-negative and finite by Assumption 9.

**Lemma D.4.** *For any $x_0 \in \mathbb{R}^d$, for any $\varepsilon > 0$, for $\mathcal{V}_1, \ldots, \mathcal{V}_{N_{crit}}$ neighborhoods of $\mathcal{C}_1, \ldots, \mathcal{C}_{N_{crit}}$ small enough, there is $\eta_0 > 0$ such that for all $0 < \eta < \eta_0$,*

$$e^{\frac{E(\mathcal{Q}|x_0) - \varepsilon}{\eta}} \leq \mathbb{E}_x[\widetilde{\tau}_\mathcal{Q}^K] \leq e^{\frac{E(\mathcal{Q}|x_0) + \varepsilon}{\eta}}. \tag{D.20}$$

*Proof.* If $x_0 \in \cup_{i=1}^K \mathcal{V}_i$, then Lemma D.3 applies and yields the result. if $x_0 \in \cup_{i=K+1}^{N_{crit}} \mathcal{V}_i$, then the result holds trivially. Let us now consider the general case where $x_0 \in \mathbb{R}^d \setminus \cup_{i=1}^{N_{crit}} \mathcal{V}_i$ and let us first prove the upper bound, the lower bound follows similarly.

By Lemma D.3 and Lemma C.13: for small enough neighborhoods $\mathcal{V}_i$ of $\mathcal{C}_i$, $i = 1, \ldots, N_{crit}$, there exists $\eta_0 > 0$ such that for all $0 < \eta < \eta_0$, $i \in \{1, \ldots, K\}$, $x' \in \mathcal{V}_i$, both

$$\mathbb{E}_{x'}[\widetilde{\tau}_\mathcal{Q}^K] \leq \exp\left( \frac{E(\mathcal{Q}|i) + \varepsilon}{\eta} \right) \tag{D.21}$$

and,

$$\mathbb{Q}^K{}_\mathcal{V}(x_0, \mathcal{V}_i) \leq \exp\left( -\frac{B_{x_0,i}}{\eta} + \frac{\varepsilon}{\eta} \right) \tag{D.22}$$

hold.

Using the strong Markov property at time $\tau_K^\eta$ (which is finite almost surely by Corollary C.1), we have:

$$\mathbb{E}_{x_0}[\widetilde{\tau}_\mathcal{Q}^K] = \sum_{i=1}^{N_{crit}} \mathbb{E}_{x_0}\left[ (1 + \mathbb{E}_{x_{\tau_K^\eta}^\eta}[\widetilde{\tau}_\mathcal{Q}^K]) \mathbb{1}\left\{ x_{\tau_K^\eta}^\eta \in \mathcal{V}_i \right\} \right]$$

$$= 1 + \sum_{i=1}^{K} \mathbb{E}_{x_0}\left[ \mathbb{E}_{x_{\tau_K^\eta}^\eta}[\widetilde{\tau}_\mathcal{Q}^K] \mathbb{1}\left\{ x_{\tau_K^\eta}^\eta \in \mathcal{V}_i \right\} \right] + \sum_{i=K+1}^{N_{crit}} 0 \times \mathbb{1}\left\{ x_{\tau_K^\eta}^\eta \in \mathcal{V}_i \right\}$$

$$\leq 1 + \sum_{i=1}^{K} \exp\left( \frac{E(\mathcal{Q}|i) - B_{x_0,i} + 2\varepsilon}{\eta} \right), \tag{D.23}$$

where we used (D.21) and (D.22) in the last inequality. Bounding the sum as follows

$$1 + \sum_{i=1}^{K} \exp\left(\frac{E(\mathcal{Q}\,|\,i) - B_{x_0,i} + 2\varepsilon}{\eta}\right) \leq (K+1) \exp\left(\frac{0 \vee \max_{i \in \{1,\dots,K\}} \left(E(\mathcal{Q}\,|\,i) - B_{x_0,i} + 2\varepsilon\right)}{\eta}\right)$$

$$= (K+1) \exp\left(\frac{E(\mathcal{Q}\,|\,x_0) + 2\varepsilon}{\eta}\right), \tag{D.24}$$

yields the upper bound.

The lower bound follows similarly using Lemma C.14 and the lower bound from Lemma D.3.

■

**Theorem D.1.** *For any $x_0 \in \mathbb{R}^d$, for any $\varepsilon > 0$, for $\mathcal{V}_1, \dots, \mathcal{V}_{N_{crit}}$ neighborhoods of $\mathcal{C}_1, \dots, \mathcal{C}_{N_{crit}}$ small enough, there is $\eta_0 > 0$ such that for all $0 < \eta < \eta_0$,*

$$e^{\frac{E(\mathcal{Q}\,|\,x_0) - \varepsilon}{\eta}} \leq \mathbb{E}_{x_0}\left[\tau_{\mathcal{Q}}^{\eta}\right] \leq e^{\frac{E(\mathcal{Q}\,|\,x_0) + \varepsilon}{\eta}}. \tag{D.25}$$

*Proof.* The result follows directly by combining Lemmas D.2 and D.4.

Specifically, by Lemma D.4, for any $\varepsilon > 0$, for small enough neighborhoods $\mathcal{V}_i$ of $\mathcal{C}_i$, $i = 1, \dots, N_{crit}$, there exists $\eta_0 > 0$ such that for all $0 < \eta < \eta_0$:

$$e^{\frac{E(\mathcal{Q}\,|\,x_0) + \varepsilon}{\eta}} \leq \mathbb{E}_{x_0}\left[\widetilde{\tau}_{\mathcal{Q}}^{K}\right] \leq e^{\frac{E(\mathcal{Q}\,|\,x_0) + \varepsilon}{\eta}}. \tag{D.26}$$

Then by Lemma D.2, potentially reducing $\eta_0$, we have:

$$\mathbb{E}_{x_0}[\widetilde{\tau}_{\mathcal{Q}}^{K}] \leq \mathbb{E}_{x_0}[\tau_{\mathcal{Q}}^{\eta}] \leq \mathbb{E}_{x_0}[\widetilde{\tau}_{\mathcal{Q}}^{K}] \times e^{\frac{\varepsilon}{\eta}}. \tag{D.27}$$

Combining these inequalities and using the fact that $\varepsilon > 0$ was arbitrary concludes the proof. ■

The following lemma, though looking at first weaker than Theorem D.1, will be useful later due to its uniformity in the initial condition in neighborhoods of the components.

**Lemma D.5.** *Under Assumption 9, for any $i \in \mathcal{V} \setminus \mathcal{Q}$, for any $\varepsilon > 0$, for $\mathcal{V}_1, \dots, \mathcal{V}_{N_{crit}}$ neighborhoods of $\mathcal{C}_1, \dots, \mathcal{C}_{N_{crit}}$ small enough, there is $\eta_0 > 0$ such that for all $0 < \eta < \eta_0$, for any $x \in \mathcal{V}_i$,*

$$e^{\frac{E(\mathcal{Q}\,|\,i) - \varepsilon}{\eta}} \leq \mathbb{E}_x\left[\tau_{\mathcal{Q}}^{\eta}\right] \leq e^{\frac{E(\mathcal{Q}\,|\,i) + \varepsilon}{\eta}}. \tag{D.28}$$

*Proof.* The result follows directly by combining Lemmas D.2 and D.3.

Specifically, by Lemma D.3, for any $\varepsilon > 0$, for small enough neighborhoods $\mathcal{V}_i$ of $\mathcal{C}_i$, $i = 1, \dots, N_{crit}$, there exists $\eta_0 > 0$ such that for all $0 < \eta < \eta_0$, for any $x \in \mathcal{V}_i$:

$$e^{\frac{E(\mathcal{Q}\,|\,i) + \varepsilon}{\eta}} \leq \mathbb{E}_x\left[\widetilde{\tau}_{\mathcal{Q}}^{K}\right] \leq e^{\frac{E(\mathcal{Q}\,|\,i) + \varepsilon}{\eta}}. \tag{D.29}$$

Then by Lemma D.2, potentially reducing $\eta_0$, we have:

$$\mathbb{E}_x[\widetilde{\tau}_{\mathcal{Q}}^{K}] \leq \mathbb{E}_x[\tau_{\mathcal{Q}}^{\eta}] \leq \mathbb{E}_x[\widetilde{\tau}_{\mathcal{Q}}^{K}] \times e^{\frac{\varepsilon}{\eta}}. \tag{D.30}$$

Combining these inequalities and using the fact that $\varepsilon > 0$ was arbitrary concludes the proof. ■

Another technical lemma that will be useful later is the following.

**Lemma D.6.** *For any $x_0 \in \mathbb{R}^d$, for any $\varepsilon > 0$, for $\mathcal{V}_1, \dots, \mathcal{V}_{N_{crit}}$ neighborhoods of $\mathcal{C}_1, \dots, \mathcal{C}_{N_{crit}}$ small enough, there is $\eta_0 > 0$ such that for all $0 < \eta < \eta_0$,*

$$e^{\frac{E(\mathcal{Q}\,|\,x_0) - \varepsilon}{\eta}} \leq \mathbb{E}_{x_0}[\widetilde{\tau}_{\mathcal{Q}}] \tag{D.31}$$

*Proof.* This result follows directly from Lemma D.4 and the fact that $\widetilde{\tau}_{\mathcal{Q}} \leq \widetilde{\tau}_{\mathcal{Q}}^{K}$. ■

### D.3 Hitting time of SGD

Leveraging the bounds on the hitting time of the accelerated process, we can now derive bounds on the hitting time of SGD. Denote by $\tau$ the hitting time of $\mathcal{V} = \bigcup_{j \in \mathcal{Q}} \mathcal{V}_j$ for the SGD sequence:

$$\tau := \inf\left\{ n \geq 0 : x_n \in \bigcup_{j \in \mathcal{Q}} \mathcal{V}_j \right\}. \tag{D.32}$$

It corresponds to the hitting time of SGD defined in the main text (10) in the simpler case where each $\mathcal{V}_j$ is of the form $\mathcal{V}_j = \mathcal{U}_\delta(\mathcal{C}_j)$ for $j \in \mathcal{Q}$. We now state our first main results on the actual SGD sequence (B.12). The theorem below is a corollary of Theorem D.1 which concerned the subsampled process (B.16).

**Theorem D.2.** *For any $x_0 \in \mathbb{R}^d$, for any $\varepsilon > 0$, for $\mathcal{V}_1, \ldots, \mathcal{V}_{N_{crit}}$ neighborhoods of $\mathcal{C}_1, \ldots, \mathcal{C}_{N_{crit}}$ small enough, there is $\eta_0 > 0$ such that for all $0 < \eta < \eta_0$,*

$$\mathbb{E}_{x_0}[\tau] \leq e^{\frac{E(\mathcal{Q}\,|\,x_0)+\varepsilon}{\eta}} . \tag{D.33}$$

*Proof.* This result follows from Theorem D.1 and the fact that, by construction,

$$\tau \leq \tau_{\mathcal{Q}}^\eta \lceil \eta^{-1} \rceil, \tag{D.34}$$

with $\lceil \eta^{-1} \rceil \leq e^{\frac{\varepsilon}{\eta}}$ for small enough $\eta$. ∎

**Corollary D.1.** *For any $x_0 \in \mathbb{R}^d$, for any $\Delta > 0$, for $\mathcal{V}_1, \ldots, \mathcal{V}_{N_{crit}}$ neighborhoods of $\mathcal{C}_1, \ldots, \mathcal{C}_{N_{crit}}$ small enough, there is $\eta_0 > 0$ such that for all $0 < \eta < \eta_0$,*

$$\mathbb{P}_x\left( \tau < \exp\left( \frac{E(\mathcal{Q}\,|\,x_0) + \Delta}{\eta} \right) \right) \leq e^{-\frac{\Delta}{2\eta}} . \tag{D.35}$$

*Proof.* By Markov's inequality and Theorem D.2, for any $x_0 \in \mathbb{R}^d$, for $\mathcal{V}_1, \ldots, \mathcal{V}_{N_{crit}}$ neighborhoods of $\mathcal{C}_1, \ldots, \mathcal{C}_{N_{crit}}$ small enough, there exists $\eta_0 > 0$ such that for all $0 < \eta < \eta_0$:

$$\mathbb{P}_x\left( \tau \geq \exp\left( \frac{E(\mathcal{Q}\,|\,x_0) + \Delta}{\eta} \right) \right) \leq \frac{\mathbb{E}_x[\tau]}{\exp\left( \frac{E(\mathcal{Q}\,|\,x_0)+\Delta}{\eta} \right)} \tag{D.36}$$

$$\leq \frac{\exp\left( \frac{E(\mathcal{Q}\,|\,x_0)+\Delta/2}{\eta} \right)}{\exp\left( \frac{E(\mathcal{Q}\,|\,x_0)+\Delta}{\eta} \right)} = \exp(-\Delta/2\eta) . \tag{D.37}$$

∎

Let us now focus on the lower-bound which requires more care. We will require the following additional assumption:

**Assumption 10.** Assume that there exist $\mu > 0$, $\overline{\sigma}_\infty^2 < +\infty$, $R > 0$, such that,

1. $\mathrm{cl}\,\mathcal{U}_R(\mathcal{Q}_j)$, $j = 1, \ldots, N_{targ}$ are pairwise disjoint.

2. for all $j = 1, \ldots, N_{targ}$, $x \in \mathrm{cl}\,\mathcal{U}_R(\mathcal{Q}_j)$, $u(x, \omega)$ is a $\overline{\sigma}_\infty^2$-sub-Gaussian:

$$\forall p \in \mathbb{R}^d, \quad \log \mathbb{E}\left[ e^{\langle p, u(x,\omega) \rangle} \right] \leq \frac{\overline{\sigma}_\infty^2}{2} \|p\|^2 . \tag{D.38}$$

3. for all $j = 1, \ldots, N_{targ}$, $x \in \mathrm{cl}\,\mathcal{U}_R(\mathcal{Q}_j)$, there exists $x' \in \mathrm{proj}_{\mathcal{Q}_j}(x)$ a projection of $x$ on $\mathcal{Q}_j$ such that:

$$\langle \nabla f(x), x - x' \rangle \geq \frac{\mu}{2} \|x - x'\|^2 \tag{D.39}$$

Note that, by Assumption 2 and compactness of $\mathrm{crit}(f)$, such a $\overline{\sigma}_{\infty}^2$ always exists since one can take

$$\overline{\sigma}_{\infty}^2 = \sup\left\{\sigma_{\infty}^2(f(x)) : x \in \bigcup_{j=1}^{N_{\mathrm{targ}}} \mathrm{cl}\,\mathcal{U}_R(\mathcal{Q}_j)\right\}. \tag{D.40}$$

In particular, if $\sigma_{\infty}^2$ is constant, then one can simply take $\overline{\sigma}_{\infty}^2 = \sigma_{\infty}^2$.

**Lemma D.7.** *Under Assumption 10, for any $j \in \mathcal{Q}$, there exists $\eta_0 > 0$ such that for all $0 < \eta < \eta_0$, for any $x_0 \in \mathbb{R}^d$ satisfying $d(x_0, \mathcal{Q}_j) \leq \frac{R}{12}$, we have, for SGD started at $x_0$:*

$$\mathbb{P}_{x_0}\big(\forall n \geq 0, \ d(x_n, \mathcal{Q}_j) \leq R\big) \geq 1 - \exp\left(-\frac{\mu R^2}{1152 \overline{\sigma}_{\infty}^2 \eta}\right). \tag{D.41}$$

*Proof.* Let us denote by $\mathcal{K} \coloneqq \mathrm{cl}\,\mathcal{U}_R(\mathcal{Q}_j)$ Given $x_0 \in \mathcal{K}$, we define the projected SGD sequence as

$$\begin{cases} p_0 = x_0 \\ p_{n+1} \in \mathrm{proj}_{\mathcal{K}}(p_n - \eta \nabla f(p_n) + \eta u_n), & \text{where } u_n = u(p_n, \omega_n). \end{cases} \tag{D.42}$$

By Assumption 10, for any $n \geq 0$, there exists $y_n \in \mathrm{proj}_{\mathcal{Q}_j}(p_n)$ such that:

$$\langle \nabla f(p_n), p_n - y_n \rangle \geq \frac{\mu}{2} \|p_n - y_n\|^2. \tag{D.43}$$

Define the martingale sequence $(M_n)_{n \geq 0}$ by:

$$M_n \coloneqq \sum_{k=0}^{n} (1 - \mu\eta)^{n-k} \langle u_k, p_k - y_k \rangle. \tag{D.44}$$

We recursively compute the moment generating function of $M_n$: for $\alpha \geq 0$, $n \geq 0$, by the tower property of conditional expectation and Assumption 10:

$$\begin{aligned} \mathbb{E}\big[e^{\alpha M_n}\big] &\leq \exp\left(\frac{\alpha^2 R^2 \overline{\sigma}_{\infty}^2}{2}\right) \exp\left(\frac{(1-\mu\eta)^2 \alpha^2 \overline{\sigma}_{\infty}^2}{2}\right) \ldots \exp\left(\frac{(1-\mu\eta)^{2n} \alpha^2 \overline{\sigma}_{\infty}^2}{2}\right) \\ &= \exp\left(\frac{\alpha^2 R^2 \overline{\sigma}_{\infty}^2}{2} \times \frac{1 - (1-\mu\eta)^{2(n+1)}}{1 - (1-\mu\eta)^2}\right) \\ &\leq \exp\left(\frac{\alpha^2 R^2 \overline{\sigma}_{\infty}^2}{2\mu\eta}\right), \end{aligned} \tag{D.45}$$

provided that $\eta$ is small enough so that $\mu\eta < 1$. Note that we used that the iterates of (D.42) are at distance at most $R$ from $\mathcal{Q}_j$ by construction.

Since $\big(e^{\alpha M_n}\big)_{n \geq 0}$ is a convex function of the martingale sequence $(M_n)_{n \geq 0}$, it is a sub-martingale. We can apply Doob's maximal inequality to the non-negative sub-martingale $(M_n)_{n \geq 0}$ to obtain that for any $\Delta > 0$, $N \geq 0$:

$$\begin{aligned} \mathbb{P}\left(\sup_{n \leq N-1} M_n \geq \Delta\right) &\leq \mathbb{P}\left(\sup_{n \leq N-1} e^{\alpha M_n} \geq e^{\alpha\Delta}\right) \\ &\leq \exp\left(\frac{\alpha^2 R^2 \overline{\sigma}_{\infty}^2}{2\mu\eta} - \alpha\Delta\right), \end{aligned} \tag{D.46}$$

where we used (D.45) in the last inequality. Optimizing the right-hand side of (D.46) with respect to $\alpha$ and setting $\alpha = \frac{\mu\eta\Delta}{R^2 \overline{\sigma}_{\infty}^2}$, we obtain:

$$\mathbb{P}\left(\sup_{n \leq N-1} M_n \geq \Delta\right) \leq \exp\left(-\frac{\mu\eta\Delta^2}{2R^2 \overline{\sigma}_{\infty}^2}\right). \tag{D.47}$$

Taking $\Delta \leftarrow \Delta/\eta$ and $N \to +\infty$, we obtain that, by monotone continuity of probability measures, for any $\Delta > 0$:

$$\mathbb{P}\left(\sup_{n \geq 0} \eta M_n \geq \Delta\right) \leq \exp\left(-\frac{\mu\Delta^2}{2\eta R^2 \overline{\sigma}_\infty^2}\right). \tag{D.48}$$

Denote by $G := \sup_{x \in \mathcal{K}, \omega \in \Omega} \|-\nabla f(x) + u(x, \omega)\|$ the bound on the gradient and on the noise on $\mathcal{K}$, which is finite by Assumptions 5 and 7.

Let us now derive a recursive inequality for the distance to $\mathcal{Q}_j$. For any iterate $n \geq 0$, by non-expansiveness of the projection, we have:

$$\begin{aligned}
d^2(p_{n+1}, \mathcal{Q}_j) &\leq \|p_{n+1} - y_n\|^2 \\
&\leq \|p_n - \eta\nabla f(p_n) + \eta u_n - y_n\|^2 \\
&= \|p_n - y_n\|^2 + \eta^2\|\nabla f(p_n) - u_n\|^2 \\
&\quad - 2\eta\langle\nabla f(p_n), p_n - y_n\rangle + 2\eta\langle u_n, p_n - y_n\rangle \\
&\leq (1 - \mu\eta)\|p_n - y_n\|^2 + \eta^2 G^2 + 2\eta\langle u_n, p_n - y_n\rangle,
\end{aligned} \tag{D.49}$$

where we used (D.43) in the last inequality.

Iterating this inequality and using that $d(p_n, \mathcal{Q}_j) = \|p_n - y_n\|$, we obtain:

$$\begin{aligned}
d^2(p_n, \mathcal{Q}_j) &\leq (1 - \mu\eta)^n d^2(x_0, \mathcal{Q}_j) + \eta^2 G^2 \sum_{i=0}^{n-1}(1 - \mu\eta)^i \\
&\quad + 2\eta \sum_{i=0}^{n-1}(1 - \mu\eta)^{n-1-i}\langle u_i, p_i - y_i\rangle \\
&= (1 - \mu\eta)^n d^2(x_0, \mathcal{Q}_j) + \eta^2 G^2 \frac{1 - (1 - \mu\eta)^n}{\mu\eta} + 2\eta M_{n-1} \\
&\leq (1 - \mu\eta)^n d^2(x_0, \mathcal{Q}_j) + \frac{2\eta G^2}{\mu} + 2\eta M_{n-1}.
\end{aligned} \tag{D.50}$$

Using that $d(x_0, \mathcal{Q}_j) \leq \frac{R}{12}$ by assumption, requiring that $\eta > 0$ be small enough so that $\frac{2\eta G^2}{\mu} \leq \frac{R^2}{12}$ and taking $\Delta \leftarrow \frac{R^2}{24}$ in (D.48), we obtain that for any $n \geq 0$, with probability at least $1 - \exp\left(-\frac{\mu R^2}{1152 \overline{\sigma}_\infty^2 \eta}\right)$:

$$d^2(p_n, \mathcal{Q}_j) \leq \frac{R^2}{4}. \tag{D.51}$$

In addition, take $\eta > 0$ small enough so that $\frac{R}{2} + \eta G \leq \frac{R}{2}$: this implies that if $x \in \mathbb{R}^d$ so that $d(x, \mathcal{Q}_j) \leq \frac{R}{2}$, then $x - \eta\nabla f(x) + \eta u(x, \omega) \in \mathcal{K}$. Combining this remark with (D.51), we can show recursively that the sequences $(x_n)_{n \geq 0}$ and $(p_n)_{n \geq 0}$ coincide with probability at least $1 - \exp\left(-\frac{\mu R^2}{1152 \overline{\sigma}_\infty^2 \eta}\right)$, yielding the desired result.

∎

Define by $\widetilde{\tau}_{\mathcal{Q}}$ the hitting time of $\bigcup_{j \in \mathcal{Q}} \mathcal{V}_j$ by the induced chain $(z_n)_{n \geq 0}$ (see Definition 8).

**Lemma D.8.** *Define*

$$\mathcal{B}_\infty := \max_{i \in \mathcal{V}\setminus\mathcal{Q}} \min_{j \in \mathcal{Q}} \mathcal{B}_{i,j} + 1. \tag{D.52}$$

*For any $x_0 \in \mathbb{R}^d$, for $\mathcal{V}_1, \ldots, \mathcal{V}_{N_{crit}}$ neighborhoods of $\mathcal{C}_1, \ldots, \mathcal{C}_{N_{crit}}$ small enough, there is $\eta_0 > 0$ such that for all $0 < \eta < \eta_0$, for any $n \geq 0$,*

$$\mathbb{P}_{x_0}(\widetilde{\tau}_{\mathcal{Q}} > n) \leq \left(1 - e^{-\frac{\mathcal{B}_\infty}{\eta}}\right)^n. \tag{D.53}$$

Note that Assumption 9 ensures that $\mathcal{B}_\infty$ is finite.

*Proof.* Fix some $x_0 \in \mathbb{R}^d$. By definition of $\mathcal{B}_\infty$, for any vertex $i \in \mathcal{V} \setminus \mathcal{Q}$, there exists $j \in \mathcal{Q}$ such that:

$$\mathcal{B}_{i,j} \leq \mathcal{B}_\infty - 1 \,. \tag{D.54}$$

Fix $\varepsilon \leftarrow 1$. By [Lemma C.4](), for small enough neighborhoods $\mathcal{V}_i$ of $\mathcal{C}_i$, $i = 1, \ldots, N_{\text{crit}}$, there exists $\eta_0 > 0$ such that for all $i \in \mathcal{V} \setminus \mathcal{Q}, x \in \mathcal{V}_i, 0 < \eta < \eta_0$:

$$\begin{aligned}
\mathbb{Q}_\mathcal{V}\left(x, \bigcup_{j \in \mathcal{Q}} \mathcal{V}_j\right) &\geq \max_{j \in \mathcal{Q}} \mathbb{Q}_\mathcal{V}(x, \mathcal{V}_j) \\
&\geq \exp\left(-\frac{\min_{j \in \mathcal{Q}} \mathcal{B}_{i,j}}{\eta} - \frac{\varepsilon}{\eta}\right) \\
&\geq \exp\left(-\frac{\mathcal{B}_\infty}{\eta}\right).
\end{aligned} \tag{D.55}$$

Therefore, for any $n \geq 1$, by the strong Markov property, we have:

$$\begin{aligned}
\mathbb{P}_{x_0}(\widetilde{\tau}_\mathcal{Q} > n) &= \mathbb{E}_{x_0}\left[\mathbb{1}\{\widetilde{\tau}_\mathcal{Q} > n - 1\} \mathbb{1}\{z_n \notin \bigcup_{j \in \mathcal{Q}} \mathcal{V}_j\}\right] \\
&= \mathbb{E}_{x_0}\left[\mathbb{1}\{\widetilde{\tau}_\mathcal{Q} > n - 1\} \mathbb{E}_{z_{n-1}}\left[\mathbb{1}\{z_n \notin \bigcup_{j \in \mathcal{Q}} \mathcal{V}_j\}\right]\right] \\
&\leq \left(1 - \exp\left(-\frac{\mathcal{B}_\infty}{\eta}\right)\right) \mathbb{P}_{x_0}(\widetilde{\tau}_\mathcal{Q} > n - 1) \,,
\end{aligned} \tag{D.56}$$

where we used the [(D.55)]() in the last inequality. Iterating [(D.56)](), we obtain the desired result.

$\blacksquare$

**Theorem D.3.** *Fix $x_0 \in \mathbb{R}^d$. Under [Assumption 10]() and assuming that,*

$$\frac{\mu R^2}{1152 \overline{\sigma}_\infty^2} > \mathcal{B}_\infty - E(\mathcal{Q} \,|\, x_0) \,, \tag{D.57}$$

*for any $\varepsilon > 0$, for $\mathcal{V}_1, \ldots, \mathcal{V}_{N_{\text{crit}}}$ neighborhoods of $\mathcal{C}_1, \ldots, \mathcal{C}_{N_{\text{crit}}}$ small enough, there is $\eta_0 > 0$ such that for all $0 < \eta < \eta_0$:*

$$\mathbb{E}_{x_0}[\tau] \geq e^{\frac{E(\mathcal{Q} \,|\, x_0) - \varepsilon}{\eta}} \,. \tag{D.58}$$

*Remark* D.1. Note that the condition [(D.57)]() is implied by the stronger condition

$$\frac{\mu R^2}{1152 \overline{\sigma}_\infty^2} > \mathcal{B}_\infty \,. \tag{D.59}$$

In particular, $\mathcal{B}_\infty$ does not depend on $f$ nor on the noise distribution on the interior of the basins of attraction of $\mathcal{Q}_1, \ldots, \mathcal{Q}_{N_{\text{targ}}}$. Indeed, by definition, $\mathcal{B}_\infty$ is equal to

$$\mathcal{B}_\infty = \max_{i \in \mathcal{V} \setminus \mathcal{Q}} \min_{j \in \mathcal{Q}} \mathcal{B}_{i,j} + 1 \,, \tag{D.60}$$

and, for $i \in \mathcal{V} \setminus \mathcal{Q}, j \in \mathcal{Q}$, thanks to [Lemma B.6](), we have that

$$\mathcal{B}_{i,j} = \inf\left\{\mathcal{S}_T(\gamma) : \gamma \in \mathcal{C}([0,T]) \, \gamma_0 \in \mathcal{C}_i, \, \gamma_T \in \text{Attr}(\mathcal{C}_j), \, \gamma_n \notin \bigcup_{k \in \mathcal{V}} \mathcal{C}_k \, \forall n \in 1, \ldots, T - 1\right\}, \tag{D.61}$$

since for any $x \in \text{Attr}(\mathcal{C}_j)$, the gradient flow started at $x$ converges to $\mathcal{C}_j$ and has zero action cost ([Lemma B.4]()).

Hence, this condition assumption will be satisfied for functions with a sufficiently sharp profile near $\mathcal{Q}_1, \ldots, \mathcal{Q}_{N_{\text{targ}}}$ and a given profile away from it.

*Proof.* Let us first begin by invoking Lemma D.6: for $\mathcal{V}_1, \ldots, \mathcal{V}_{N_{\text{crit}}}$ neighborhoods of $\mathcal{C}_1, \ldots, \mathcal{C}_{N_{\text{crit}}}$ small enough, there is $\eta_0 > 0$ such that for all $0 < \eta < \eta_0$,

$$e^{\frac{E(\mathcal{Q}\,|\,x_0) - \varepsilon}{\eta}} \leq \mathbb{E}_{x_0}[\widetilde{\tau}_{\mathcal{Q}}]. \tag{D.62}$$

Also require that, for all $j \in \mathcal{Q}$, $\mathcal{V}_j \subset \left\{x \in \mathbb{R}^d : d\left(x, \mathcal{Q}_j\right) \leq \frac{R}{12}\right\}$ with $R$ as in Assumption 10.

If $x_0 \in \bigcup_{j \in \mathcal{Q}} \mathcal{V}_j$, then the result is trivial so let us assume that $x_0 \notin \bigcup_{j \in \mathcal{Q}} \mathcal{V}_j$.

Let us denote by

$$H := \left\{\forall n \geq 0, \, d(x_{n+\tau}, \mathcal{Q}_j) \leq R \text{ with } x_\tau \in \mathcal{V}_j\right\} \tag{D.63}$$

the event that, after SGD hits $\bigcup_{k \in \mathcal{Q}} \mathcal{V}_k$ at $\mathcal{V}_j$, it stays in $\mathcal{V}_j$ forever. Note that Theorem D.2 ensures that $\tau$ is finite almost surely. By the strong Markov property, we thus obtain from Lemma D.7 that

$$\mathbb{P}_{x_0}(H) \geq 1 - \exp\left(-\frac{\mu R^2}{1152 \overline{\sigma}_\infty^2 \eta}\right). \tag{D.64}$$

Let us now start from $\mathbb{E}_{x_0}[\widetilde{\tau}_{\mathcal{Q}}]$ where $\widetilde{\tau}_{\mathcal{Q}}$ is the hitting time of $\bigcup_{j \in \mathcal{Q}} \mathcal{V}_j$ by the induced chain $(z_n)_{n \geq 0}$ (see Definition 8), for which we have a lower-bound from Lemma D.6. We write

$$\mathbb{E}_{x_0}[\widetilde{\tau}_{\mathcal{Q}}] = \mathbb{E}_{x_0}[\widetilde{\tau}_{\mathcal{Q}} \mathbb{1}_H] + \mathbb{E}_{x_0}[\widetilde{\tau}_{\mathcal{Q}} \mathbb{1}_{H^C}]. \tag{D.65}$$

We begin with the first term of the RHS of (D.65). With the notations from Definition 8, let us consider $k \geq 0$ such that $\tau_k^\eta \lfloor \eta^{-1} \rfloor < \tau \leq \tau_{k+1}^\eta \lfloor \eta^{-1} \rfloor$ (which is is possible since these hitting times are finite almost surely and $\tau > 0$). But, on the event $H$, $z_{k+1} = x_{\tau_{k+1}^\eta}^\eta = x_{\tau_{k+1}^\eta \lfloor \eta^{-1} \rfloor}$ must be in $\mathcal{V}_j$. This means that, on $H$, $\widetilde{\tau}_{\mathcal{Q}} = k+1$. Moreover, $\tau \leq k \lfloor \eta^{-1} \rfloor$ so that, on $H$

$$\widetilde{\tau}_{\mathcal{Q}} \leq \frac{\tau}{\lfloor \eta^{-1} \rfloor} + 1. \tag{D.66}$$

and,

$$\mathbb{E}_{x_0}[\widetilde{\tau}_{\mathcal{Q}} \mathbb{1}_H] \leq \mathbb{E}_{x_0}\left[\frac{\tau}{\lfloor \eta^{-1} \rfloor} + 1\right]. \tag{D.67}$$

Let us now focus on the second term of the RHS of (D.65). By Fubini's theorem for non-negative integrands, we can rewrite the expectation as:

$$\mathbb{E}_{x_0}[\widetilde{\tau}_{\mathcal{Q}} \mathbb{1}_{H^C}] = \mathbb{E}_{x_0}\left[\int_0^{+\infty} \mathbb{1}\{\widetilde{\tau}_{\mathcal{Q}} > t\} \mathbb{1}_{H^C} \, dt\right]$$

$$= \int_0^{+\infty} \mathbb{P}_{x_0}\left(\widetilde{\tau}_{\mathcal{Q}} > t, H^C\right) dt. \tag{D.68}$$

Let us now split the integral into two parts. For $T > 0$, we have:

$$\mathbb{E}_{x_0}[\widetilde{\tau}_{\mathcal{Q}} \mathbb{1}_{H^C}] = \int_0^T \mathbb{P}_{x_0}\left(\widetilde{\tau}_{\mathcal{Q}} > t, H^C\right) dt + \int_T^{+\infty} \mathbb{P}_{x_0}\left(\widetilde{\tau}_{\mathcal{Q}} > t, H^C\right) dt$$

$$\leq \int_0^T \mathbb{P}_{x_0}\left(H^C\right) dt + \int_T^{+\infty} \mathbb{P}_{x_0}(\widetilde{\tau}_{\mathcal{Q}} > t) dt. \tag{D.69}$$

By (D.64), the first term of the RHS of (D.69) is upper-bounded as:

$$\int_0^T \mathbb{P}_{x_0}\left(H^C\right) dt \leq T \exp\left(-\frac{\mu R^2}{1152 \overline{\sigma}_\infty^2 \eta}\right). \tag{D.70}$$

By (D.57), take $\varepsilon > 0$ small enough so that

$$\frac{\mu R^2}{1152 \overline{\sigma}_\infty^2} \geq \mathcal{B}_\infty - E(\mathcal{Q}\,|\,x_0) + 3\varepsilon, \tag{D.71}$$

and define

$$T := \exp\left(\frac{\mu R^2}{1152\overline{\sigma}_\infty^2 \eta} + \frac{E(\mathcal{Q}\,|\,x_0) - 2\varepsilon}{\eta}\right). \tag{D.72}$$

(D.70) now becomes:

$$\int_0^T \mathbb{P}_{x_0}\left(H^C\right)dt \le e^{\frac{E(\mathcal{Q}\,|\,x_0) - 2\varepsilon}{\eta}} \tag{D.73}$$

By Lemma D.8, the second term of the RHS of (D.69) can be upper-bounded as:

$$
\begin{aligned}
\int_T^{+\infty} \mathbb{P}_{x_0}(\widetilde{\tau}_\mathcal{Q} > t)dt &\le \int_T^{+\infty}\left(1 - e^{-\frac{\mathcal{B}_\infty}{\eta}}\right)^t dt \\
&\le \int_T^{+\infty} \exp\left(e^{-\frac{\mathcal{B}_\infty}{\eta}}t\right)dt \\
&= e^{\frac{\mathcal{B}_\infty}{\eta}}\exp\left(-\frac{\mathcal{B}_\infty}{\eta}T\right),
\end{aligned}
\tag{D.74}
$$

where we used $1 - x \le e^{-x}$ for any $x \in \mathbb{R}$ in the last inequality. Injecting the definition of $T$ (D.72), this bound becomes

$$
\begin{aligned}
\int_T^{+\infty} \mathbb{P}_{x_0}(\widetilde{\tau}_\mathcal{Q} > t)dt &\le \exp\left(\frac{\mathcal{B}_\infty}{\eta} - \exp\left(\frac{\mu R^2}{1152\overline{\sigma}_\infty^2 \eta} + \frac{E(\mathcal{Q}\,|\,x_0) - \mathcal{B}_\infty - 2\varepsilon}{\eta}\right)\right) \\
&\le \exp\left(\frac{\mathcal{B}_\infty}{\eta} - \exp\left(\frac{\varepsilon}{\eta}\right)\right),
\end{aligned}
\tag{D.75}
$$

where we used (D.71) in the last inequality. With $\eta > 0$ small enough, we finally obtain

$$\int_T^{+\infty} \mathbb{P}_{x_0}(\widetilde{\tau}_\mathcal{Q} > t)dt \le 1. \tag{D.76}$$

Combining (D.73) and (D.76) in (D.69), we obtain:

$$\mathbb{E}_{x_0}[\widetilde{\tau}_\mathcal{Q}\, \mathbb{1}_{H^C}] \le e^{\frac{E(\mathcal{Q}\,|\,x_0) - 2\varepsilon}{\eta}} + 1, \tag{D.77}$$

and, combining (D.67) and (D.77) above in (D.65), we get:

$$\mathbb{E}_{x_0}[\widetilde{\tau}_\mathcal{Q}] \le \mathbb{E}_{x_0}\left[\frac{\tau}{\lfloor \eta^{-1} \rfloor} + 1\right] + e^{\frac{E(\mathcal{Q}\,|\,x_0) - 2\varepsilon}{\eta}} + 1. \tag{D.78}$$

Plugging in (D.62) yields:

$$e^{\frac{E(\mathcal{Q}\,|\,x_0) - \varepsilon}{\eta}} \le \mathbb{E}_{x_0}\left[\frac{\tau}{\lfloor \eta^{-1} \rfloor} + 1\right] + e^{\frac{E(\mathcal{Q}\,|\,x_0) - 2\varepsilon}{\eta}} + 1, \tag{D.79}$$

which yields the desired result by taking $\eta > 0$ small enough.

∎

Let us state a variant of Theorem D.3 that will be useful later.

**Lemma D.9.** *Fix $i \in \{1, \dots, N_{crit}\}$. Under Assumption 10 and assuming that,*

$$\frac{\mu R^2}{1152\overline{\sigma}_\infty^2} > \mathcal{B}_\infty - E(\mathcal{Q}\,|\,i), \tag{D.80}$$

*for any $\varepsilon > 0$, for $\mathcal{V}_1, \dots, \mathcal{V}_{N_{crit}}$ neighborhoods of $\mathcal{C}_1, \dots, \mathcal{C}_{N_{crit}}$ small enough, there is $\eta_0 > 0$ such that for all $0 < \eta < \eta_0$, $x \in \mathcal{V}_i$:*

$$\mathbb{E}_x[\tau] \ge e^{\frac{E(\mathcal{Q}\,|\,i) - \varepsilon}{\eta}}. \tag{D.81}$$

The only difference with Theorem D.3 is that the result is uniform over the initial state in $\mathcal{V}_i$ instead of being specific to a given initial point. Its proof is identical to the proof of Theorem D.3: it simply uses Lemma D.5 to obtain the necessary variant of Lemma D.6 and then follows the same steps. It is therefore omitted.

## D.4 Basin-dependent convergence result

Let us recall the potential function from [5].

**Definition 15** (Potential, [5, Def. 4]). Define, for $x \in \mathbb{R}^d$

$$U_\infty(x) = 2\alpha_\infty \circ f(x) \tag{D.82}$$

where $\alpha_\infty : \mathbb{R}^d \to \mathbb{R}$ is a twice continuously differentiable primitive of $1/\sigma_\infty^2$.

**Lemma D.10.** *For any $\gamma \in \mathcal{C}([0,T])$, $t \in [0,T]$, we have*

$$U_\infty(\gamma(t)) - U_\infty(\gamma(0)) \leq \mathcal{S}_{0,T}(\gamma) . \tag{D.83}$$

*Proof.* We have that, with $U_\infty$ defined in Definition 15, by Young's inequality,

$$
\begin{aligned}
U_\infty(\gamma_t) - U_\infty(\gamma_0) &= 2 \int_0^t \frac{\langle \dot{\gamma}_s, \nabla f(\gamma_s) \rangle}{\sigma_\infty^2 \circ f(\gamma_s)} \, ds \\
&\leq \int_0^t \frac{\|\dot{\gamma}_s\|^2}{2\sigma_\infty^2 \circ f(\gamma_s)} \, ds + \int_0^t \frac{\|\nabla f(\gamma_s)\|^2}{2\sigma_\infty^2 \circ f(\gamma_s)} \, ds + \int_0^t \frac{\langle \dot{\gamma}_s, \nabla f(\gamma_s) \rangle}{\sigma_\infty^2 \circ f(\gamma_s)} \, ds \\
&= \int_0^t \frac{\|\dot{\gamma}_s + \nabla f(\gamma_s)\|^2}{2\sigma_\infty^2 \circ f(\gamma_s)} \, ds \\
&\leq \int_0^t \mathcal{L}(\gamma_s, \dot{\gamma}_s) \, ds \\
&\leq \mathcal{S}_{0,T}(\gamma) . 
\end{aligned}
\tag{D.84}
$$

where we used Lemma B.2 in the last inequality. ∎

**Lemma D.11.** $U_\infty$ *is coercive i.e., $U_\infty(x) \to +\infty$ as $\|x\| \to +\infty$.*

*Proof.* By Assumption 8, $\sigma_\infty^2(s) = o(s^2)$ as $s \to \infty$ and therefore $\alpha$ a primitive of $1/\sigma_\infty^2$ must be coercive. By coercivity of $f$ (Assumption 5), $U_\infty = 2\alpha \circ f$ is also coercive. ∎

**Lemma D.12.** *For any $\mathcal{K} \subset \mathbb{R}^d$ compact, for any $c > 0$, the set*

$$\left\{ \gamma(t) : t \in [0,T], \gamma \in \mathcal{C}([0,T]), T \in \mathbb{N}, \gamma_0 \in \mathcal{K}, \mathcal{S}_{t,T}(\gamma) \leq c \right\} \tag{D.85}$$

*is included in the set*

$$\left\{ x \in \mathbb{R}^d : U_\infty(x) \leq c + \sup_{\mathcal{K}} U_\infty \right\}, \tag{D.86}$$

*which is compact.*

*Proof.* Consider such a path $\gamma \in \mathcal{C}([0,T])$ with $T \in \mathbb{N}$. By Lemma D.10, we have, for any $t \in [0,T]$,

$$U_\infty(\gamma(t)) - U_\infty(\gamma(0)) \leq \mathcal{S}_{0,T}(\gamma) \leq c . \tag{D.87}$$

Hence the set from (D.85) is included in

$$\left\{ x \in \mathbb{R}^d : U_\infty(x) \leq c + \sup_{\mathcal{K}} U_\infty \right\}. \tag{D.88}$$

Now, by Lemma D.11, the set of (D.88) is compact, so that the set of (D.85) is bounded. ∎

The following lemma is a slight refinement of [5, Lem. D.33]. For any $C \subset \mathrm{crit}(f)$ connected component of the set of critical points, we define its basin of attraction as the set of points from which the gradient flow converges to $C$:

$$\mathrm{Attr}(C) := \left\{ x \in \mathbb{R}^d : \lim_{t \to +\infty} d(\Theta_t(x), C) = 0 \right\}. \tag{D.89}$$

**Lemma D.13.** *For any $C \subset \mathrm{crit}(f)$ connected component which is minimizing, for $\delta > 0$ small enough, for $\delta' \in [0, \delta)$ small enough,*

$$\mathcal{B}(\mathcal{U}_{\delta'}(C), \mathbb{R}^d \setminus \mathcal{U}_\delta(C)) > 0. \tag{D.90}$$

*Moreover, $\mathcal{U}_\delta(C)$ can be assumed to be contained in $\mathrm{Attr}(C)$.*

*Proof.* Since $C$ is minimizing, there exists $\delta_0 > 0$ such that, for any $x \in \mathcal{U}_{\delta_0}(C) \setminus C$, $f(x) > f_C$ where $f_C$ is the value of $f$ on $C$. Moreover, by Lemma B.9, taking $\delta > 0$ small enough also ensures that $\mathcal{U}_{\delta_0}(C) \subset \mathrm{Attr}(C)$.

Take $\delta \leq \delta_0$, $\mathcal{U} := \mathcal{U}_\delta(C)$ and $\Delta := \min\{U_\infty(x) - \alpha_\infty(f_C) : x \in \mathbb{R}^d, \ d(x, C) = \delta/2\}$. By the continuity of $U_\infty$ and the fact that $\alpha_\infty$ is (strictly) increasing, we have that $\Delta > 0$.

Finally, take $\delta' \in [0, \delta/2)$ small enough so that, for any $x \in \mathcal{U}_{\delta'}(C)$, $f(x) \leq \alpha_\infty(f_C) + \Delta/2$. To conclude the proof of this lemma, we now show that $\mathcal{B}(\mathcal{U}_{\delta'}(C), \mathbb{R}^d \setminus \mathcal{U}_\delta(C)) > 0$. Consider some $T > 0$ and $\gamma \in \mathcal{C}([0, T], \mathbb{R}^d)$ such that $\gamma_0 \in \mathcal{U}_{\delta'}(C)$, $\gamma_T \in \mathbb{R}^d \setminus \mathcal{U}_\delta(C)$. By continuity of $\gamma$ and $d(\cdot, C)$, there exists $t \in [0, T]$ such that $d(\gamma_t, C) = \delta/2$. By Lemma D.10, we have that

$$\begin{aligned}
\Delta &\leq U_\infty(\gamma_t) - \alpha_\infty(f_C) \\
&= U_\infty(\gamma_t) - U_\infty(\gamma_0) + U_\infty(\gamma_0) - \alpha_\infty(f_C) \\
&\leq \mathcal{S}_{0,T}(\gamma) + \frac{\Delta}{2}.
\end{aligned} \tag{D.91}$$

Since this is valid for any $\gamma$, we obtain that $\mathcal{B}(\mathcal{U}_{\delta'}(C), \mathbb{R}^d \setminus \mathcal{U}_\delta(C)) \geq \Delta/2 > 0$. ∎

We now proceed to the key result of this section.

**Lemma D.14.** *For any $C \subset \mathrm{crit}(f)$ connected component which is minimizing, for any $x \in \mathrm{Attr}(C)$,*

$$\mathcal{B}(x, \mathbb{R}^d \setminus \mathrm{Attr}(C)) > 0. \tag{D.92}$$

*Proof.* First, let us invoke Lemma D.13 to obtain some $0 < \delta' < \delta$ such that

$$\mathcal{B}(\mathcal{U}_{\delta'}(C), \mathbb{R}^d \setminus \mathcal{U}_\delta(C)) > 0. \tag{D.93}$$

and $\mathcal{U}_\delta(C) \subset \mathrm{Attr}(C)$.

Then, by Lemma D.12 and continuity of $f$ and $\sigma_\infty^2$, the set

$$\left\{ \sigma_\infty^2(f(\gamma(t))) : t \in [0, T], \ \gamma \in \mathcal{C}([0, T]), T \in \mathbb{N}, \ \gamma_0 = x, \ \mathcal{S}_{t,T}(\gamma) \leq 1 \right\} \tag{D.94}$$

is bounded by some finite constant $a \in (0, +\infty)$.

With $(\Theta_t(x))_{t \geq 0}$ the gradient flow starting from $x$ (Definition 4), by definition of $\mathrm{Attr}(C)$, there exists $S > 0$ such that

$$\Theta_S(x) \in \mathcal{U}_{\delta/2}(C). \tag{D.95}$$

Now, consider some $T \in \mathbb{N}$ and $\gamma \in \mathcal{C}([0, T])$ such that $\gamma_0 = x$, $\gamma_T \in \mathbb{R}^d \setminus \mathrm{Attr}(C)$ and $\mathcal{S}_T(\gamma) \leq 1$.

We now define

$$g_t := \frac{1}{2} \|\gamma_t - \Theta_t(x)\|^2. \tag{D.96}$$

In particular, $\gamma$ must be differentiable almost everywhere and so we have

$$\begin{aligned}
\dot{g}_t &= \langle \dot{\gamma}_t - \dot{\Theta}_t(x), \gamma_t - \Theta_t(x) \rangle \\
&= \langle \nabla f(\Theta_t(x)) - \nabla f(\gamma_t), \gamma_t - \Theta_t(x) \rangle + \langle \dot{\gamma}_t + \nabla f(\gamma_t), \gamma_t - \Theta_t(x) \rangle \\
&\leq \beta \|\gamma_t - \Theta_t(x)\|^2 + \frac{1}{2} \|\dot{\gamma}_t + \nabla f(\gamma_t)\|^2 + \frac{1}{2} \|\gamma_t - \Theta_t(x)\|^2 \\
&= \left( \beta + \frac{1}{2} \right) g_t + \frac{1}{2} \|\dot{\gamma}_t + \nabla f(\gamma_t)\|^2,
\end{aligned} \tag{D.97}$$

where in the last inequality we used the smoothness of $f$ (Assumption 5) and Young's inequality.

Rewriting (D.97), we have

$$
\begin{aligned}
\frac{d}{dt}\left(e^{-(\beta+\frac{1}{2})t} g_t\right) &\le e^{-(\beta+1)t} \frac{1}{2} \|\dot{\gamma}_t + \nabla f(\gamma_t)\|^2 \\
&\le \frac{1}{2} \|\dot{\gamma}_t + \nabla f(\gamma_t)\|^2 ,
\end{aligned}
\tag{D.98}
$$

so that integrating yields, for any $t \in [0, T]$,

$$
g_t \le \frac{e^{(\beta+\frac{1}{2})t}}{2} \int_0^t \|\dot{\gamma}_s + \nabla f(\gamma_s)\|^2 \, ds
\tag{D.99}
$$

where we used that $\Theta_0(x) = x = \gamma_0$.

Furthermore, introducing $a$, we have

$$
\begin{aligned}
g_t &\le a e^{(\beta+\frac{1}{2})t} \int_0^t \frac{\|\dot{\gamma}_s + \nabla f(\gamma_s)\|^2}{\sigma_\infty^2(f(\gamma_s))} \, ds \\
&\le a e^{(\beta+\frac{1}{2})t} \mathcal{S}_{0,t}(\gamma) ,
\end{aligned}
\tag{D.100}
$$

by Lemma B.2.

Hence, we have that, for any $t \in [0, T]$,

$$
\|\gamma_t - \Theta_t(x)\|^2 \le 2a e^{(\beta+\frac{1}{2})t} \mathcal{S}_{0,t}(\gamma) .
\tag{D.101}
$$

Therefore, if we consider $\gamma$ such that, in addition, $\mathcal{S}_T(\gamma) \le \frac{e^{-(\beta+\frac{1}{2})S}}{2a} \times \frac{\delta'^2}{4}$, we have that, for $t = \min(T, S)$,

$$
\|\gamma_t - \Theta_t(x)\| \le \frac{\delta'}{2} .
\tag{D.102}
$$

We now distinguish two cases:

- If $t = S$ i.e., $S \le T$, then (D.102) combined with (D.95) yields that $\gamma_S \in \mathcal{U}_{\delta'}(C)$. But $\gamma_T$ must be in $\mathbb{R}^d \setminus \text{Attr}(C)$ and so in $\mathbb{R}^d \setminus \mathcal{U}_\delta(C)$, so that one must have

$$
\mathcal{A}_T(\gamma) \ge \mathcal{B}(\mathcal{U}_{\delta'}(C), \mathbb{R}^d \setminus \text{Attr}(C)) .
\tag{D.103}
$$

- If $t = T$ i.e., $T \le S$, we extend the path $\gamma$ to $\varphi \in \mathcal{C}([0, S])$ such that $\varphi_t = \gamma_t$ for $t \in [0, T]$ and $\varphi_t = \Theta_t(x)$ for $t \in [T, S]$. In particular, $\mathcal{S}_S(\varphi) = \mathcal{S}_T(\gamma) \le \frac{e^{-(\beta+\frac{1}{2})T}}{2a} \times \frac{\delta'^2}{4}$ so that the same computations leading to (D.102) can be applied to $\varphi$ to yield that

$$
\|\varphi_S - \Theta_S(x)\| \le \frac{\delta'}{2} .
\tag{D.104}
$$

This means that $\varphi_S \in \mathcal{U}_{\delta'}(C)$ which is included in $\text{Attr}(C)$. This is contradiction since this means that $\gamma_T$ is in $\text{Attr}(C)$. Therefore, this case cannot happen.

Therefore, we have shown that, for any $\gamma \in \mathcal{C}([0, T])$ such that $\gamma_0 = x$ and $\gamma_T \in \mathbb{R}^d \setminus \text{Attr}(C)$, it holds that

$$
\mathcal{S}_T(\gamma) \ge \min\left(1, \frac{e^{-(\beta+\frac{1}{2})T}}{2a} \times \frac{\delta'^2}{4}, \mathcal{B}\left(\mathcal{U}_{\delta'}(C), \mathbb{R}^d \setminus \mathcal{U}_\delta(C)\right)\right),
\tag{D.105}
$$

which is positive by (D.93). ∎

We are now ready to prove the main result of this section.

**Theorem D.4.** *Consider $x_0 \in \mathbb{R}^d$ that belongs to $\mathrm{Attr}(\mathcal{C}_i)$ for some $i \in \{1, \ldots, N_{crit}\}$ such that $\mathcal{C}_i$ is minimizing. Then, for any $\varepsilon > 0$, for $\mathcal{V}_1, \ldots, \mathcal{V}_{N_{crit}}$ neighborhoods of $\mathcal{C}_1, \ldots, \mathcal{C}_{N_{crit}}$ small enough, there is $\eta_0 > 0$, an event H and a positive constant A such that for all $0 < \eta < \eta_0$,*

$$\mathbb{P}_{x_0}(H) \geq 1 - e^{-A/\eta} \tag{D.106}$$

*and,*

$$e^{\frac{E(\mathcal{Q}|i)-\varepsilon}{\eta}} \leq \mathbb{E}_{x_0}\left[\tau_{\mathcal{Q}}^{\eta} \mid H\right] \leq e^{\frac{E(\mathcal{Q}|i)+\varepsilon}{\eta}} . \tag{D.107}$$

*Proof.* First, Lemma D.14 ensures that $\mathcal{B}(x_0, \mathbb{R}^d \setminus \mathrm{Attr}(\mathcal{C}_i)) > 0$ so that, for any $j \in \mathcal{V}$, $j \neq i$, it holds $\mathcal{B}(x_0, \mathcal{C}_j) > 0$. Define

$$A := \min_{j \in \mathcal{V},\, j \neq i} \mathcal{B}(x_0, \mathcal{C}_j) > 0 . \tag{D.108}$$

Let us now apply Lemma C.11 to obtain that, for any small enough neighborhoods $\mathcal{V}_i$ of $\mathcal{C}_i$, $i = 1, \ldots, N_{crit}$, there is some $\eta_0 > 0$ such that for all $j \in \{1, \ldots, N_{crit}\}$, $0 < \eta < \eta_0$,

$$\mathbb{Q}_{\mathcal{V}}(x, \mathcal{V}_j) \leq e^{-\frac{A}{2\eta}} . \tag{D.109}$$

Now define the event $H$ as

$$H := \left\{ x_{\tau_1^{\eta}}^{\eta} \in \mathcal{V}_i \right\} \tag{D.110}$$

with the notation of Definition 8. (D.109) then ensures that

$$\mathbb{P}_{x_0}(H) \geq 1 - e^{-\frac{A}{4\eta}} , \tag{D.111}$$

provided that $\eta$ is small enough so that $e^{-\frac{A}{4\eta}} N_{crit} \leq 1$. If $i \in \mathcal{Q}$, the result is immediate. In the following, we assume that $i \notin \mathcal{Q}$.

Let us now invoke Lemma D.5 to obtain that, at the potential cost of requiring smaller $\mathcal{V}_1, \ldots, \mathcal{V}_{N_{crit}}$ neighborhoods of $\mathcal{C}_1, \ldots, \mathcal{C}_{N_{crit}}$ or of reducing $\eta_0$, it holds that, for any $0 < \eta < \eta_0$, $x \in x_0$,

$$e^{\frac{E(\mathcal{Q}|i)-\varepsilon}{\eta}} \leq \mathbb{E}_x\left[\tau_{\mathcal{Q}}^{\eta}\right] \leq e^{\frac{E(\mathcal{Q}|i)+\varepsilon}{\eta}} . \tag{D.112}$$

To conclude the proof, write, by the strong Markov property,

$$\begin{aligned}
\mathbb{E}_{x_0}\left[\tau_{\mathcal{Q}}^{\eta} \mid H\right] &= \frac{\mathbb{E}_{x_0}\left[\mathbb{1}_H \tau_{\mathcal{Q}}^{\eta}\right]}{\mathbb{P}_{x_0}(H)} \\
&= \frac{\mathbb{E}_{x_0}\left[\mathbb{1}_H\left(1 + \mathbb{E}_{x_{\tau_1^{\eta}}^{\eta}}\left[\tau_{\mathcal{Q}}^{\eta}\right]\right)\right]}{\mathbb{P}_{x_0}(H)} \\
&\leq \frac{\mathbb{E}_{x_0}[\mathbb{1}_H] e^{\frac{E(\mathcal{Q}|i)+2\varepsilon}{\eta}}}{\mathbb{P}_{x_0}(H)} \\
&= e^{\frac{E(\mathcal{Q}|i)+2\varepsilon}{\eta}} ,
\end{aligned} \tag{D.113}$$

where we used (D.112) in the last inequality and taking $\eta$ small enough so that $e^{\frac{\varepsilon}{\eta}} \geq 2$. We have thus shown the RHS of the inequality of the statement, and the LHS is obtained by the same argument.

∎

Similarly to how Theorem D.2 was obtained from Theorem D.1, we immediately obtain the following corollary.

**Theorem D.5.** *Consider $x_0 \in \mathbb{R}^d$ that belongs to* $\text{Attr}(\mathcal{C}_i)$ *for some* $i \in \{1, \ldots, N_{crit}\}$ *such that* $\mathcal{C}_i$ *is minimizing. Then, for any* $\varepsilon > 0$, *for* $\mathcal{V}_1, \ldots, \mathcal{V}_{N_{crit}}$ *neighborhoods of* $\mathcal{C}_1, \ldots, \mathcal{C}_{N_{crit}}$ *small enough, there is* $\eta_0 > 0$, *an event H and a positive constant A such that for all* $0 < \eta < \eta_0$,

$$\mathbb{P}_{x_0}(H) \geq 1 - e^{-A/\eta} \tag{D.114}$$

*and,*

$$\mathbb{E}_{x_0}[\tau \mid H] \leq e^{\frac{E(\mathcal{Q}\mid i)+\varepsilon}{\eta}}. \tag{D.115}$$

We now turn our attention to the lower-bound counterpart of Theorem D.5.

**Theorem D.6.** *Consider $x_0 \in \mathbb{R}^d$ that belongs to* $\text{Attr}(\mathcal{C}_i)$ *for some* $i \in \{1, \ldots, N_{crit}\}$ *such that* $\mathcal{C}_i$ *is minimizing. Under Assumption 10 and assuming that,*

$$\frac{\mu R^2}{1152\overline{\sigma}_\infty^2} > \mathcal{B}_\infty - E(\mathcal{Q}\mid i), \tag{D.116}$$

*for any* $\varepsilon > 0$, *for* $\mathcal{V}_1, \ldots, \mathcal{V}_{N_{crit}}$ *neighborhoods of* $\mathcal{C}_1, \ldots, \mathcal{C}_{N_{crit}}$ *small enough, there is* $\eta_0 > 0$, *an event H and a positive constant A such that for all* $0 < \eta < \eta_0$,

$$\mathbb{P}_{x_0}(H) \geq 1 - e^{-A/\eta} \tag{D.117}$$

*and,*

$$\mathbb{E}_{x_0}[\tau] \geq e^{\frac{E(\mathcal{Q}\mid i)-\varepsilon}{\eta}}. \tag{D.118}$$

Theorem D.6 is obtained by combining Lemma D.9 with Lemma D.14 in the same way as Theorem D.4 was obtained from Lemma D.5.

## D.5 Full transition costs

**Definition 16.** We define, for $i, j \in \{1, \ldots, N_{\text{crit}}\}$,

$$\begin{aligned} C_{i,j} &:= \inf\{\mathcal{A}_N(\xi) : N \geq 1, \xi \in \mathcal{D}_r(N), \xi_0 \in \mathcal{C}_i, \xi_{N-1} \in \mathcal{C}_j\} \\ &= \inf\{\mathcal{S}_T(\gamma) : T \in \mathbb{N}, \gamma \in \mathcal{D}_r(T), \gamma_0 \in \mathcal{C}_i, \gamma_T \in \mathcal{C}_j\}. \end{aligned} \tag{D.119}$$

With the same arguments as Lemma C.2, we can show the alternate expression for $C_{i,j}$.

**Lemma D.15.** *For any* $i, j \in \{1, \ldots, N_{crit}\}$, *it holds that*

$$C_{i,j} = \lim_{\delta \to 0} C_{i,j}^\delta, \tag{D.120}$$

*where*

$$\begin{aligned} C_{i,j}^\delta &:= \inf\{\mathcal{A}_N(\xi) : N \geq 1, \xi \in \mathcal{D}_r(N), \xi_0 \in \mathcal{U}_\delta(\mathcal{C}_i), \xi_{N-1} \in \mathcal{U}_\delta(\mathcal{C}_j)\} \\ &= \inf\{\mathcal{S}_T(\gamma) : T \in \mathbb{N}, \gamma \in \mathcal{D}_r(T), \gamma_0 \in \mathcal{U}_\delta(\mathcal{C}_i), \gamma_T \in \mathcal{U}_\delta(\mathcal{C}_j)\}. \end{aligned} \tag{D.121}$$

We can also relate $C_{i,j}$ in terms $B_{i,j}$.

**Lemma D.16.** *For any* $i \in \{1, \ldots, K\}$, $j \in \{1, \ldots, N_{crit}\}$, *it holds that*

$$C_{i,j} \leq B_{i,j}, \tag{D.122}$$

*and, if* $C_{i,j} < B_{i,j}$, *then there must exist* $l \in \mathcal{Q}$ *such that* $B_{i,l} \leq C_{i,j} < B_{i,j}$.

This lemma is a direct consequence of the definitions of $C_{i,j}$ (Definition 16) and $B_{i,j}$ (Definition 10) and is therefore omitted.

**Lemma D.17.** *For any* $i \in \{1, \ldots, N_{crit}\}$,

$$E(\mathcal{Q}\mid i) \leq \min\left\{\sum_{k \to l \in g} C_{k,l} : g \in \mathcal{G}(\mathcal{Q})\right\} - \min\left\{\sum_{k \to l \in g} C_{k,l} : g \in \mathcal{G}(i \nrightarrow \mathcal{Q})\right\}. \tag{D.123}$$

*Proof.* By Definition 13, we have that

$$E(\mathcal{Q} \mid i) = E(\mathcal{Q}) - E(i \not\rightsquigarrow \mathcal{Q}).$$ (D.124)

Let us begin by lower-bounding the second term. By Lemma D.16, it holds that

$$
\begin{aligned}
E(i \not\rightsquigarrow \mathcal{Q}) &= \min\left\{ \sum_{k \to l \in g} B_{k,l} : g \in \mathcal{G}(i \not\rightsquigarrow \mathcal{Q}) \right\} \\
&\geq \min\left\{ \sum_{k \to l \in g} C_{k,l} : g \in \mathcal{G}(i \not\rightsquigarrow \mathcal{Q}) \right\}.
\end{aligned}
$$ (D.125)

Now, let us turn our attention to the first term: by Lemma D.16, we have that

$$
\begin{aligned}
E(\mathcal{Q}) &:= \min\left\{ \sum_{k \to l \in g} B_{k,l} : g \in \mathcal{G}(\mathcal{Q}) \right\} \\
&\geq \min\left\{ \sum_{k \to l \in g} C_{k,l} : g \in \mathcal{G}(\mathcal{Q}) \right\}.
\end{aligned}
$$ (D.126)

But, for any $g \in \mathcal{G}(\mathcal{Q})$ which reaches that minimum, all its edges $k \to l$ must be such $B_{k,l} = C_{k,l}$ by Lemma D.16 (otherwise we would replace it by an edge to $\mathcal{Q}$ with a lower cost). Therefore, we must also have that

$$\min\left\{ \sum_{k \to l \in g} C_{k,l} : g \in \mathcal{G}(\mathcal{Q}) \right\} \geq \min\left\{ \sum_{k \to l \in g} B_{k,l} : g \in \mathcal{G}(\mathcal{Q}) \right\},$$ (D.127)

which shows that

$$E(\mathcal{Q}) = \min\left\{ \sum_{k \to l \in g} C_{k,l} : g \in \mathcal{G}(\mathcal{Q}) \right\}.$$ (D.128)

This concludes the proof. ∎

Finally, let us restate a result from [5] that will be relevant in the following.

**Lemma D.18** ([5, Lem. D.31]). *If $\mathcal{C}_i$ is not asymptotically stable, then there exists $j \in \mathcal{V}$ such that $B_{i,j} = 0$ and such that $\mathcal{C}_j$ is asymptotically stable.*

### D.6 Link to topological properties of the loss landscape

The goal of this section is to prove the following result.

**Proposition D.1.** *The following properties are equivalent:*

*(i) For all $x \in \mathbb{R}^d$, $E(\mathcal{Q} \mid x) = 0$.*

*(ii) For all $i \in \{1, \ldots, K\}$, $E(\mathcal{Q} \mid i) = 0$.*

*(iii) For all $i \in \{1, \ldots, K\}$, $\mathcal{K}_i$ is not locally minimizing.*

*Proof.* Items $(i)$ and $(ii)$ are equivalent by definition of $E(\mathcal{Q} \mid x)$ (Definition 14). We thus focus on the equivalence of $(ii)$ and $(iii)$.

First, let us consider the case where there is some $i \in \{1, \ldots, K\}$ such that $\mathcal{K}_i$ is locally minimizing. By Lemma D.13, there exists $\delta > 0$ small enough such that

$$\mathcal{B}(\mathcal{K}_i, \mathbb{R}^d \setminus \mathcal{U}_\delta(\mathcal{K}_i)) > 0.$$ (D.129)

As a consequence, for any $j \neq i$, $B_{i,j} > 0$.

Consider $g \in \mathcal{G}(\mathcal{V})$ such that

$$E(\mathcal{Q}) = \sum_{k \to l \in g} B_{k,l}.$$ (D.130)

Consider the graph $g'$ obtained by removing the edge in graph that exits $i$. This edge must have positive cost and therefore

$$\sum_{k\to l\in g} B_{k,l} > \sum_{k\to l\in g'} B_{k,l} \geq E(i \not\to Q). \tag{D.131}$$

Combining (D.130) and (D.131), we obtain that $E(Q\,|\,i) > 0$.

Let us now turn to the case where there is no $i \in \{1, \ldots, K\}$ such that $\mathcal{K}_i$ is locally minimizing. Take any $i \in \{1, \ldots, K\}$. We bound $E(Q\,|\,i)$ with Lemma D.17:

$$E(Q\,|\,i) \leq \min\left\{\sum_{k\to l\in g} C_{k,l} : g \in \mathcal{G}(Q)\right\} - \min\left\{\sum_{k\to l\in g} C_{k,l} : g \in \mathcal{G}(i \not\to Q)\right\}. \tag{D.132}$$

Take any $g \in \mathcal{G}(i \not\to Q)$. By Lemma D.18, there exists $j \in \mathcal{V}$ such that $B_{i,j} = 0$ and $\mathcal{K}_j$ is asymptotically stable. By assumption, $j$ must thus belong to $Q$. Consider $g' \in \mathcal{G}(Q)$ obtained by adding the edge $i \to j$ to $g$. (D.132) then yields that $E(Q\,|\,i) = 0$. $\blacksquare$

# E   Potential-based Bounds

## E.1   Noise assumptions

Assumption 11 introduces key requirements for the noise behavior through bounds on $\bar{\mathcal{H}}$ and $\bar{\mathcal{L}}$ around the gradient $\nabla f(x)$. Intuitively, the bound on $\bar{\mathcal{H}}$ controls how large the noise can be (upper bound), while the bound on $\bar{\mathcal{L}}$ ensures the noise maintains sufficient variance in all directions (lower bound) around the gradient. We then present two ways to satisfy these requirements:

**Gaussian noise.**  The first approach, detailed in Lemma E.2, considers the case when the noise follows a truncated Gaussian distribution. This is a standard Gaussian with variance $\sigma^2(f(x))$, but restricted to stay within a ball of radius $R(f(x))$.

**Support and covariance conditions.**  Lemma E.1 provides a more general approach with two main requirements:

- A support condition ensuring the noise can be large enough in any direction around $\nabla f(x)$ with some positive probability
- A lower bound on the covariance matrix of the noise

This second approach is more flexible as it allows for non-Gaussian noise distributions. A key example is discrete noise distributions, where $u(x, \omega)$ takes values in a finite set. The support condition then requires that some points in the support are sufficiently far from $\nabla f(x)$ in every direction.

**Assumption 11.**  For some $X \subset \mathbb{R}^d$, there exist $\overline{\sigma}^2, \underline{\sigma}^2 : \mathbb{R} \to (0, +\infty)$ continuous functions such that, for all $x \in X$, it holds that

$$\forall p \in \mathbb{R}^d \text{ s.t. } \|p\| \leq \frac{2\|\nabla f(x)\|}{\overline{\sigma}^2(f(x))}, \quad \bar{\mathcal{H}}(x, p) \leq \frac{\overline{\sigma}^2(f(x))\|p\|^2}{2}$$

$$\forall v \in \mathbb{R}^d \text{ s.t. } \|v - \nabla f(x)\| \leq \|\nabla f(x)\|, \quad \bar{\mathcal{L}}(x, v) \leq \frac{\|v\|^2}{2\underline{\sigma}^2(f(x))}. \tag{E.1}$$

*Remark* E.1.  Assumption 11 is a generalized and detailed version of Assumption 4. Indeed, our blanket assumptions on the noise Assumption 7 already imply an upper-bound on $\bar{\mathcal{H}}(x, p)$ with $\overline{\sigma}^2 = \sigma_\infty^2$ for any $x, p \in \mathbb{R}^d$. Assumption 11 allows for refined results when a tighter upper-bound on $\bar{\mathcal{H}}(x, p)$ is available for a restricted set of $x$ and $p$.

Note that, in general, we will have $\overline{\sigma}^2 \geq \underline{\sigma}^2$, hence the notation.

**Lemma E.1.**  *Assume that, for some $X \subset \mathbb{R}^d$ relatively compact,*

- *there exists $a > 0$ such that,*

$$b := \inf\left\{\mathbb{P}(\langle u(x, \omega) - \nabla f(x), q \rangle \geq a + \|\nabla f(x)\|) : x \in X, q \in \mathbb{S}^{d-1}\right\} > 0, \tag{E.2}$$

- *there exists $\zeta^2 : \mathbb{R} \to (0, +\infty)$ continuous function such that, for all $x \in X$, it holds that*

$$\text{cov}\, u(x, \omega) \succcurlyeq \zeta^2(f(x))I, \tag{E.3}$$

*Then, for any $x \in X$, $v \in \mathbb{R}^d$ such that $\|v - \nabla f(x)\| \leq \|\nabla f(x)\|$, it holds that*

$$\bar{\mathcal{L}}(x, v) \leq \max\left(1, \frac{\log b^{-1}}{ac}\right)\frac{\|v\|^2}{\zeta^2(f(x))}, \tag{E.4}$$

*where $c > 0$ is a constant small enough such that*

$$\forall x \in X, p \in \overline{\mathbb{B}}(0, c), \quad \left\|\text{Hess}_p\, \bar{\mathcal{H}}(x, p) - \text{cov}\, u(x, \omega)\right\| \leq \frac{1}{2}\zeta^2(f(x)), \tag{E.5}$$

*for the matrix norm associated with the Euclidean norm.*

Note that $c$ always exists by the relative compactness of $X$ and the fact that

$$\text{Hess}_p\, \bar{\mathcal{H}}(x, 0) = \text{cov}\, u(x, \omega). \tag{E.6}$$

*Proof.* The first part of the proof consists in showing that, for $x \in X$, $p \in \mathbb{R}^d$, it holds that

$$\bar{\mathcal{H}}(x, p) \leq \frac{\zeta^2(f(x)) \min(c, \|p\|) \|p\|}{4} . \tag{E.7}$$

Let us first consider the case where $\|p\| \leq c$. Since both $\bar{\mathcal{H}}(x, 0) = 0$ and $\nabla_p \bar{\mathcal{H}}(x, 0) = \mathbb{E}[u(x, \omega)] = 0$, we have that

$$
\begin{aligned}
\bar{\mathcal{H}}(x, p) &= \frac{1}{2} \int_0^1 \langle p, \mathrm{Hess}_p\, \bar{\mathcal{H}}(x, tp) p \rangle dt \\
&\geq \frac{1}{2} \left( \int_0^1 \langle p, \mathrm{cov}\, u(x, \omega) p \rangle dt - \frac{1}{2} \zeta^2(f(x)) \|p\|^2 \right) \\
&\geq \frac{\zeta^2(f(x)) \|p\|^2}{4} ,
\end{aligned}
\tag{E.8}
$$

where we successively used Cauchy-Schwarz inequality, (E.5) and (E.3). (E.8) readily yields (E.7) when $\|p\| \leq c$.

Let us now consider the case where $\|p\| > c$. By concavity of the function $s \mapsto s^{c/\|p\|} s$ on $\mathbb{R}_+$ and Jensen's inequality, we have that

$$
\begin{aligned}
\bar{\mathcal{H}}(x, p) &\geq \frac{\|p\|}{c} \bar{\mathcal{H}}\left(x, c \frac{p}{\|p\|}\right) \\
&\geq \frac{\|p\|}{c} \frac{\zeta^2(f(x)) \|c \frac{p}{\|p\|}\|^2}{4} = \frac{\zeta^2(f(x)) c \|p\|}{4} ,
\end{aligned}
\tag{E.9}
$$

where we used (E.8) in the last inequality. This concludes the proof of (E.7).

The second step of this proof now consists in showing that, for $x \in X$, $v \in \mathbb{R}^d$ such that $\|v - \nabla f(x)\| \leq \|\nabla f(x)\|$, it holds that

$$\arg\max_{p \in \mathbb{R}^d} \{\langle p, v \rangle - \bar{\mathcal{H}}(x, p)\} = \arg\min_{p \in \mathbb{R}^d} \{\bar{\mathcal{H}}(x, p) - \langle p, v \rangle\} \subset \bar{\mathbb{B}}\left(0, \frac{\log b^{-1}}{a}\right). \tag{E.10}$$

For $x \in X$, $p \in \mathbb{R}^d$, we can lower-bound $\bar{\mathcal{H}}(x, p)$ using (E.2):

$$
\begin{aligned}
\bar{\mathcal{H}}(x, p) &= \log \mathbb{E}[\exp(\langle p, u(x, \omega) \rangle)] \\
&= \langle p, \nabla f(x) \rangle + \log \mathbb{E}[\exp(\langle p, u(x, \omega) - \nabla f(x) \rangle)] \\
&\geq \langle p, \nabla f(x) \rangle + \log \mathbb{E}\left[ e^{\|p\|(a + \|\nabla f(x)\|)} \mathbb{1}\left\{\left\langle u(x, \omega) - \nabla f(x), \frac{p}{\|p\|} \right\rangle \geq a + \|\nabla f(x)\|\right\}\right] \\
&\geq \langle p, \nabla f(x) \rangle + \|p\|(a + \|\nabla f(x)\|) + \log b .
\end{aligned}
\tag{E.11}
$$

As a consequence, for $v \in \mathbb{R}^d$ such that $\|v - \nabla f(x)\| \leq \|\nabla f(x)\|$, we obtain that

$$
\begin{aligned}
\bar{\mathcal{H}}(x, p) - \langle p, v \rangle &\geq \langle p, \nabla f(x) - v \rangle + \|p\|(a + \|\nabla f(x)\|) + \log b \\
&\geq \|p\| a + \log b ,
\end{aligned}
\tag{E.12}
$$

where we used Cauchy-Schwarz inequality and the condition on $v$.

In particular, (E.12) implies that $p \mapsto \bar{\mathcal{H}}(x, p) - \langle p, v \rangle$ is coercive on $\mathbb{R}^d$ and therefore $\arg\min_{p \in \mathbb{R}^d} \{\bar{\mathcal{H}}(x, p) - \langle p, v \rangle\}$ is well-defined.

Remarking that $\bar{\mathcal{H}}(x, 0) - \langle 0, v \rangle = 0$, we can now conclude that $\arg\min_{p \in \mathbb{R}^d} \{\bar{\mathcal{H}}(x, p) - \langle p, v \rangle\}$ must be included in

$$\left\{p \in \mathbb{R}^d : \|p\| a + \log b \leq 0\right\} = \bar{\mathbb{B}}\left(0, \frac{\log b^{-1}}{a}\right), \tag{E.13}$$

which completes the proof of (E.10).

To conclude the proof of this lemma, we now use (E.10) to obtain that, for $x \in X$, $v \in \mathbb{R}^d$ such that $\|v - \nabla f(x)\| \le \|\nabla f(x)\|$,

$$
\begin{aligned}
\bar{\mathcal{L}}(x, v) &= \sup_{p \in \mathbb{R}^d} \left\{ \langle p, v \rangle - \bar{\mathcal{H}}(x, p) \right\} \\
&= \sup \left\{ \langle p, v \rangle - \bar{\mathcal{H}}(x, p) : p \in \overline{\mathbb{B}} \left( 0, \frac{\log b^{-1}}{a} \right) \right\} \\
&\le \sup \left\{ \langle p, v \rangle - \frac{\zeta^2(f(x)) \min(c, \|p\|) \|p\|}{4} : p \in \overline{\mathbb{B}} \left( 0, \frac{\log b^{-1}}{a} \right) \right\},
\end{aligned}
\tag{E.14}
$$

where we used (E.7) in the last inequality.

Now, we note that, for any $p \in \overline{\mathbb{B}} \left( 0, \frac{\log b^{-1}}{a} \right)$, it holds that

$$
\frac{\zeta^2(f(x)) \min(c, \|p\|) \|p\|}{4} \ge 
\begin{cases}
\frac{\zeta^2(f(x)) \|p\|^2}{4} & \text{if } \|p\| \le c \\
\frac{a}{\log b^{-1}} \frac{\zeta^2(f(x)) c \|p\|^2}{4} & \text{if } \|p\| > c
\end{cases}
$$

$$
\ge \min \left( 1, \frac{ac}{\log b^{-1}} \right) \frac{\zeta^2(f(x)) \|p\|^2}{4}.
\tag{E.15}
$$

Plugging (E.15) into (E.14), we obtain that

$$
\begin{aligned}
\bar{\mathcal{L}}(x, v) &\le \sup \left\{ \langle p, v \rangle - \min \left( 1, \frac{ac}{\log b^{-1}} \right) \frac{\zeta^2(f(x)) \|p\|^2}{4} : p \in \overline{\mathbb{B}} \left( 0, \frac{\log b^{-1}}{a} \right) \right\} \\
&\le \sup \left\{ \langle p, v \rangle - \min \left( 1, \frac{ac}{\log b^{-1}} \right) \frac{\zeta^2(f(x)) \|p\|^2}{4} : p \in \mathbb{R}^d \right\} \\
&= \max \left( 1, \frac{\log b^{-1}}{ac} \right) \frac{\|v\|^2}{\zeta^2(f(x))},
\end{aligned}
\tag{E.16}
$$

which concludes the proof of this lemma. ∎

The following lemma is an immediate consequence of Lemma G.1.

**Lemma E.2.** *Consider $\varepsilon > 0$. Assume that $u(x, \omega)$ follows a Gaussian distribution $\mathcal{N}(0, \sigma^2(f(x))I)$ conditioned on being in $\overline{\mathbb{B}}(0, R(f(x))$ for all $x \in \mathbb{R}^d$ and some continuous function $\sigma^2, R : \mathbb{R} \to (0, +\infty)$. Assume that $8\|\nabla f(x)\| \le R(f(x))$ for all $x \in \mathbb{R}^d$ and that $\inf_{x \in \mathbb{R}^d} \frac{R(f(x))}{\sigma(f(x))} \ge \mathcal{O}(\log 1/\varepsilon)$. Then Assumption 11 holds with $X = \mathbb{R}^d$, $\overline{\sigma}^2(f(x)) = (1 + \varepsilon)\sigma^2(f(x))$ and $\underline{\sigma}^2(f(x)) = (1 - \varepsilon)\sigma^2(f(x))$ for all $x \in \mathbb{R}^d$.*

### E.2 Potentials and path reversal

Let us now define the potentials associated to the variance functions $\overline{\sigma}^2$ and $\underline{\sigma}^2$ introduced in Assumption 11.

**Definition 17.** *Given $\overline{\sigma}^2, \underline{\sigma}^2 : \mathbb{R} \to (0, +\infty)$ continuous functions, we define the potentials $\overline{U}, \underline{U} : \mathbb{R} \to \mathbb{R}$ as*

$$
\overline{U}(x) := 2\overline{\alpha}(f(x)) \quad \text{where } \overline{\alpha} : \mathbb{R} \to \mathbb{R} \text{ is a primitive of } 1/\underline{\sigma}^2
\tag{E.17}
$$

$$
\underline{U}(x) := 2\underline{\alpha}(f(x)) \quad \text{where } \underline{\alpha} : \mathbb{R} \to \mathbb{R} \text{ is a primitive of } 1/\overline{\sigma}^2.
\tag{E.18}
$$

We begin by adapting [5, Lem .E.1] to the current setting.

**Lemma E.3.** *Consider $\gamma \in \mathcal{C}([0, T])$. Then, there exists $\widetilde{\gamma} \in \mathcal{C}([0, S])$ a reparametrization of $\gamma$ such that, for any $t \in [0, S]$,*

$$
\|\dot{\widetilde{\gamma}}_s\| = \|\nabla f(\widetilde{\gamma}_s)\|.
\tag{E.19}
$$

*and, under Assumption 11,*

$$
\mathcal{S}_T(\gamma) \ge \int_0^S \frac{\|\dot{\widetilde{\gamma}}_s + \nabla f(\widetilde{\gamma}_s)\|^2}{2\overline{\sigma}^2(f(\widetilde{\gamma}_s))} \, ds.
\tag{E.20}
$$

*Proof.* By the proof Freidlin & Wentzell [22, Chap. 4, Lem. 3.1], there exists $t(s)$ change of time such that, with $\widetilde{\gamma}_s = \gamma_{t(s)}$, $\|\dot{\widetilde{\gamma}}_s\| = \|\nabla f(\widetilde{\gamma}_s)\|$.

We have that

$$\mathcal{S}_T(\gamma) = \int_0^{t^{-1}(T)} \dot{t}(s) \mathcal{L}(\widetilde{\gamma}_s, (\dot{t}(s))^{-1}\dot{\widetilde{\gamma}}_s)\, ds\,, \tag{E.21}$$

so it suffices to bound $\mathcal{L}(\widetilde{\gamma}_s, \dot{\widetilde{\gamma}}_s)$ from below: by definition, we have

$$\mathcal{L}(\widetilde{\gamma}_s, (\dot{t}(s))^{-1}\dot{\widetilde{\gamma}}_s) \geq \sup\left\{ \langle p, (\dot{t}(s))^{-1}\dot{\widetilde{\gamma}}_s + \nabla f(\widetilde{\gamma}_s)\rangle - \bar{\mathcal{H}}(\widetilde{\gamma}_s, p) : \|p\| \leq \frac{2\|\nabla f(\widetilde{\gamma}_s)\|}{\overline{\sigma}^2(f(\widetilde{\gamma}_s))} \right\}$$

$$\geq \sup\left\{ \langle p, (\dot{t}(s))^{-1}\dot{\widetilde{\gamma}}_s + \nabla f(\widetilde{\gamma}_s)\rangle - \frac{\overline{\sigma}^2(f(\widetilde{\gamma}_s))}{2}\|p\|^2 : \|p\| \leq \frac{2\|\nabla f(\widetilde{\gamma}_s)\|}{\overline{\sigma}^2(f(\widetilde{\gamma}_s))} \right\}, \tag{E.22}$$

by Assumption 11. Applying Lemma G.2 with $v \leftarrow (\dot{t}(s))^{-1}\dot{\widetilde{\gamma}}_t$, $w \leftarrow \nabla f(\widetilde{\gamma}_s)$ and $\lambda \leftarrow (\dot{t}(s))^{-1}$, now exactly yields, for almost all $s$,

$$\mathcal{L}(\widetilde{\gamma}_s, (\dot{t}(s))^{-1}\dot{\widetilde{\gamma}}_s) \geq (\dot{t}(s))^{-1} \sup\left\{ \langle p, \dot{\widetilde{\gamma}}_s + \nabla f(\widetilde{\gamma}_s)\rangle - \frac{\overline{\sigma}^2(f(\widetilde{\gamma}_s))}{2}\|p\|^2 : \|p\| \leq \frac{2\|\nabla f(\widetilde{\gamma}_s)\|}{\overline{\sigma}^2(f(\widetilde{\gamma}_s))} \right\}$$

$$= (\dot{t}(s))^{-1} \frac{\|\dot{\widetilde{\gamma}}_s + \nabla f(\widetilde{\gamma}_s)\|^2}{2\overline{\sigma}^2(f(\widetilde{\gamma}_s))}\,, \tag{E.23}$$

since $p \mapsto \langle p, \dot{\widetilde{\gamma}}_s + \nabla f(\widetilde{\gamma}_s)\rangle - \frac{\overline{\sigma}^2(f(\widetilde{\gamma}_s))}{2}\|p\|^2$ reaches it maximum at $p^* = \frac{\dot{\widetilde{\gamma}}_s + \nabla f(\widetilde{\gamma}_s)}{\overline{\sigma}^2(f(\widetilde{\gamma}_s))}$ whose norm satisfies $\|p^*\| \leq \frac{2\|\nabla f(\widetilde{\gamma}_s)\|}{\overline{\sigma}^2(f(\widetilde{\gamma}_s))}$. ∎

An important consequence of Assumption 11 is that the action cost of going from $x$ to $x'$ can be related to the action cost of going from $x'$ to $x$. This is formalized in Lemma E.4.

**Lemma E.4.** *For any $T > 0$, $x, x' \in \mathbb{R}^d$, $\gamma \in \mathcal{C}([0,T])$ such that $\gamma_0 = x$ and $\gamma_T = x'$, under Assumption 11, if $\gamma$ is contained in X, it holds that,*

$$\mathcal{S}_T(\gamma) \geq \inf_{x \in X} \frac{\sigma^2(f(x))}{\overline{\sigma}^2(f(x))} \times \left( \mathcal{B}(x, x') + \overline{U}(x') - \overline{U}(x) \right), \tag{E.24}$$

*Proof.* Let us first invoke Lemma E.3: there exists $\widetilde{\gamma} \in \mathcal{C}([0,S])$ a reparametrization of $\gamma$ such that, for any $t \in [0,S]$,

$$\|\dot{\widetilde{\gamma}}_s\| = \|\nabla f(\widetilde{\gamma}_s)\|\,. \tag{E.25}$$

and,

$$\mathcal{S}_T(\gamma) \geq \int_0^S \frac{\|\dot{\widetilde{\gamma}}_s + \nabla f(\widetilde{\gamma}_s)\|^2}{2\overline{\sigma}^2(f(\widetilde{\gamma}_s))}\, ds$$

$$\geq \inf_{x \in X} \frac{\sigma^2(f(x))}{\overline{\sigma}^2(f(x))} \times \left( \int_0^S \frac{\|\dot{\widetilde{\gamma}}_s + \nabla f(\widetilde{\gamma}_s)\|^2}{2\underline{\sigma}^2(f(\widetilde{\gamma}_s))}\, ds \right). \tag{E.26}$$

It now suffices to lower-bound the integral on the RHS of (E.26). We have

$$\int_0^S \frac{\|\dot{\widetilde{\gamma}}_s + \nabla f(\widetilde{\gamma}_s)\|^2}{2\underline{\sigma}^2(f(\widetilde{\gamma}_s))}\, ds = \int_0^S \frac{\|-\dot{\widetilde{\gamma}}_s + \nabla f(\widetilde{\gamma}_s)\|^2}{2\underline{\sigma}^2(f(\widetilde{\gamma}_s))}\, ds + \int_0^S \frac{2\langle \dot{\widetilde{\gamma}}_s, \nabla f(\widetilde{\gamma}_s)\rangle}{\underline{\sigma}^2(f(\widetilde{\gamma}_s))}\, ds\,. \tag{E.27}$$

By definition of $\overline{U}$ (Definition 17) the second integral is equal to

$$\int_0^S \frac{2\langle \dot{\widetilde{\gamma}}_s, \nabla f(\widetilde{\gamma}_s)\rangle}{\underline{\sigma}^2(f(\widetilde{\gamma}_s))}\, ds = \int_0^S \langle \dot{\widetilde{\gamma}}_s, \nabla \overline{U}(\widetilde{\gamma}_s)\rangle\, ds$$

$$= \overline{U}(x') - \overline{U}(x) \,. \tag{E.28}$$

For the first integral in (E.27), define the path $(\varphi_s)_{s \in [0,S]}$ by $\varphi_s = \widetilde{\gamma}_{S-s}$. We have that

$$
\begin{aligned}
\int_0^S \frac{\|-\dot{\widetilde{\gamma}}_s + \nabla f(\widetilde{\gamma}_s)\|^2}{2\underline{\sigma}^2(f(\widetilde{\gamma}_s))} \, ds &= \int_0^S \frac{\|\dot{\varphi}_s + \nabla f(\varphi_s)\|^2}{2\underline{\sigma}^2 f(\varphi_s))} \, ds \\
&\geq \int_0^S \bar{\mathcal{L}}(\varphi_s, \dot{\varphi}_s + \nabla f(\varphi_s)) \, ds \\
&= \mathcal{S}_S(\varphi) \\
&\geq \mathcal{B}(x, x') \,, \tag{E.29}
\end{aligned}
$$

where we used Assumption 11 in the first inequality with $v \leftarrow \dot{\varphi}_s + \nabla f(\varphi_s)$ whose norm satisfies the condition of Assumption 11 by (E.25). Plugging (E.29) and (E.29) into (E.27) and then into (E.26) yields the desired result. ∎

### E.3 Graph representation

Let us reuse the notation of Appendix D. In Appendix D, we consider graphs over the set of vertices $\mathcal{V} = \{1, \ldots, N_{\mathrm{crit}}\}$ and the set of edges $(\mathcal{V} \setminus \mathcal{Q}) \times \mathcal{V}$ where $\mathcal{Q} = \{1, \ldots, N_{\mathrm{targ}}\}$.

Let us consider a set of particular edges $\widetilde{\mathcal{E}} \subset (\mathcal{V} \setminus \mathcal{Q}) \times \mathcal{V}$ such that

$$\widetilde{\mathcal{E}} := \left\{ i \to j \in (\mathcal{V} \setminus \mathcal{Q}) \times \mathcal{V} : i \neq j, \, \mathrm{cl}\,\mathrm{Attr}(\mathcal{C}_i) \cap \mathrm{cl}\,\mathrm{Attr}(\mathcal{C}_j) \neq \emptyset \right\} \,. \tag{E.30}$$

Note that by Lemma B.8, we have that $\mathrm{Attr}(\mathcal{C}_i)$, $i \in \{1, \ldots, N_{\mathrm{crit}}\}$ is a partition of $\mathbb{R}^d$. Therefore, $(\mathcal{V}, \widetilde{\mathcal{E}})$ still satisfies that, from every $i \in \mathcal{V}$ there is a path that leads to $\mathcal{Q}$.

**Lemma E.5.** *For any $i \to j \in \widetilde{\mathcal{E}}$, let us define*

$$x_{i,j} \in \underset{x \in \mathrm{cl}\,\mathrm{Attr}(\mathcal{C}_i) \cap \mathrm{cl}\,\mathrm{Attr}(\mathcal{C}_j)}{\arg\min} f(x) \,. \tag{E.31}$$

*Then, under Assumption 11, if $X$ is large enough so that it contains the bounded set*

$$\left\{ x \in \mathbb{R}^d : f(x) \leq \max_{i \in \mathcal{V} \setminus \mathcal{Q}, j \in \mathcal{V}, i \neq j} f(x_{i,j}) + 1 \right\} \,, \tag{E.32}$$

*for any $i \to j \in \widetilde{\mathcal{E}}$, it holds that*

$$C_{i,j} \leq \overline{U}(x_{i,j}) - \overline{U}(\mathcal{C}_i) \,. \tag{E.33}$$

*Proof.* Note that the coercivity of $f$ (Assumption 5) ensure that $x_{i,j}$ is well-defined and the set of (E.32) is bounded.

Take $i \in \mathcal{V} \setminus \mathcal{Q}$, $j \in \mathcal{V}, i \neq j$. We first consider the case where $\nabla f(x_{i,j}) \neq 0$. Fix $\varepsilon > 0$.

By Lemma D.15, we have that there exists $\delta_0 > 0$ such that, for any $\delta \in (0, \delta_0)$,

$$C_{i,j} \leq C_{i,j}^\delta + \varepsilon \,. \tag{E.34}$$

At the potential cost of reducing $\delta_0 > 0$, assume that, for any $x \in \mathbb{B}(x_{i,j}, \delta_0/2)$, it holds that

$$a \leq \|\nabla f(x)\| \leq b \quad \text{and} \quad \overline{\sigma}_\infty^2(f(x)) \geq c \,, \tag{E.35}$$

for some $0 < a \leq b$ and $c > 0$. Moreover, by continuity of $f$ and $\overline{U}$, also assume that $\delta_0$ is small enough so that, for any $x \in \mathbb{B}(x_{i,j}, \delta_0/2)$, it holds that

$$\overline{U}(x) \leq \overline{U}(x_{i,j}) + \varepsilon \tag{E.36}$$

and,

$$f(x) \leq f(x_{i,j}) + 1 \,. \tag{E.37}$$

Finally, also assume that $\delta_0$ is small enough such that, for any $x \in \mathcal{U}_\delta(\mathcal{C}_i)$, it holds that

$$\overline{U}(x) \geq \overline{U}(\mathcal{C}_i) - \varepsilon. \tag{E.38}$$

Fix $\delta \in (0, \delta_0)$ such that $\delta \leq \varepsilon$. Now, since $x_{i,j} \in \text{cl Attr}(\mathcal{C}_i) \cap \text{cl Attr}(\mathcal{C}_j)$, there exist $x_i \in \text{Attr}(\mathcal{C}_i) \cap \mathbb{B}(x_{i,j}, \delta/2)$ and $x_j \in \text{Attr}(\mathcal{C}_j) \cap \mathbb{B}(x_{i,j}, \delta/2)$. By construction, the gradient flows started at $x_i$ and $x_j$ converge respectively to $\mathcal{C}_i$ and $\mathcal{C}_j$. Hence, there exists $T \geq 1$ integer such that $\Theta_{T-\alpha/2}(x_i) \in \mathcal{U}_\delta(\mathcal{C}_i)$ and $\Theta_{T-\alpha/2}(x_j) \in \mathcal{U}_\delta(\mathcal{C}_j)$ for all $\alpha := \delta/a$.

Let us now estimates the action costs of the paths $(\Theta_{T-\alpha/2-t}(x_i))_{t \in [0, T-\alpha/2]}$ and $(\Theta_t(x_j))_{t \in [0, T-\alpha/2]}$. By (E.37) and definition of the flow, the whole path $(\Theta_{T-\alpha/2-t}(x_i))_{t \in [0, T-\alpha/2]}$ is contained in (E.32) and therefore in $X$. Therefore, by Assumption 11, we have that

$$\begin{aligned}
\mathcal{S}_{T-\alpha/2}(\Theta_{T-\alpha/2-\cdot}(x_i)) &= \int_0^{T-\alpha/2} \mathcal{L}(\Theta_{T-\alpha/2-t}(x_i), \nabla f(\Theta_{T-\alpha/2-t}(x_i)))\, dt \\
&= \int_0^{T-\alpha/2} \bar{\mathcal{L}}(\Theta_{T-\alpha/2-t}(x_i), 2\nabla f(\Theta_{T-\alpha/2-t}(x_i)))\, dt \\
&\leq \int_0^{T-\alpha/2} \frac{4\|\nabla f(\Theta_{T-\alpha/2-t}(x_i))\|^2}{2\overline{\sigma}^2(f(\Theta_{T-\alpha/2-t}(x_i)))}\, dt\,,
\end{aligned} \tag{E.39}$$

where we used Assumption 11 in the last inequality. Rewriting this integral we obtain,

$$\begin{aligned}
\mathcal{S}_{T-\alpha/2}(\Theta_{T-\alpha/2-\cdot}(x_i)) &\leq \int_0^{T-\alpha/2} \frac{2\langle \nabla f(\Theta_{T-\alpha/2-t}(x_i)), \dot{\Theta}_{T-\alpha/2-t}(x_i) \rangle}{\overline{\sigma}^2(f(\Theta_{T-\alpha/2-t}(x_i)))}\, dt \\
&= \int_0^{T-\alpha/2} 2\langle \nabla \overline{U}(\Theta_{T-\alpha/2-t}(x_i)), \dot{\Theta}_{T-\alpha/2-t}(x_i) \rangle\, dt \\
&= \overline{U}(x_i) - \overline{U}(\Theta_{T-\alpha/2}(x_i)) \\
&\leq \overline{U}(x_{i,j}) - \overline{U}(\mathcal{C}_i) + 2\varepsilon\,,
\end{aligned} \tag{E.40}$$

by (E.36) and (E.38).

For the path $(\Theta_t(x_j))_{t \in [0, T-\alpha/2]}$, since it is a trajectory of the gradient flow, we have that

$$\mathcal{S}_{T-\alpha/2}(\Theta_\cdot(x_j)) = 0. \tag{E.41}$$

Finally, let us consider the path $(\gamma_t)_{t \in [0, \alpha]}$ defined by $\gamma_t = x_i + \alpha^{-1} t(x_j - x_i)$. We have $\dot{\gamma}_t = \alpha^{-1}(x_j - x_i)$ so that $\|\dot{\gamma}_t\| \leq \delta/\alpha = a \leq \|\nabla f(\gamma_t)\|$ by (E.35). Moreover, (E.37) ensures that $\gamma_t \in X$ for all $t \in [0, \alpha]$.

Therefore, by Assumption 11, we have that

$$\begin{aligned}
\mathcal{S}_\alpha(\gamma) &= \int_0^\alpha \mathcal{L}(\gamma_t, \dot{\gamma}_t)\, dt \\
&= \int_0^\alpha \bar{\mathcal{L}}(\gamma_t, \nabla f(\gamma_t)\dot{\gamma}_t)\, dt \\
&\leq \int_0^\alpha \frac{\|\dot{\gamma}_t + \nabla f(\gamma_t)\|^2}{2\overline{\sigma}^2(f(\gamma_t))}\, dt \\
&\leq \alpha \frac{(a+b)^2}{2c} \\
&= \frac{\delta(a+b)^2}{2ac} \\
&\leq \frac{\varepsilon(a+b)^2}{2ac}\,.
\end{aligned} \tag{E.42}$$

Putting the three paths $(\Theta_{T-\alpha/2-\cdot}(x_i))_{t\in[0,T-\alpha/2]}$, $(\gamma_t)_{t\in[0,\alpha]}$ and $(\Theta_t(x_j))_{t\in[0,T-\alpha/2]}$ together, we obtain a path that goes from $\mathcal{U}_\delta(\mathcal{C}_i)$ to $\mathcal{U}_\delta(\mathcal{C}_j)$ in integer time $2T$ and whose action cost is upper-bounded by

$$\overline{U}(x_{i,j}) - \overline{U}(\mathcal{C}_i) + 2\varepsilon + \frac{\varepsilon(a+b)^2}{2ac} \,. \tag{E.43}$$

Combined with (E.34), this yields

$$C_{i,j} \leq \overline{U}(x_{i,j}) - \overline{U}(\mathcal{C}_i) + 3\varepsilon + \frac{\varepsilon(a+b)^2}{2ac} \,, \tag{E.44}$$

which yields the result.

Let us now briefly discuss the case where $\nabla f(x_{i,j}) = 0$. In this case, $x_{i,j}$ belongs to some $\mathcal{C}_l$, $l \in \mathcal{V}$. To adapt the proof above, it suffices to require that $x_i$ and $x_j$ are close enough to $x_{i,j}$ so that they belong to $\mathcal{W}_\varepsilon(\mathcal{C}_l)$ (give by Lemma B.6) and to use Lemma B.7 to build a path from $x_i$ to $x_j$ with cost at most $\varepsilon$. ∎

We provide an iterated version of Lemma E.5 that will be convenient in the following.

**Corollary E.1.** *For any $i \in \mathcal{V} \setminus \mathcal{Q}$, $j \in \mathcal{V}$, define*

$$\overline{C}_{i,j} := \min\left\{ \sum_{n=0}^{N-1} \overline{U}(x_{j_n,j_{n+1}}) - \overline{U}(\mathcal{C}_{j_n}) : j_0 = i,\ j_N = j,\ j_n \to j_{n+1} \in \widetilde{\mathcal{E}} \right\}. \tag{E.45}$$

*Then, under the same assumptions as Lemma E.5, it holds that*

$$C_{i,j} \leq \overline{C}_{i,j} \,. \tag{E.46}$$

Let us now specify how big enough we will need the set $X$ from Assumption 11 to be.

**Lemma E.6.** *Define*

$$C_\infty := \max_{i,j \in \mathcal{V} \setminus \mathcal{Q}} C_{i,j} + 1 < +\infty \,. \tag{E.47}$$

*Then, if $X$ is large enough so that it contains the bounded set*

$$\left\{ x \in \mathbb{R}^d : U_\infty(x) \leq \sup_{i \in \mathcal{V} \setminus \mathcal{Q}, x \in \mathcal{C}_i} \overline{U}(x) + C_\infty \right\}, \tag{E.48}$$

*then, for any $i, j \in \mathcal{V} \setminus \mathcal{Q}$, it holds that $C_{i,j}$ is equal to*

$$\inf\left\{ \mathcal{S}_T(\gamma) : \gamma \in \mathcal{C}([0,T]),\ T \in \mathbb{N},\ \gamma_0 \in \mathcal{C}_i,\ \gamma_T \in \mathcal{C}_j,\ \{\gamma(t) : t \in [0,T]\} \subset X \right\}. \tag{E.49}$$

*Proof.* Note that $C_\infty$ is finite by Assumption 9. The rest of the result is then a direct consequence of Lemma D.12. ∎

We can now combine Lemma E.4 with Lemma E.6 to obtain the following result.

**Lemma E.7.** *Under Assumption 11, if $X$ is large enough so that it satisfies the requirement of Lemma E.6, then for any $i, j \in \mathcal{V} \setminus \mathcal{Q}$, $i \neq j$, it holds that*

$$C_{j,i} - C_{i,j} \leq \Delta C_{j,i} + (1-\Delta)(\overline{U}_i - \overline{U}_j) \,, \tag{E.50}$$

*where $\Delta \in [0,1]$ satisfies*

$$\Delta \geq \sup_{x \in X} 1 - \frac{\underline{\sigma}^2(f(x))}{\overline{\sigma}^2(f(x))} \,. \tag{E.51}$$

*Proof.* Lemma E.6 allows us to consider only paths that are contained in $X$ to compute $C_{i,j}$. Taking the infimum over such paths in Lemma E.4 yields:

$$C_{i,j} \geq (1-\Delta)(C_{j,i} + (\overline{U}_j - \overline{U}_i)) \,. \tag{E.52}$$

Rearranging the terms, we obtain the result. ∎

To simplify the notation for the following lemma, let us define, for $i \in \mathcal{V} \setminus \mathcal{Q}, j \in \mathcal{V}, i \neq j$,

$$\overline{U}_{i,j} := \overline{U}(x_{i,j}) \quad \text{and} \quad \overline{U}_j := \overline{U}(\mathcal{C}_j). \tag{E.53}$$

**Lemma E.8.** *Assume that Assumption 11 holds with X satisfying the assumptions of Lemmas E.5 and E.6.*

*For any $i_0 \in \mathcal{V} \setminus \mathcal{Q}, g \in \mathcal{G}(i_0 \twoheadrightarrow \mathcal{Q})$, there exists $i \in \mathcal{V} \setminus \mathcal{Q}$ such that there is a path from $i_0$ to $i$ in $g$ and there is $g' \in \mathcal{G}$ such that the cost difference between $g$ and $g'$*

$$\sum_{k \to l \in g'} C_{k,l} - \sum_{k \to l \in g} C_{k,l} \tag{E.54}$$

*is upper-bounded by*

$$\min \left\{ \max_{l=0,\dots,p-1} \max \left( \overline{U}_{j_l,j_{l+1}} - \overline{U}_{j_l}, \overline{U}_{j_l,j_{l+1}} - \overline{U}_i \right) + 2\Delta(N_{crit}^2 + 1) \sup_X \overline{U} : \right.$$

$$\left. j_0 = i, j_p \in \mathcal{Q}, j_l \in \mathcal{V} \setminus \mathcal{Q}, j_l \to j_{l+1} \in \widetilde{\mathcal{E}}, l = 0, \dots, p-1 \right\}, \tag{E.55}$$

*where $\Delta \in [0, 1]$ satisfies*

$$\Delta \geq \sup_{x \in X} 1 - \frac{\underline{\sigma}^2(f(x))}{\overline{\sigma}^2(f(x))}. \tag{E.56}$$

*Proof.* By definition of $\mathcal{G}(i_0 \twoheadrightarrow \mathcal{Q})$ (Appendix D.1), there exists $i \in \mathcal{V} \setminus \mathcal{Q}, j \in \mathcal{V}, i \neq j$ such that if we were to add the edge $i \to j$ to $g$, it would be a $\mathcal{Q}$-graph. In other words, $i \to j$ is not in $g$ and there are paths from $i_0 \to i$ and $j \to \mathcal{Q}$ in $g$.

Now, take consider a path that goes from $i$ to $j$ with only edges from $\widetilde{\mathcal{E}}$: $j_0 = i, j_p = j$ and $j_l \in \mathcal{V} \setminus \mathcal{Q}, j_l \to j_{l+1} \in \widetilde{\mathcal{E}}$, $l = 0, \dots, p-1$.

Define $k \geq 0$ to be the smallest integer such that there is a path from $j_k$ to $\mathcal{Q}$ in $g$. Such a $k$ exists since there is a path from $j$ to $\mathcal{Q}$ in $g$, and $k \geq 1$ since there is no path from $i$ to $\mathcal{Q}$ in $g$. In particular, there is no path from $j_{k-1}$ to $\mathcal{Q}$ in $g$. From the definition of the graphs $\mathcal{G}(i_0 \twoheadrightarrow \mathcal{Q})$, this implies either there is a path from $i$ to $j_{k-1}$ in $g$ or a path from $j_{k-1}$ to $i$ in $g$.

- If there is a path from $i$ to $j_{k-1}$ in $g$, then we consider the graph $g' \in \mathcal{G}(\mathcal{Q})$ obtained by adding the edge $j_{k-1} \to j_k$ to $g$. Since $j_{k-1} \to j_k$ is in $\widetilde{\mathcal{E}}$, by Lemma E.5, the cost of adding that edge is bounded as

$$C_{j_{k-1},j_k} \leq \overline{U}_{j_{k-1},j_k} - \overline{U}_{j_{k-1}}, \tag{E.57}$$

  in which case (E.55) is immediately satisfied.

- If there is a path from $j_{k-1}$ to $i$ in $g$, then we consider the graph $g' \in \mathcal{G}(\mathcal{Q})$ obtained by reversing the path from $j_{k-1}$ to $i$ in $g$ and adding the edge $j_{k-1} \to j_k$. Let us denote by $m_0 = j_{k-1}, m_q = i$ and $m_l \in \mathcal{V} \setminus \mathcal{Q}, m_l \to m_{l+1} \in g, l = 0, \dots, q-1$ the path from $j_{k-1}$ to $i$. By Lemmas E.5 and E.7 the cost of reversing that path and adding the edge $j_{k-1} \to j_k$ is upper-bounded by

$$\sum_{l=0}^{q-1} \left( C_{m_{l+1},m_l} - \sum_{l=0}^{p-1} C_{m_l,m_{l+1}} \right) + C_{j_{k-1},j_k} \leq \sum_{l=0}^{q-1} \left( \Delta C_{m_{l+1},m_l} + (1-\Delta)(\overline{U}_{m_l} - \overline{U}_{m_{l+1}}) \right) + \overline{U}_{j_{k-1},j_k} - \overline{U}_{j_{k-1}}$$

$$= \sum_{l=0}^{q-1} \Delta C_{m_{l+1},m_l} + (1-\Delta)(\overline{U}_{j_{k-1}} - \overline{U}_i) + \overline{U}_{j_{k-1},j_k} - \overline{U}_{j_{k-1}}. \tag{E.58}$$

Now, Corollary E.1 ensures that the sum $\sum_{l=0}^{q-1} \Delta C_{m_{l+1},m_l}$ is upper-bounded by $2N_{crit}^2 \Delta \sup_X \overline{U}$ so we obtain that the RHS of (E.58) is upper-bounded by

$$2N_{crit}^2 \Delta \sup_X \overline{U} + \overline{U}_{j_{k-1},j_k} - \overline{U}_i + 2\Delta \sup_X \overline{U}, \tag{E.59}$$

which concludes the proof.

■

To obtain the main result, we will require the following lemma.

**Lemma E.9.** *Under Assumption 11, if X is large enough so that it satisfies the assumptions of Lemma E.6, then for any $i, j \in \mathcal{V} \setminus \mathcal{Q}$, $i \neq j$, it holds that For any $i \in \mathcal{V} \setminus \mathcal{Q}$, $j \in \mathcal{V}$, $i \neq j$, it holds that*

$$C_{i,j} \geq \underline{C}_{i,j} := \inf\left\{\sup_{s<t}\left(\underline{U}(\gamma_t) - \underline{U}(\gamma_s)\right) : \gamma \in \mathcal{C}([0,T]), T \in \mathbb{N}, \gamma_0 \in \mathcal{C}_i, \gamma_T \in \mathcal{C}_j\right\}. \tag{E.60}$$

*Proof.* This lemma follows the same proof as Lemma D.10 with the additional use of Lemma E.6 to be able to leverage Assumption 11. ∎

We are now finally ready to prove the main result.

**Theorem E.1.** *Assume that Assumption 11 holds with X satisfying the assumptions of Lemmas E.5 and E.6.*

*For any $i_0 \in \mathcal{V} \setminus \mathcal{Q}$, define*

$$r := \min\left\{\sum_{k \to l \in g} \overline{C}_{k,l} : g \in \mathcal{G}(i_0 \not\rightarrow \mathcal{Q})\right\}. \tag{E.61}$$

*there exists $i \in \mathcal{V} \setminus \mathcal{Q}$ such that $\underline{C}_{i_0,i} \leq r$ such that $E(\mathcal{Q} \mid i_0)$ is upper-bounded by*

$$\min\left\{\max_{l=0,\ldots,p-1} \max\left(\overline{U}_{j_l,j_{l+1}} - \overline{U}_{j_l}, \overline{U}_{j_l,j_{l+1}} - \overline{U}_i\right) + 2\Delta(N_{crit}^2 + 1)\sup_X \overline{U} : \right.$$
$$\left. j_0 = i, j_p \in \mathcal{Q}, j_l \in \mathcal{V} \setminus \mathcal{Q}, j_l \to j_{l+1} \in \widetilde{\mathcal{E}}, l = 0,\ldots,p-1\right\}, \tag{E.62}$$

*where $\Delta \in [0,1]$ satisfies*

$$\Delta \geq \sup_{x \in X} 1 - \frac{\underline{\sigma}^2(f(x))}{\overline{\sigma}^2(f(x))}. \tag{E.63}$$

*Proof.* By Lemma D.16, $E(\mathcal{Q} \mid i_0)$ is upper-bounded as

$$E(\mathcal{Q} \mid i_0) \leq \min\left\{\sum_{k \to l \in g} C_{k,l} : g \in \mathcal{G}(\mathcal{Q})\right\} - \min\left\{\sum_{k \to l \in g} C_{k,l} : g \in \mathcal{G}(i_0 \not\rightarrow \mathcal{Q})\right\}. \tag{E.64}$$

By Corollary E.1, we have that

$$\min\left\{\sum_{k \to l \in g} C_{k,l} : g \in \mathcal{G}(i_0 \not\rightarrow \mathcal{Q})\right\} \leq \min\left\{\sum_{k \to l \in g} \overline{C}_{k,l} : g \in \mathcal{G}(i_0 \not\rightarrow \mathcal{Q})\right\} = r. \tag{E.65}$$

Hence, take $g \in \arg\min\left\{\sum_{k \to l \in g} C_{k,l} : g \in \mathcal{G}(\mathcal{Q} \twoheadrightarrow \mathcal{Q})\right\}$ and it must satisfy

$$\sum_{k \to l \in g} C_{k,l} \leq r. \tag{E.66}$$

Apply Lemma E.8 to obtain that there exists $i \in \mathcal{V} \setminus \mathcal{Q}$ such that there is a path from $i_0$ to $i$ in $g$ and there is $g' \in \mathcal{G}$ such that the cost difference between $g$ and $g'$ is bounded by (E.62). Combined with (E.64), this yields that $E(\mathcal{Q} \mid i_0)$ is upper-bounded by (E.62).

Finally, denote by $j_0 = i_0$, $j_p = i$ and $j_l \in \mathcal{V} \setminus \mathcal{Q}$, $j_l \to j_{l+1} \in g$, $l = 0,\ldots,p-1$ the path from $i_0$ to $i$ in $g$. (E.66) ensures that

$$r \geq \sum_{l=0}^{p-1} C_{j_l,j_{l+1}} \geq \underline{C}_{i_0,i}, \tag{E.67}$$

which, combined with Lemma E.9, implies that $\underline{C}_{i_0,i} \leq r$. ∎

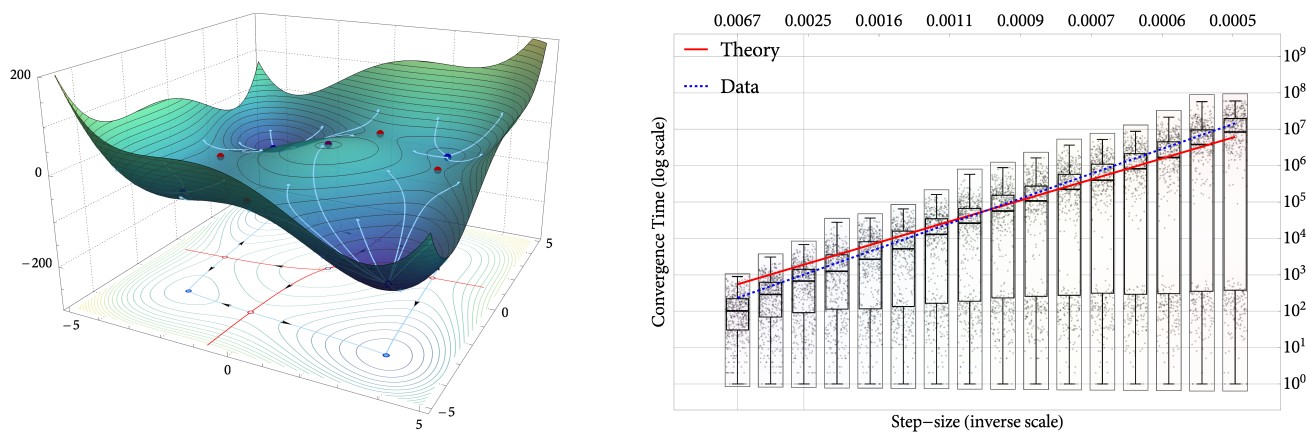

**(a)** Loss landscape and hitting time statistics for the Styblinski–Tang function (F.1a).

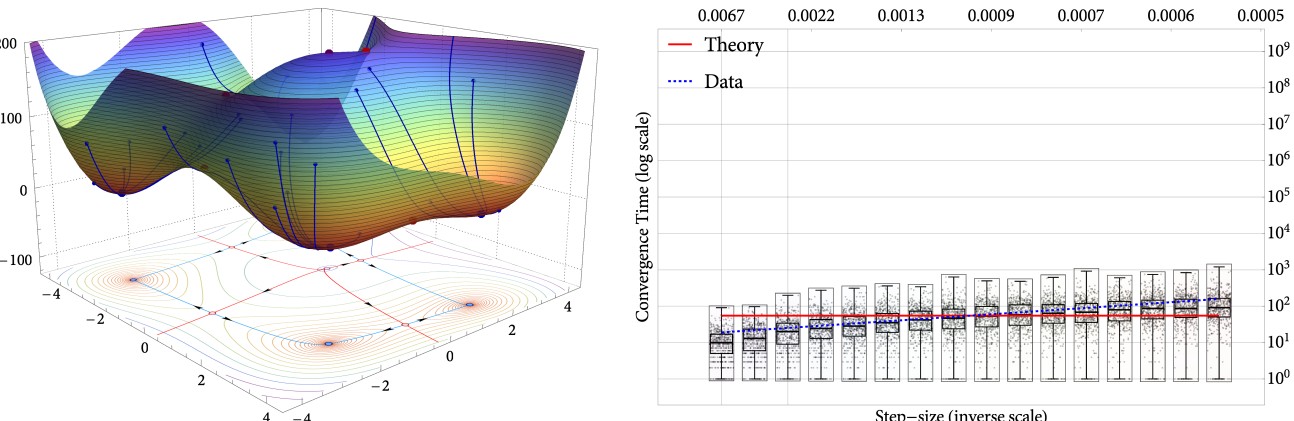

**(b)** Loss landscape and hitting time statistics for the Himmelblau function (F.1b).

**Figure 4:** Visualization and hitting time statistics for the Styblinski–Tang (top) and Himmelblau functions (bottom). In both cases, the loss landscape is superimposed on the stable and unstable manifolds of the function's critical points, which define the edges of its transition graph (the arrows indicate the direction of the gradient flow, and hence the transitions which come at no cost).

## F    Numerical validation

In this last appendix, we provide some more numerical experiments as validation of the estimates provided by Theorems 1 and 2. For concreteness, we consider the following multimodal test functions:

(*a*) *The Styblinski–Tang function:*

$$f(x_1, x_2) = \frac{1}{2} \sum_{i=1}^{d} (x_i^4 - 16x_i^2 + 5x_i) \tag{F.1a}$$

(*b*) *The Himmelblau function:*

$$f(x_1, x_2) = (x_1^2 + x_2 - 11)^2 + (x_1 + x_2^2 - 7)^2 \tag{F.1b}$$

For the Styblinski–Tang function, we treat the 2-dimensional case in order to facilitate comparisons with the other examples, and for ease of visualization.

In Fig. 4, we plot the landscape of $f$ for each of the above cases, as well as the runtime statistics of (SGD) in each case. As in Fig. 2, we considered different values of the algorithm's learning rate $\eta$, and for each value of $\eta$, we performed 500 runs of (SGD) with Gaussian noise ($\sigma = 50$). Then, for each "particle" of (SGD), we recorded the number of iterations required to reach the vicinity of $p_1$ (within $\delta = 10^{-2}$) from the same, fixed initialization. The resulting boxplots are displayed in Fig. 4 in log-inverse scale, with a point density overlay to illustrate the full empirical distribution of $\tau$ for every $\eta$.

In both cases, we plotted the theoretical prediction of Theorem 1 (labeled "Theory"), against a linear regression plot of the observed runtime data (labeled "Data").[13] For the Styblinski–Tang function, we observe a situation similar to the three-hump-camel example of Fig. 2: Both theory and experiment indicate that the global convergence time $\tau$ of (SGD) scales exponentially in the inverse of the algorithm's step-size $\eta$, and there is excellent agreement between theory and experiment ($R^2 = 0.973$ and $R^2 = 0.989$ respectively). By contrast, in the Himmelblau case, runtimes are significantly slower, and the slope of the theoretical estimator is 0: This is because the Himmelblau function does not have any spurious minima, so $\tau$ enters the subexponential regime; this is confirmed by the fact that the algorithm's runtime is several orders of magnitude smaller, with strong agreeemnt between our theoretical estimate and the numerically measured values ($R^2 = 0.913$ and $R^2 = 0.976$ respectively).

---

[13]In more detail, we plotted a linear fit with slope given by Theorem 1.

# G  Auxiliary results

## G.1  Truncated Gaussian distribution

Let us restate two technical lemmas from [5] on which we rely. The first one provides bounds on the Hamiltonian and Lagrangian of a truncated Gaussian distribution.

**Lemma G.1** ([5, Lem. F.2]). *Consider $X \sim \mathcal{N}(0, \sigma^2 I)$ a multivariate Gaussian distribution with $\sigma^2 > 0$. For $R > 0$, define the truncated Gaussian random variable (r.v.) $X_R$ by conditioning $X$ to the ball $\overline{\mathbb{B}}(0, R)$. Define*

$$\bar{\mathcal{H}}(p) := \log \mathbb{E}\left[ e^{\langle p, X_R \rangle} \right] \tag{G.1}$$

$$\bar{\mathcal{L}}(p) := \bar{\mathcal{H}}^*(p), \tag{G.2}$$

*and*

$$E(\sigma^2, R) := e^{-\frac{R^2}{16\sigma^2}} 2^{d+3}(d + 1) \tag{G.3}$$

*and assume that $R > 0$ satisfies*

$$R \geq 4\sigma \sqrt{(d + 3) \log 2 + \log(d + 1)}. \tag{G.4}$$

*Then, for any $p \in \mathbb{R}^d$ such that $\|p\| \leq \frac{R}{2\sigma^2}$, $v \in \mathbb{R}^d$ such that $\|v\| \leq \frac{R}{4}$, it holds that*

$$\left(1 - E(\sigma^2, R)\right) \frac{\sigma^2 \|p\|^2}{2} \leq \bar{\mathcal{H}}(p) \leq \left(1 + E(\sigma^2, R)\right) \frac{\sigma^2 \|p\|^2}{2} \tag{G.5}$$

$$\left(1 - 2E(\sigma^2, R)\right) \frac{\|v\|^2}{2\sigma^2} \leq \bar{\mathcal{L}}(v) \leq \left(1 + 2E(\sigma^2, R)\right) \frac{\|v\|^2}{2\sigma^2}. \tag{G.6}$$

We will also require the following technical lemma.

**Lemma G.2** ([5, Lem. F.3]). *Consider $v, w \in \mathbb{R}^d$ such that $0 < \|w\| \leq \frac{\mu R}{2}$ for some $R, \mu > 0$. Define,*

$$f(u) = \sup_{p \in \mathbb{R}^d : \|p\| \leq R} \langle p, u \rangle - \frac{\mu}{2} \|p\|^2, \tag{G.7}$$

*then, with $\lambda = \frac{\|v\|}{\|w\|}$,*

$$\lambda f\left(\frac{v}{\lambda} + w\right) \leq f(v + w). \tag{G.8}$$

