# OpenReview forum: "The Global Convergence Time of Stochastic Gradient Descent in Non-Convex Landscapes: Sharp Estimates via Large Deviations"
_ICML.cc/2025/Conference — ICML 2025 poster_

### Official Review · Reviewer_Qj72 · 2025-03-09

**Overall Recommendation:** 4

**Summary:**

The paper studies the global convergence time of stochastic gradient descent (SGD) in non-convex optimisation landscapes. By employing tools of large deviation theory and randomly perturbed dynamical systems, the authors provide upper and lower bounds on the global convergence time. These bounds are dominated by the most costly paths (in the sense of an energy function depending on the global geometry of the landscape and on the initialisation of the algorithm) the algorithm might take in the loss landscape.


**Update after rebuttal**

I thank the authors for the detailed response. I consider the paper a solid and relevant contribution, so I recommend its acceptance.

**Claims And Evidence:**

Although I have not checked the proofs in details, the claims are made based on well-established methods in the literature. Additionally, the authors provide a qualitative verification of their results using the three-hump camel function. Perhaps other benchmark functions like the Himmelblau function could be used to further validate the results

**Essential References Not Discussed:**

The related literature is well-cited.

**Experimental Designs Or Analyses:**

Since the result is a bound on the global convergence time, there is not a direct experimental way to validate the result. Nevertheless, the authors provide a qualitative verification using the three-hump camel function. In Fig. 1, it is clear that the algorithm spends most of the time around critical points, which is consistent with the main result of the paper.

**Methods And Evaluation Criteria:**

The authors make use of large deviation theory and randomly perturbed dynamical systems to provide bounds on the global convergence time of SGD. These methods have become popular within the machine learning community to study the convergence of stochastic optimisation algorithms and are well-suited for the problem at hand. The authors also provide a qualitative verification of their results using the three-hump camel function.

**Other Comments Or Suggestions:**

I have not found any typos in the paper.

**Other Strengths And Weaknesses:**

Strengths:

1) The paper is well-written and discusses the results in a clear and concise manner.
2) It addresses an important problem in the field of optimisation and machine learning and uses well-established methods to provide bounds on the global convergence time of SGD.

Weaknesses:

1) See questions for authors below.

**Questions For Authors:**

1)  Both Theorems 1 and 2 are valid under the conditions that the learning rate $\eta$ should be "small enough". What exactly do the authors mean by that? I believe this information is given in the appendices, which, unfortunately, I could not go through. In any case, why not precisely state in the main text how small the learning rate should be?

2) In Appendix A3, the diffusion approximation studying escaping from minima of [Mori et al. 2022] is cited. Assuming the system takes more time to escape from minima than from saddle points, I would expect somehow the global convergence time to be dominated by the time to escape from minima ([Mori et al. 2022] claims one should consider a log exponential barrier between minima and saddle instead of a linear barrier). Could the authors please comment on this? Clarifying if they see any connection between their work and the results of [Mori et al. 2022] would be very interesting.

3) Assumption 2(c) provides a bound that does not depend on $x$. Nevertheless, in Assumptions 7-8, the authors relax this assumption, letting the bound depend on $f(x)$. So, the general picture related to the global convergence time does not crucially depend on the dependence of the noise with $x$? I ask because one of the big challenges in studying the effects of noise in SGD throughout the optimisation dynamics is how to deal with the dependence of the noise on the current iterate.

4) Does Assumption 4 change if one considers Assumptions 7-8 instead of Assumptions 2(c)?

**Relation To Broader Scientific Literature:**

The paper is well-positioned within the broader scientific literature. The authors provide a comprehensive review of the related work, citing not only closely related papers but also works that are more tangentially related.

**Theoretical Claims:**

I have not checked the proofs in detail.

---

> ### Author Rebuttal · Authors · 2025-04-01
>
> Thank you for your time, input, and positive evaluation! We reply to your remarks and questions below:
>
> - **Use of other benchmarks.** Sure thing! We took advantage of the rebuttal period to run numerical experiments on three standard non-convex benchmarks: the three-humped camel benchmark (already mentioned in the paper), Styblinski-Tang, and the Himmelblau example that you suggested. In all examples, we computed the various energy barriers exactly, and our experiments showed an exceedingly good fit between theory and practice, cf. the anonymized link [here](https://tinyurl.com/2k5ay5c6).
>
> - **The precise meaning of "small enough".** We mean here that there exists some constant, positive $\eta_0$, depending on $\varepsilon$ and the characteristics of SGD via $f$ and $Z$, such that the statement of the theorem is valid for all $\eta ≤ \eta_0$. Our proofs yield an explicit constructive estimate for $\eta_0$, which means that one could, in principle, exhibit the explicit dependence of "small enough" on $\epsilon$ and and $f, Z$. However, the resulting expression is the composition of several intermediate expressions, bounds and estimates, making it rather unwieldy to present (especially as it depends on certain quantities that are introduced and analyzed in the appendix). For this reason, we did not include a more precise description in the main part of the paper; nonetheless, it is worth noting that our simulations show an excellent fit between theory and practice for standard step-size parameter ranges (between 10E-2 and 10E-4). We will make sure to take advantage of the extra page to explain this at the first revision opportunity.
>
> - **Escape times and Mori et al (2022).**  Excellent question, thanks for giving us the opportunity to discuss this. In fact, a simple way to see that escape times are not enough to obtain the global convergence time of SGD is already given by our introductory example in Section 3. Indeed, in this case, the escape time of $p_3$ would scale exponentially in $f(p_2) - f(p_3)$ (the depth of $p_3$) while the escape time of $p_5$ would scale exponentially in $f(p_4)  - f(p_5)$. However, the global convergence time of SGD as determined by (4) is greater than either of these two exponentials: indeed, our result takes into account that, when in the vicinity of $p_3$, SGD could transition to $p_5$ instead of $p_1$ an arbitrary number of times. Hence, characterizing escape times is not enough to describe the global convergence time of SGD in general.
>
>   Finally, regarding on the modelling of the noise by [Mori et al. 2022]: under their noise model, our general framework of assumptions 7-8 and 11 allow us to obtain analogues of Thm.3 that depend on logarithmic barriers instead of linear barriers.
>
> - **On Assumption 2(c).** In our generic bounds (Thm. 1-2), the global convergence time depends on the dependence of the noise with $x$ because the computation of the energy involves the Hamiltation $\mathcal{H}_G(x, \omega)$, which fully characterizes how the noise distribution depends on $x$. It is true that, in Section 5, for simplicity, we consider bounds which are relevant when $\underline{\sigma} \approx \bar{\sigma}$, i.e. when the magnitude of the noise is roughly constant. That being said, as mentionned above, our framework of Section 5 encompasses the noise model of [Mori et al. 2022] for Gaussian noise with variance $\sigma^2(f(x)) \propto f(x)$, we will make this clear in the revision.
>
> - **On Assumption 4.** Indeed, if one considers Assumptions 7-8 instead of Assumption 2(c), Assumption 4 becomes (E.1) in Assumption 11 in Section E.11, where $\underline{\sigma}$ now depends on $f(x)$. We will provide a reference to this assumption in Appendix B.1 for clarity.
>
> ---
> We thank you again for your constructive input and positive evaluation, and we look forward to any other comments you may have during the rebuttal phase.
>
> Kind regards,
>
> The authors

---

> > ### Comment · Reviewer_Qj72 · 2025-04-03
> >
> > I thank the authors for the detailed response. I consider the paper a solid and relevant contribution, so I recommend its acceptance.

---

> > > ### Author Response · Authors · 2025-04-04
> > >
> > > Thank you for your kind words and encouraging remarks - we are sincerely grateful for your time and your input!

---

### Official Review · Reviewer_WEdV · 2025-03-10

**Overall Recommendation:** 3

**Summary:**

This paper investigates the time required for SGD to attain the global minimum of a general, non-convex loss function. The authors approach this problem using large deviations theory and randomly perturbed dynamical systems and offer a exact characterization of SGD's hitting times with matching upper and lower bounds.

**Claims And Evidence:**

Yes. This is a theoretical paper and all of its claims are proved.

**Essential References Not Discussed:**

See weeknesses.

**Experimental Designs Or Analyses:**

No. I focused on reading the main text.

**Methods And Evaluation Criteria:**

N.A

**Other Comments Or Suggestions:**

See weaknesses.

**Other Strengths And Weaknesses:**

Strengths:

1. The paper is well-written, clear, and conceptually sound.
2. The idea of interpreting the convergence rate of SGD as the hitting time of a dynamical system reaching the set of global minimizers is both novel and interesting.

Weaknesses and Suggestions for Improvement:

My primary concern lies with the main result of the paper. The characterization of SGD's convergence time is based on the notion of energy, which quantifies the difficulty of reaching the minimizer set from the initialization. However, it is unclear whether bounding convergence by energy provides meaningful insights or practical improvements in understanding SGD's behavior.

Can the authors address the following points:

Practical Relevance of Energy – Are there cases where the energy can be explicitly computed and shown to be useful for improving training? Any theoretical or empirical exploration of this aspect would be valuable.

Comparison to Existing Bounds – Are there scenarios where your bound leads to an improvement over known convergence bounds for SGD? Providing such cases would clarify the significance of the result.

Implications for Lower Bounds – In the context of lower bounds, can you identify cases where your analysis proves that learning is inherently difficult? Establishing such results would strengthen the theoretical contribution.

Addressing these points would help clarify the practical and theoretical impact of your results.

**Questions For Authors:**

See weaknesses.

**Relation To Broader Scientific Literature:**

This paper characterizes the performance of SGD on general non-convex functions, a widely studied setting. The analysis relies on treating the training process as a dynamical system, a commonly used approach. However, to the best of my knowledge, the idea of analyzing the "hitting time" on the set of minimizers is novel.

**Theoretical Claims:**

No. I focused on reading the main text.

---

> ### Author Rebuttal · Authors · 2025-04-01
>
> Dear reviewer,
>
> Thank you for your input and your assessment that "the idea of analyzing the hitting time on the set of minimizers is novel". We reply to your questions and remarks below, and we will of course integrate this discussion in the paper at the first revision opportunity.
>
> - **Practrical relevance of the energy.**  First, we should emphasize that, unlike standard results on the convergence rate of SGD to criticality, our result is not an upper bound but a characterization, valid up to a prescribed tolerance level. In particular, the energy describes exactly which features of the loss landscape and the statistics of the noise end up affecting the convergence time of SGD, and in what way.
>
>   In terms of concrete take-aways, the results of Section 4 and the "deep-dive" of Section 5 show that the convergence time of SGD is exponential in a) the inverse of the step-size; b) the variance of the noise; and c) the depth of any spurious minima of $f$. Among others, this has explicit ramifications for (overparameterized) neural nets: in order to escape the exponential regime, the depth of spurious minimizers must scale logarithmically in the dimension of the problem; in turn, by known structural bounds on the loss surfaces of neural nets (such as the works we mention in at the end of section 5 and the works of Ben-Arous and co-authors), it is possible to estimate precisely the depth (or width, depending on the model) of a neural net required to attain the "shallow spurious minima" regime, which in turn would guarantee subexponential global convergence times.
>
>   Finally, from a broader theoretical perspective, we believe that the value of our results for the ML community goes beyond the explicit energy characterization of the global convergence time of SGD. Our analysis provides a toolbox for the analysis of a wide range of stochastic gradient-based algorithms, and is, in a sense, the first step through the door to obtain similar results for, say, SGD with oscillatory learning rate schedules, Adam, etc.
>
> - **Explicit computation of the energy.** We provide a series of explicit expressions and approximations for the energy in Section 5 (this is how we derived the concrete expression in Section 2). It is true that the current version of Section 5 is somewhat "crammed", but we would be happy to take advantage of the extra page to extend and enrich it at the first revision opportunity.
>
>   Incidentally, to showcase the tightness of our analysis, we used the rebuttal period to perform a series of numerical experiments on some standard global optimization benchmarks, which show an exceedingly good fit between theory and practice, see anonymized link [here](https://tinyurl.com/2k5ay5c6).
>
> - **Comparison to existing bounds.** We are not aware of any bounds in the literature (upper or lower) providing even an estimate regarding the global convergence time of SGD in bona fide non-convex objectives. Reviewer U6wN echoes this, as they state that "the results and developed mathematical tools presented in this paper is completely new". Do you have a specific reference in mind? If so, we would be happy to discuss and position it accordingly.
>
> - **Implications from lower bounds.** The example provided in Section 2 already provides such an implication: even though the "obstacle gap" $f(p_2) - f(p_5)$ is not encountered on the shortest path toward the global minimum of $f$, it is what controls the speed and efficiency of learning in this case. Thus, a slight increase this gap would end up having an exponential impact on the global convergence time of SGD, indicating the difficult of learning even in this simple example.
>
> ---
> We thank you again for your time, and we look forward to an open-minded discussion during the rebuttal phase.
>
> Kind regards,
>
> The authors

---

### Official Review · Reviewer_U6wN · 2025-03-12

**Overall Recommendation:** 5

**Summary:**

This paper answers a hard question in the optimization theory: how long it takes for the SGD to reach the global minimum of a general non-convex loss function. The answer is given in Theorem 1 and Theorem 2: the expected time is exponentially proportional to $E[Q]/\eta$. Later, the author characterizes $E[Q]$, which describes the attracting strength determined by many factors.

**Claims And Evidence:**

All claims made in this submission are well supported by clear justification and convincing evidence.

**Essential References Not Discussed:**

All essential references have been covered.

**Experimental Designs Or Analyses:**

NA

**Methods And Evaluation Criteria:**

NA

**Other Comments Or Suggestions:**

I may suggest the author to submit work similar to this submission to top-tier journals instead of conferences.

**Other Strengths And Weaknesses:**

Other strengths: This paper is well-written and well-organized. The gentle start is very helpful for understanding the main result of this paper.

**Questions For Authors:**

I have to admit that I didn't fully understand the whole picture of the proof. Even worse: I cannot tell if it is because the author didn't make this part clear or it is because I lack sufficient backgrounds in understanding the proof. If possible, I may hope the author could clarify several points:

1. The beginning part of Section 4 Analysis and results (more explicitly, everything before Theorem 1) seems to be some introduction and definitions of mathematical notations. They don't describe how to prove Theorem 1. Do I understand it correctly?

2. From my perspective, even if I know how these notations are defined, it is still hard to understand how the bounds in Theorem 1 are obtained. There is a large gap between knowing the proof technique and knowing how to prove it. Therefore, I may expect the author to include a brief proof sketch.

3. Does this work assume the number of critical sets is finite? On page 6, the vertices have the maximum number $N_{\text{all}}$. Is it possible to have infinite critical sets, e.g. $f(x) = \sin x$?

4. How should I understand Theorem 2? Does it simply take $x$ as a specific point $p$?

**Relation To Broader Scientific Literature:**

The results and developed mathematical tools presented in this paper is completely new. Though it has some overlappings with Ref. [4] on using the large deviation theory, this paper goes further and resolves an important but challenging problem in the optimization theory: how long the SGD takes to reach the global minimum. Unlike other finite-time analysis (they only consider the convergence to the stationary point, or make additional assumptions), this submission only requires sufficiently mild assumptions and presents much stronger conclusion.

[4] Azizian, W., Iutzeler, F., Malick, J., and Mertikopoulos, P.
What is the long-run distribution of stochastic gradient descent? A large deviations analysis. In ICML ’24: Proceedings of the 41st International Conference on Machine Learning, 2024.

**Theoretical Claims:**

I checked "An illustrative use case." to see if Assumption 1 will be satisfied. These claims should be correct.

I also checked other parts of the main paper, but I cannot simply conclude if these results are correct or not. These parts are beyond my mathematical knowledge; I will tend to believe these theoretical proofs are correct, as the results make sense by my guess.

---

> ### Author Rebuttal · Authors · 2025-04-01
>
> Dear reviewer,
>
> Thank you for your time, input, positive evaluation and appreciation! We reply to your remarks and questions below:
> - **On the beginning of Section 4.** Yes, we designed this part as a ramp-up to the technical apparatus required to state our results. As you mentionned, this is just a "gentle start [...] for understanding the main result of this paper."
> - **On the inclusion of a proof sketch.** We hesitated to go into a lengthier presentation of the technical trajectory leading to our results because of space constraints. We will be happy to take advantage of the first revision opportunity to include a sketch.
>
>   To lay the groundwork, we prepared a flowchart of the logical structure of the proof in the anonymized link [here](https://tinyurl.com/mb4r9udc). In short, the main steps are as follows:
>   1. Our framework is built on a a LDP for SGD (B.3): it yields precise estimations of the probability that SGD approximately follows an arbitrary continuous-time path.
>   2. In particular, it allows us (App. C) to conduct a precise study of the transitions of SGD between critical points: we carefully estimate both the transition probabilities and the transition times.
>   3. We then construct an induced chain that lives only the set of critical points and show that it captures the global convergence of properties. We may therefore restrict our focus to this induced chain which is a finite state space Markov chain.
>   4. Subsequently, in Appendix D, we leverage quantitaive results for finite state space Markov of such chains to obtain global convergence bounds for the induced chain that we then translate back to SGD. This last step is trickier for the case of lower-bound and we remedy that difficulty by exploiting the attracting strength assumptions.
>
> - **On infinite numbers of critical components.** Formally, yes, the number of critical components is required to be finite (though, of course, the number of critical points could be uncountable). Note however, that functions like $\sin x$ are ruled out by the gradient coercivity assumption. In fact, even though it is possible to construct examples with an infinite number of critical components that  satisfy all other assumptions, these examples are highly pathological, and thus not central to the considerations of our paper.
>
> - **On the role of $p$ in Theorem 2.**  In a way, yes. Theorem 2 essentially states that all initial points in the basin of $p$ will move to a small neighborhood of $p$ with overwhelmingly high probability. Because this transition takes relatively little time, the global convergence time from $x$ to the global minimum is approximately equal to the convergence time from $p$. This reduction in turn allows us to obtain the more detailed bounds of Section 5.
>
> ---
> We thank you again for your time and positive evaluation! Please let us know if any of the above points is not sufficiently clear.
>
> Kind regards,
>
> The authors

---

> > ### Comment · Reviewer_U6wN · 2025-04-02
> >
> > I appreciate the author's proof sketch and illustration of the appendix structure. It really helps me understand the proof (but to be honest, not too much. So, I expect the author to include more detailed sketch in the camera-ready version).
> >
> > I have multiple minor comments/questions after reading the proof sketch (and the appendix):
> > 1. (typo?) From Line 1059 to Line 1063, the author defines $B^\delta_{i,j}$ in (C.2); later, the author explains $\widetilde{B}^\delta_{i,j}$ will prove helpful. Are $\widetilde{B}^\delta_{i,j}$ defined somewhere else? I feel confused about it because both $\widetilde{B}^\delta_{i,j}$ and $B^\delta_{i,j}$  are used in Lemma C.1.
> > 2. In Appendix C.3 (Lemma C.3 and Lemma C.4), the author has evaluated the transition probability $Q_\mathcal{V}(x, \mathcal{V}_j)$. It means the probability of transitioning from the point $x$ to the critical component $\mathcal{V}_j$. I have multiple questions on that:
> >     * What is the $\mathcal{V}$ in the subscript of $Q_\mathcal{V}(x, \mathcal{V}_j)$? I would understand that $\mathcal{V}$ is defined as the union of $\mathcal{V}_j$. Do you need to use a different subscript in the proof? I notice that in Line 1301, the author is using $Q\_\mathcal{W}$ but I am not clear what it means.
> >     * I cannot clearly understand the difference between "C.3 Estimates of the transition probabilities" and "C.4 Accelerated induced chain". It seems that Lemma C.3 and Lemma C.4 have built a probability, then this probability is evaluated again for the accelerated induced chain.  Why is it needed to define the accelerated induced chain in Definition 10 and to derive the transition probability again?
> > 3. What is $K$ in (C.57)?
> > 4. (typo?) In Lemma D.3, should it be "For any $i\in V \setminus Q$"?  The author is using $i_0$ but the remaining of this lemma is using $i$.
> > 5. What is $\tau_Q$ in (D.37)? Has it been defined before? As we already have Theorem 4, why do we still need to have Theorem 5?

---

> > > ### Author Response · Authors · 2025-04-03
> > >
> > > Dear reviewer,
> > >
> > > Thank you for your interest in our work, we are sincerely grateful for your careful and meticulous input. We reply to your questions below, and we will make sure to correct any typos and inconsistencies of notation between the main text and the appendix - thanks again for the very detailed read.
> > >
> > > > 1. (typo?) From Line 1059 to Line 1063, the author defines $B^\delta_{i,j}$ in (C.2); later, the author explains $\tilde B^\delta_{i,j}$ will prove helpful. Are $\tilde B^\delta_{i,j}$ and $B^\delta_{i,j}$ defined somewhere else? I feel confused about it because both $\tilde B^\delta_{i,j}$ and $B^\delta_{i,j}$ are used in Lemma C.1.
> > >
> > > Thank you for catching this, $\tilde B^\delta_{i,j}$ is a typo, it should read $B^\delta_{i,j}$, which is defined in Def.8 eq. (C.2) just above of Lemma C.1.
> > >
> > > > 2. In Appendix C.3 (Lemma C.3 and Lemma C.4), the author has evaluated the transition probability $Q_\mathcal{V}(x, \mathcal{V}_j)$. It means the probability of transitioning from the point $x$ to the critical component $\mathcal{V}_j$.
> > >
> > > Correct: as defined in Def. 9 in App. C.2, $Q_\mathcal{V}(x, \mathcal{V}_j)$ is the probability that, starting from $x$, SGD (or more exaclty the sequence of B.17) SGD enters $\mathcal{V}_j$ before any other $\mathcal{V}_l$ for $l \neq j$.
> > >
> > > > I have multiple questions on that:
> > > >   - What is the $\mathcal V$ in the subscript of $Q_V(x,\mathcal V_j)$? I would understand that $\mathcal V$ is defined as the union of $\mathcal V_j$. Do you need to use a different subscript in the proof? I notice that in Line 1301, the author is using $Q_\mathcal W$ but I am not clear what it means.
> > >
> > > $\mathcal{V}$ is indeed defined as the union of the neighborhoods $\mathcal{V}_j$, as per the first line of Definition 9. For line 1301, yes, this should be $\mathcal{V}$ (there is no $\mathcal{W}$ defined here), good catch - thanks!
> > >
> > > >I cannot clearly understand the difference between "C.3 Estimates of the transition probabilities" and "C.4 Accelerated induced chain". It seems that Lemma C.3 and Lemma C.4 have built a probability, then this probability is evaluated again for the accelerated induced chain. Why is it needed to define the accelerated induced chain in Definition 10 and to derive the transition probability again?
> > >
> > > Thank you for your question. In section C.3, we study the transition probabilities of the induced chain $z_n$ (Def. 9). We quantify the following probabilities: if SGD starts at $\mathcal{V_i}$, what is the probability that it enters $\mathcal{V}_j$ before any other $\mathcal{V}_l$ for $l \neq j$? In section C.4, we consider a subsampled version of $z_n$ (Def. 10): we look only at the value of $z_n$ every $K$ steps. The probability that we quantify here is the probability that SGD, starting at $\mathcal{V_i}$, enters $\mathcal{V}_j$ the $K$-th time it enters such a neighborhood from the ensemble of $\mathcal{V}_l$'s.
> > >
> > > Now, why this is needed: In Lemmas C.3-4, the transition cost fron $i$ to $j$ may not be finite in some degenerate case. That would be the case if, for instance, to go from $\mathcal{V}_i$ to $\mathcal{V}_j$, SGD cannot avoid $\mathcal{V}_l$. In this case, the transition probability would be 0. However, Lemma D.1 requires positive transition probabilities to be applied. Thanks to assumption 9 (assumption 3 in the main text), the transition probabilities estimated in section C.4 are finite and we can apply Lemma D.1. We will add this discussion in the appendix.
> > >
> > > > 3. What is $K$ in (C.57)?
> > >
> > > $K$ denotes the number of connected components of the critical set of $f$ that are not part of the global minimum, see Def. 1. Note that this is the same $K$ that is used to define the subsamples, or accelerated, sequence that we mention above (Def. 10).
> > >
> > > > 4. (typo?) In Lemma D.3, should it be "For any $i \in V \setminus Q$"? The author is using $i_0$ but the remaining of this lemma is using $i$.
> > >
> > > Indeed, thank you, it should be $i$.
> > >
> > > > 5. What is $\tau_Q$ in (D.37)? Has it been defined before? As we already have Theorem 4, why do we still need to have Theorem 5?
> > >
> > > Thank you, this is indeed a typo and it should read $\tau$ (the $\tau_Q$ was a leftover from an earlier convention). As mentioned in the main text, Thms 5-6 together exactly corresponds to Theorem 1 in the  main text.
> > > Now, as to why Theorem 4 is not enough: as defined at the beginning of section D.2, $\tau_\mathcal{Q}^\eta$ is the hitting time of the global minimum for the process $x^\eta_n$ defined in $(B.17)$ in section B.2, which is a subsampled version of SGD by a factor $1/\eta$. Even though it is straightfoward, Theorem 5 translates the upper bound on the global convergence time from the subsampled process back to original SGD sequence.
> > >
> > > ---
> > > Thank you again for your comments and positive evaluation! We are not in a position to post further replies (we think...), but we are at your disposal - through the AC/SAC or otherwise - if you have any further questions or remarks.
> > >
> > > Regards,
> > >
> > > The authors

---

### Official Review · Reviewer_jzu6 · 2025-03-14

**Overall Recommendation:** 2

**Summary:**

This article analyzes global convergence of SGD on non-convex optimization from the large deviation theory in probability. The question of how long does it take SGD to reach the vicinity of a global minimum of a loss f is studied. A tight theoretical estimation about this time is obtained under suitable assumptions. The contribution of the article is based on non-trivial technical contents. The results rely on the geometry of the loss function, SGD noise and its initialization. It gives an interesting picture of SGD for practitioners.

**Claims And Evidence:**

The main theorem (theorem 1) gives a tight bound of the stopping time tau, when the choice of the set Q is large. I am not able to check the proof as I do not understand the key concept “quasi-potential” which seems not to be standard in the literature.

It seems me that it might also contain some flaw in its definition (eq. 16). Are you sure that the B(x,x’) which takes account only the curves gamma on [0,T], will not go through the set Q over [0,T]? It is not so clear from the definition as in C_T(x,x’,Q), only the points at time n=1..T_1 are restricted to be outside Q. But gamma is a continuous curve, so it might hit the Q at some other time.

**Essential References Not Discussed:**

No

**Experimental Designs Or Analyses:**

Na

**Methods And Evaluation Criteria:**

In eq. 11, the attracting strength is introduced, but it seems not have been used in the article. Could the authors clarify this?

**Other Comments Or Suggestions:**

The current article has a large overlap (page 4-6) with the contents in Azizian+2024. It would better to reduce the overlapped part to focus on the main contributions.

Regarding theorem 1, it would be better to make the statement more precise, by citing properly the needed conditions in appendix.

It is not so clear why the random seed omega in Section 3 (a) line 173, needs to be a compact subset. Could the authors clarify this?


[Azizian+2024]WHAT IS THE LONG-RUN DISTRIBUTION OF STOCHASTIC GRADIENT DESCENT? A LARGE DEVIATIONS ANALYSIS. WAÏSS AZIZIANc,∗, FRANCK IUTZELER♯, JÉRÔME MALICK∗, AND PANAYOTIS MERTIKOPOULOS⋄

**Other Strengths And Weaknesses:**

The theoretical results seem to be very hard to verify numerically since the first-hitting time tau can not be computed without knowing the global optima. Therefore it is unclear how the insights could be transferred to practical training using SGD. It would be good to discuss this aspect in the article.

**Questions For Authors:**

The definition of the Q in eq. 16 is not so clear. Is it the same as the Q in Theorem 1? Or it is defined in eq. 10?
What is the sigma_inf below eq. 7? Is it bar(sigma)?

**Relation To Broader Scientific Literature:**

Na

**Theoretical Claims:**

I did not have time to check the proof.

---

> ### Author Rebuttal · Authors · 2025-04-01
>
> Dear reviewer,
>
> Thank you for your time and input. We reply to your questions and remarks below, and we will of course update our paper accordingly at the first revision opportunity.
>
> - **The notion of the quasi-potential.** This concept plays a central role in the theory of large deviations as developed by Freidlin and Wentzell, see e.g., Chapters 4.2 and 4.3 of [21]. We agree that this definition is not standard in the ML literature: it is for this reason that we devoted pages 4-5 to build up to it, and we would be happy to expand on this in our revision.
> - **Continuous curves hitting Q in non-integer times.** This is a great point, thanks for bringing it up! Indeed, the curves that go in the definition (16) of the quasi-potential are only required to avoid Q at integer times. This is not a mistake: while it is possible for a continuous curve to enter Q at non-integer times, we only need to exclude integer times because we are only interested in the hitting time of the discrete-time process, not its continuous-time interpolations. There is, of course, a lot of technical work that is required to justify this, starting with the large-deviations principle in Appendix B.3 as well the analysis of transitions between critical points in Appendix C (where the peculiar curves you mention are introduced).
> - **The attracting strength.** This notion is only required for the statement of our lower bounds in Thms 1 & 2. The reason is highly technical, and we explain it in detail in Appendices D.3 and D.4.
> - **Numerical verification.** We took advantage of the rebuttal period to perform a series of numerical experiments on some standard global optimization benchmarks, which show an exceedingly good fit between theory and practice, see [here](https://tinyurl.com/2k5ay5c6) (anonymous link). More broadly, we agree that it may be difficult to perform a numerical validation campaign in full generality but, at the same time, this is what we believe is the power of our theory: our results do not provide an upper or lower bound of the convergence time of SGD, but a tight characterization thereof in terms of the features of the problem and the parameters of the method, in cases where nothing can be said otherwise.
>
>   In terms of practical insights, our results already show that the global convergence time scales as exp(1/variance/η) in terms of the method's step-size and the variance of the noise, which is a concrete take-away for practitioners. Moreover, in Section 5, we describe the precise way in which the energy increases as spurious local minima get deeper, which dovetails with results on loss landscapes of neural networks showing that these depths vanish as the number of parameters grows. Thus, even without reaching the overparametrization regime, increasing the number of parameters intrinsically improves the optimization process.
>
>   Overall, we see our framework as a first step towards understanding modern step-size selections strategies, such as periodic schedules or adaptive algorithms, which cannot be otherwised approached theoretically.
> - **Uniformity of notation with [4] (Azizian et al.).** It is true that our choice of notation and terminology in pp. 4-6 follows [4] which, in turn, roughly follows the textbook of Freidlin & Wentzell [21]. This is by design: The setup required for the large deviations machinery is highly non-standard in the ML and optimization literatures, so we wanted to carefully introduce everything. The similarity in notation with [4] was intended to provide an anchor that would make it easier for the reader to connect and compare our setup with [4], which assumptions and definitions are similar, which aren't, and so forth. We made this choice consciously, even at the expense of presenting a more detailed proof sketch and overview of the technical trajectory leading to our results, because we felt that providing a simple pointer to [4] would make for a non-self-contained presentation that would be very difficult to follow. We did not enter a detailed point-by-point "compare-and-contrast" discussion for each assumption and definition in our paper because the focus of our paper and that of [4] is completely different (long-run distribution versus global convergence time), but if you think this would be warranted, we would be happy to provide more details.
> - **Statement of Theorem 1.** Are you referring to the attracting strength? This is defined in (1), L242.
> - **Compactness of Ω.** This is a purely technical assumption facilitating the treatment of the noise - it is standard for minibatch noise and/or inverse transform sampling.
> - **The definition of Q.** Throughout the main, $\mathcal{Q} = \arg\min f$, as per (10). Eq. (16) and Thm 1 refer to that with the caveat that, to lighten notation, we are identifying nodes of the graph with Q itself.
> - **On $\sigma_{\infty}$:** Yes, this was supposed to be $\underline{\sigma}$, thanks!
>
> ---
> Looking forward to a fruitful discussion! Regards,
>
> The authors

---

> > ### Comment · Reviewer_jzu6 · 2025-04-07
> >
> > Thanks for your answers. Could you clarify further the following points?
> > - Continuous curves hitting Q in non-integer times. How do you make correspondence between the continuous curve gamma and the SGD discrete state x_n over time? I thought that the continuous time step should be of the order of the learning rate eta, but you are suggesting that the time step is 1 (this is why only integer times are considered in quasi-potential?). This point is not clear by reading Appendix B.3 and C.
> >
> > - Statement of Theorem 1. Yes, I do refer to the meaning of the attracting strength being large enough. I understand that it is defined in (1), L242. But as it is an important concept, some remarks about it in the main text would be better.

---

> > > ### Author Response · Authors · 2025-04-08
> > >
> > > Dear reviewer,
> > >
> > > Thank you for your follow-up comments, we greatly appreciate your time and interest. We reply to both below:
> > >
> > > > Continuous curves hitting Q in non-integer times. How do you make correspondence between the continuous curve gamma and the SGD discrete state x_n over time? I thought that the continuous time step should be of the order of the learning rate eta, but you are suggesting that the time step is 1 (this is why only integer times are considered in quasi-potential?). This point is not clear by reading Appendix B.3 and C.
> > >
> > > There are several moving parts here:
> > > 1. First, there is the SGD process itself $x_n$, $n=0,1,\dots$, which is the object of ultimate interest for us.
> > > 2. Second, there is the continuous-time interpolation $X(t)$ of $x_n$, which is constructed in the standard way of stochastic approximation, namely
> > >    $$X(t) = x_n + \frac{t-n\eta}{\eta}(x_{n+1} - x_n) \quad \text{for $t\in[n\eta,(n+1)\eta]$}\,.$$
> > >    In words, as far as this continuous-time interpolation is concerned, one iteration of SGD corresponds to $\eta$ units of continuous time or, equivalently, one unit of continuous time corresponds to $\approx 1/\eta$ iterations of SGD (so, for example, if $\eta = 10^{-2}$, $X(1)$ would correspond to $x_{100}$).
> > > 3. Finally, there is the subsampled, accelerated process defined in (B.17), that is
> > >    $$x_n^\eta = x_{\lfloor n/η\rfloor}$$
> > >    This process essentially looks at the iterates of SGD at intervals of width $\approx 1/\eta$, so, modulo unimportant indexing details, $X(k)$ and $x_k^\eta$ coincide. In terms of technical content, this is needed to "wash out" the noise in the short-term time-scale of SGD, in order to then apply the derived large-deviations principle to compare the discrete-time process to action-minimizing continuous-time paths. [Regarding $X(t)$ and $x_n^\eta$, we follow the notation and setup of [4].]
> > >
> > > As you correctly observed, the continuous curves that avoid $\mathcal Q$ at integer times correspond to $X(t)$ avoiding $\mathcal Q$ at integer times, which then translates to the *accelerated process* $x_n^\eta$ avoiding $\mathcal Q$ for all $n$ less than the estimated hitting time. It is also correct that this does not immediately translate to $x_n$ avoiding $\mathcal Q$: to carry out this comparison, we need the additional machinery that we develop in D.3 where Thm. 5 and 6 provide our main result in terms of the actual SGD sequence.
> > >
> > >
> > > >Statement of Theorem 1. Yes, I do refer to the meaning of the attracting strength being large enough. I understand that it is defined in (1), L242. But as it is an important concept, some remarks about it in the main text would be better.
> > >
> > > Agreed. The reason we did not include any more details in the first place was lack of space. We will be happy to take advantage of the first revision opportunity to provide a richer and more detailed presentation for both matters above - which actually dovetails very well with the technical flowchart of the proof that we prepared in our response to Reviewer U6wN [here](https://tinyurl.com/mb4r9udc).
> > >
> > > ---
> > > Thank you again for your follow-up comments! The way that OpenReview has been set up, we will not be in a position to post further replies, but we are at your disposal - through the AC/SAC or otherwise - if you have any further questions or remarks.
> > >
> > > Regards,
> > >
> > > The authors

---

### Decision · Program_Chairs · 2025-05-01

**Decision:**

Accept (poster)

**Comment:**

This paper analyzes the time required for the global convergence of SGD under general non-convex functions. The main result provides a bound on the hitting time, indicating the time needed to reach the set of minima. The proof techniques rely on large deviation theory and the dynamical system theory of random perturbations. While many components are relatively standard, the proofs are generally well-written and easy to follow.

The reviewers raised several questions regarding presentation and clarity, but no critical technical flaws were identified. Although the novelty of the proof techniques remains somewhat unclear, the overall explanations are careful and have been evaluated positively by the reviewers on average. Furthermore, the authors' responses were clear and facilitated good communication throughout the review process.